# A VIEW OF MINI-BATCH SGD VIA GENERATING FUNCTIONS: CONDITIONS OF CONVERGENCE, PHASE TRANSITIONS, BENEFIT FROM NEGATIVE MOMENTA

**Maksim Velikanov** [*]
Technology Innovation Institute,
CMAP, Ecole Polytechnique
maksim.velikanov@tii.ae

**Denis Kuznedelev**
Skoltech,
Yandex Research
denis.kuznedelev@skoltech.ru

**Dmitry Yarotsky**
Skoltech
d.yarotsky@skoltech.ru

## ABSTRACT

Mini-batch SGD with momentum is a fundamental algorithm for learning large predictive models. In this paper we develop a new analytic framework to analyze noise-averaged properties of mini-batch SGD for linear models at constant learning rates, momenta and sizes of batches. Our key idea is to consider the dynamics of the second moments of model parameters for a special family of "Spectrally Expressible" approximations. This allows to obtain an explicit expression for the generating function of the sequence of loss values. By analyzing this generating function, we find, in particular, that 1) the SGD dynamics exhibits several convergent and divergent regimes depending on the spectral distributions of the problem; 2) the convergent regimes admit explicit stability conditions, and explicit loss asymptotics in the case of power-law spectral distributions; 3) the optimal convergence rate can be achieved at negative momenta. We verify our theoretical predictions by extensive experiments with MNIST, CIFAR10 and synthetic problems, and find a good quantitative agreement.

## 1 INTRODUCTION

We consider a classical mini-batch Stochastic Gradient Descent (SGD) algorithm (Robbins & Monro, 1951; Bottou & Bousquet, 2007) with momentum (Polyak, 1964):

$$\mathbf{w}_{t+1} = \mathbf{w}_t + \mathbf{v}_{t+1}, \quad \mathbf{v}_{t+1} = -\alpha_t \nabla_{\mathbf{w}} L_{B_t}(\mathbf{w}_t) + \beta_t \mathbf{v}_t. \tag{1}$$

Here, $L_B(\mathbf{w}) = \frac{1}{b} \sum_{i=1}^b l(f(\mathbf{x}_i, \mathbf{w}), y_i)$ is the sampled loss of a model $\widehat{y} = f(\mathbf{x}, \mathbf{w})$, computed using a pointwise loss $l(\widehat{y}, y)$ on a mini-batch $B = \{(\mathbf{x}_i, y_i)\}_{s=1}^b$ of $b$ data points representing the target function $y = f^*(\mathbf{x})$. The *momentum* term $\mathbf{v}_n$ represents information about gradients from previous iterations and is well-known to significantly improve convergence both generally (Polyak, 1987) and for neural networks (Sutskever et al., 2013). Re-sampling of the mini-batch $B_t$ at each SGD iteration $t$ creates a specific gradient noise, structured according to both the local geometry of the model $f(\mathbf{x}, \mathbf{w})$ and the quality of current approximation $\widehat{y}$. In the context of modern deep learning, $f(\mathbf{x}, \mathbf{w})$ is usually very complex, and the quantitative prediction of the SGD behavior becomes a challenging task that is far from being complete at the moment.

Our goal is to obtain explicit expressions characterizing the average case convergence of mini-batch SGD for the classical least-squares problem of minimizing quadratic objective $L(\mathbf{w})$. This setup is directly related to modern neural networks trained with a quadratic loss function, since networks can often be well described – e.g., in the large-width limit (Jacot et al., 2018; Lee et al., 2019) or during the late stage of training (Fort et al., 2020) – by their linearization w.r.t. parameters $\mathbf{w}$.

---

[*] A significant part of the work was done while at Skoltech.

A fundamental way to characterize least squares problems is through their *spectral distributions*: the eigenvalues $\lambda_k$ of the Hessian and the coefficients $c_k$ of the expansion of the optimal solution $\mathbf{w}^*$ over the Hessian eigenvectors. Then, one can estimate certain metrics of the problem through *spectral expressions*, i.e. explicit formulas that operate with spectral distributions $\lambda_k, c_k$ but not with other details of the solution $\mathbf{w}^*$ or the Hessian. A simple example is the standard stability condition for full-batch gradient descent (GD): $\alpha < 2/\lambda_{\max}$. Various exact or approximate spectral expressions are available for full-batch GD-based algorithms (Fischer, 1996) and ridge regression (Canatar et al., 2021; Wei et al., 2022). Here, we aim at obtaining spectral expressions and associated results (stability conditions, phase structure, loss asymptotics,...) for average train loss under mini-batch SGD.

An important feature of spectral distributions in deep learning problems is that they often obey macroscopic laws – quite commonly a power law with a long tail of eigenvalues converging to 0 (see Cui et al. (2021); Bahri et al. (2021); Kopitkov & Indelman (2020); Velikanov & Yarotsky (2021); Atanasov et al. (2021); Basri et al. (2020) and Figs. 1, 9). The typically simple form of macroscopic laws allows to theoretically analyze spectral expressions and obtain fine-grained results.

As an illustration, consider the full-batch GD for least squares regression on a MNIST dataset. Standard optimization results (Polyak, 1987) do not take into account fine spectral details and give either non-strongly convex bound $L_{\mathrm{GD}}(\mathbf{w}_t) = O(t^{-1})$ or strongly-convex bound $L_{\mathrm{GD}}(\mathbf{w}_t) \leq L(\mathbf{w}_0)(\frac{\lambda_{\max} - \lambda_{\min}}{\lambda_{\max} + \lambda_{\min}})^{2t}$. Both these bounds are rather crude and poorly agree with the experimentally observed (Bordelon & Pehlevan, 2021; Velikanov & Yarotsky, 2022) loss trajectory which can be approximately described as $L(\mathbf{w}_t) \sim Ct^{-\xi},\ \xi \approx 0.25$ (cf. our Fig. 1). In contrast, fitting power-laws to both eigenvalues $\lambda_k$ and coefficients $c_k$ and using the spectral expression $L_{\mathrm{GD}}(\mathbf{w}_t) = \sum_k (1 - \alpha\lambda_k)^{2t}\lambda_k c_k^2$ allows to accurately predict both exponent $\xi$ and constant $C$. Accordingly, one of the purposes of the present paper is to investigate whether similar predictions can be made for mini-batch SGD under power-law spectral distributions.

**Outline and main contributions.** We develop a new, spectrum based analytic approach to the study of mini-batch SGD. The results obtained within this approach and its key steps are naturally divided into three parts:

1. We show that in contrast to the full-batch GD, loss trajectories of the mini-batch SGD cannot be determined merely from the spectral properties of the problem. To overcome this difficulty, we propose a natural family of **Spectrally Expressible (SE)** approximations for SGD dynamics that admit an analytic solution. We provide multiple justifications for these approximations, including theoretical scenarios where they are exact and empirical evidence of their accuracy for describing optimization of models on MNIST and CIFAR10.

2. To characterize SGD dynamics under SE approximation, we derive explicit spectral expressions for the **generating function** of the sequence of loss values, $\widetilde{L}(z) \equiv \sum_t L(\mathbf{w}_t)z^t$, and show that it decomposes into the "signal" $\widetilde{V}(z)$ and "noise" $\widetilde{U}(z)$ generating functions. Analyzing $\widetilde{U}(z)$, we derive a novel **stability condition** of mini-batch SGD in terms of only the problem spectrum $\lambda_k$. In the practically relevant case of large momentum parameter $\beta \approx 1$, stability condition simplifies to the restriction of effective learning rate $\alpha_{\mathrm{eff}} \equiv \frac{\alpha}{1-\beta} < \frac{2b}{\lambda_{\mathrm{crit}}}$ with some critical value $\lambda_{\mathrm{crit}}$ determined by the spectrum. Finally, we find the characteristic **divergence time** when stability condition is violated.

3. By assuming power-law distributions for both eigenvalues $\lambda_k \propto k^{-\nu}$ and coefficient partial sums $S_k = \sum_{l \leq k} \lambda_l c_l^2 \propto k^{-\varkappa}$, we show that SGD exhibits distinct **"signal-dominated"** and **"noise-dominated"** convergence regimes (previously known for SGD without momenta (Varre et al., 2021)) depending on the sign of $\varkappa + 1 - 2\nu$. For both regimes we obtain power-law loss convergence rates and find the explicit constant in the leading term. Using these rates, we demonstrate a **dynamical phase transition** between the phases and find its characteristic transition time. Finally, we analyze optimal hyper parameters in both phases. In particular, we show that **negative momenta** can be beneficial in the "noise-dominated" phase but not in the "signal-dominated" phase.

We discuss related work in Appendix A and experimental details [1] in Appendix F.

---

[1] Our code: `https://anonymous.4open.science/r/PowerLawOptimization-1401/`

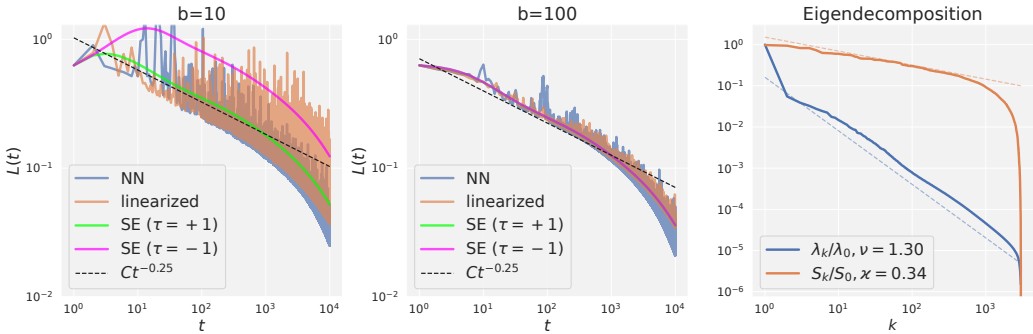

Figure 1: Optimization trajectories and spectral distributions for MNIST. **Left, center**: Loss trajectories of different training regimes for batch sizes 10 and 100 (see Sec. 3). We see good agreement of NN with SE approximation at $\tau = 1$. **Right**: Eigenvalues $\lambda_k$ and partial sums $S_k = \sum_{l=k}^{N} \lambda_l C_{ll,0}$ characterizing $\mathbf{H}$ and $\mathbf{C}_0$. We see good agreement with assumed spectral power laws $\lambda_k \approx \Lambda k^{-\nu}, S_k \approx K k^{-\varkappa}$ as in Eq. (25). We estimate the values $\Lambda, K, \nu, \varkappa$ from the curves in the right figure. The dotted lines in the left and central figures then show $L_{\text{approx}}(t) = Ct^{-\zeta}$, with $C$ computed as $C_{\text{signal}}$ in Eq. (28a) and $\zeta = \frac{\varkappa}{\nu} \approx 0.25$. We see good agreement between Theorem 2 and experiment. Similar experiments for CIFAR10 with ResNet/MobileNet are given in Fig. 9 and also agree well with theory.

## 2 THE SETTING AND SGD DYNAMICS

We consider a linear model $f(\mathbf{x}, \mathbf{w}) = \langle \mathbf{w}, \boldsymbol{\psi}(\mathbf{x}) \rangle$ with non-linear features $\boldsymbol{\psi}(\mathbf{x})$ trained to approximate a target function $y^*(\mathbf{x})$ by minimizing quadratic loss $L(\mathbf{w}) = \frac{1}{2N} \sum_{i=1}^{N} (\langle \mathbf{w}, \boldsymbol{\psi}(\mathbf{x}_i) \rangle - y^*(\mathbf{x}_i))^2$ over training dataset $\mathcal{D} = \{\mathbf{x}_i\}_{i=1}^{N}$. The model's Jacobian is given by $\boldsymbol{\Psi} = \big( \boldsymbol{\psi}(\mathbf{x}_i) \big)_{i=1}^{N}$, the Hessian $\mathbf{H} = \frac{1}{N} \sum_{i=1}^{N} \boldsymbol{\psi}(\mathbf{x}_i) \otimes \boldsymbol{\psi}(\mathbf{x}_i) = \frac{1}{N} \boldsymbol{\Psi} \boldsymbol{\Psi}^T$ and the tangent kernel $K(\mathbf{x}, \mathbf{x}') = \langle \boldsymbol{\psi}(\mathbf{x}), \boldsymbol{\psi}(\mathbf{x}') \rangle$. Assuming that the target function is representable as $y^*(\mathbf{x}) = \langle \mathbf{w}^*, \boldsymbol{\psi}(\mathbf{x}) \rangle$, the loss takes the form $L(\mathbf{w}) = \frac{1}{2} \langle \mathbf{w} - \mathbf{w}^*, \mathbf{H}(\mathbf{w} - \mathbf{w}^*) \rangle$. We allow model parameters to be either finite dimensional vectors or belong to a Hilbert space $\mathcal{H}$. Similarly, dataset $\mathcal{D}$ can be either finite ($N < \infty$) or infinite, and in the latter case we only require a finite norm of the target function $y^*(\mathbf{x})$ but not of the solution $\mathbf{w}^*$. Note that this setting is quite rich and can accommodate kernel methods, linearization of real neural networks, and infinitely wide neural networks in the NTK regime. See Appendix B for details. In the experimental results, we refer to this setting as *linearized* (to be distinguished from nonlinear NN's and the SE approximation introduced in the sequel).

Denoting deviation from the optimum as $\Delta \mathbf{w} \equiv \mathbf{w} - \mathbf{w}^*$, we can write a single SGD step at iteration $t$ with a randomly chosen batch $B_t$ of size $b$ as

$$\begin{pmatrix} \Delta \mathbf{w}_{t+1} \\ \mathbf{v}_{t+1} \end{pmatrix} = \begin{pmatrix} \mathbf{I} - \alpha_t \mathbf{H}(B_t) & \beta_t \mathbf{I} \\ -\alpha_t \mathbf{H}(B_t) & \beta_t \mathbf{I} \end{pmatrix} \begin{pmatrix} \Delta \mathbf{w}_t \\ \mathbf{v}_t \end{pmatrix}, \quad \mathbf{H}(B_t) \equiv \frac{1}{b} \sum_{i \in B_t} \boldsymbol{\psi}(\mathbf{x}_i) \otimes \boldsymbol{\psi}(\mathbf{x}_i). \quad (2)$$

An important feature of the multiplicative noise introduced by the random choice of $B_t$ is that the $n$'th moments of parameter and momentum vectors $\Delta \mathbf{w}_{t+1}, \mathbf{v}_{t+1}$ in dynamics (2) are fully determined from the same moments at the previous step $t$. As our loss function is quadratic, we focus on the second moments

$$\mathbf{C} \equiv \mathbb{E}[\Delta \mathbf{w} \otimes \Delta \mathbf{w}], \quad \mathbf{J} \equiv \mathbb{E}[\Delta \mathbf{w} \otimes \mathbf{v}], \quad \mathbf{V} \equiv \mathbb{E}[\mathbf{v} \otimes \mathbf{v}], \quad \mathbf{M} \equiv \begin{pmatrix} \mathbf{C} & \mathbf{J} \\ \mathbf{J}^\dagger & \mathbf{V} \end{pmatrix}. \quad (3)$$

Then the average loss is $\mathbb{E}[L(\mathbf{w})] = \frac{1}{2} \text{Tr}(\mathbf{H} \mathbf{C})$ and for combined second moment matrix we have

**Proposition 1.** *Consider SGD* (1) *with learning rates* $\alpha_t$, *momentum* $\beta_t$, *batch size* $b$ *and random uniform choice of the batch* $B_t$. *Then the update of second moments* (3) *is*

$$\begin{pmatrix} \mathbf{C}_{t+1} & \mathbf{J}_{t+1} \\ \mathbf{J}_{t+1}^\dagger & \mathbf{V}_{t+1} \end{pmatrix} = \begin{pmatrix} \mathbf{I} - \alpha_t \mathbf{H} & \beta_t \mathbf{I} \\ -\alpha_t \mathbf{H} & \beta_t \mathbf{I} \end{pmatrix} \begin{pmatrix} \mathbf{C}_t & \mathbf{J}_t \\ \mathbf{J}_t^\dagger & \mathbf{V}_t \end{pmatrix} \begin{pmatrix} \mathbf{I} - \alpha_t \mathbf{H} & \beta_t \mathbf{I} \\ -\alpha_t \mathbf{H} & \beta_t \mathbf{I} \end{pmatrix}^T + \gamma \alpha_t^2 \begin{pmatrix} \boldsymbol{\Sigma}(\mathbf{C}_t) & \boldsymbol{\Sigma}(\mathbf{C}_t) \\ \boldsymbol{\Sigma}(\mathbf{C}_t) & \boldsymbol{\Sigma}(\mathbf{C}_t) \end{pmatrix}$$
$$(4)$$

Here $\gamma\mathbf{\Sigma}(\mathbf{C}_t)$ is the covariance of gradient noise due to mini-batch sampling, with $\gamma = \frac{N-b}{(N-1)b}$ and

$$\mathbf{\Sigma}(\mathbf{C}) = \frac{1}{N}\sum_{i=1}^{N}\langle\boldsymbol{\psi}(\mathbf{x}_i),\mathbf{C}\boldsymbol{\psi}(\mathbf{x}_i)\rangle\boldsymbol{\psi}(\mathbf{x}_i)\otimes\boldsymbol{\psi}(\mathbf{x}_i) - \mathbf{HCH}. \tag{5}$$

## 3 SPECTRALLY EXPRESSIBLE APPROXIMATION

An important property of non-stochastic GD is that it can be "solved" by diagonalizing the Hessian $\mathbf{H}$. Namely, typical quantities of interest in the context of optimization have the form $\mathrm{Tr}[\phi(\mathbf{H})\mathbf{C}]$ with some function $\phi$: e.g., $\phi(x) = \frac{x}{2}$ for the loss, $\phi(x) = 1$ for the parameter displacement, $\phi(\mathbf{x}) = x^2$ for the squared gradient norm. Given a Hessian $\mathbf{H}$, by Riesz–Markov theorem we can associate to a positive semi-definite $\mathbf{C}$ the respective *spectral measure* $\rho_\mathbf{C}(d\lambda)$:

$$\mathrm{Tr}[\phi(\mathbf{H})\mathbf{C}] = \int\phi(\lambda)\rho_\mathbf{C}(d\lambda), \quad \forall\phi\in C(\mathrm{spec}(\mathbf{H})). \tag{6}$$

We will assume that $\mathbf{H}$ has an eigenbasis $\mathbf{u}_k$ with eigenvalues $\lambda_k$; in this case $\rho_\mathbf{C} = \sum_k C_{kk}\delta_{\lambda_k}$, where $C_{kk} = \langle\mathbf{u}_k,\mathbf{C}\mathbf{u}_k\rangle$. For vanilla GD, the spectral measure of the second moment matrix $\mathbf{C}_t$ at iteration $t$ is given by $\rho_{\mathbf{C}_t}(d\lambda) = p_t^2(\lambda)\rho_{\mathbf{C}_0}(d\lambda)$, where $p_t(x)$ is a polynomial fully determined by the learning algorithm (Fischer, 1996). Thus, the information encoded in the initial spectral measure $\rho_{\mathbf{C}_0}$ is, in principle, sufficient to compute main characteristics of the optimization trajectories.

This conclusion, however, does not hold for SGD, since the first component of the noise term (5) uses finer, non-spectral details of the problem. For a problem with fixed spectrum $\lambda_k$ and measure $\rho_\mathbf{C}$, we show (Sec. D.1) that the spectral components $\Sigma_{kk} = \langle\mathbf{u}_k,\mathbf{\Sigma}\mathbf{u}_k\rangle$ of the noise term (5) can take any value in the wide interval $\left[\lambda_k(\mathrm{Tr}[\mathbf{HC}] - \lambda_k C_{kk}),(N-1)\lambda_k^2 C_{kk}\right]$, so that non-spectral details can make a strong impact on the optimization trajectory.

Yet, in many theoretical and practical cases (see below), we find that non-spectral details are not significant, and components $\Sigma_{kk}$ can be reduced to a spectral expression. Specifically, for a certain choice of parameters $\tau_1,\tau_2\in\mathbb{R},\tau_1\geq\tau_2$, the noise measure $\rho_\mathbf{\Sigma}$ is determined by $\rho_\mathbf{C}$ via

$$\rho_\mathbf{\Sigma}(d\lambda) = \tau_1\Big(\int\lambda'\rho_\mathbf{C}(d\lambda')\Big)\rho_\mathbf{H}(d\lambda) - \tau_2\lambda^2\rho_\mathbf{C}(d\lambda), \quad \rho_\mathbf{H} = \sum_k\lambda_k\delta_{\lambda_k}. \tag{7}$$

We refer to this as the *Spectrally Expressible (SE)* approximation. Under this approximation we again can, in principle, fully reconstruct the trajectory of observables $\mathrm{Tr}[\phi(\mathbf{H})\mathbf{C}_t]$ from the initial $\mathbf{C}_0$. There are multiple reasons justifying this approximation.

**1)** There are scenarios where Eq. (7) holds exactly. First, we prove this for translation-invariant models, with $\tau_1 = 1, \tau_2 = 1$ (see Sec. D.5):

**Proposition 2.** *Consider a problem with regular grid dataset* $\mathcal{D} = \{\mathbf{x_i} = (\frac{2\pi i_1}{N_1},\ldots,\frac{2\pi i_d}{N_d})\}$, $\mathbf{i}\in(\mathbb{Z}/N_1\mathbb{Z})\times\cdots\times(\mathbb{Z}/N_d\mathbb{Z})$ *on the d-dimensional torus* $\mathbb{T}^d = (\mathbb{R}/2\pi\mathbb{Z})^d$ *and with translation invariant kernel* $K(\mathbf{x},\mathbf{x}') = K(\mathbf{x} - \mathbf{x}')$. *Then (7) holds exactly with* $\tau_1 = 1, \tau_2 = 1$.

**2)** Also, SE approximation with $\tau_1 = \tau_2 = 1$ generally results if pointwise losses $L(\mathbf{x})$ are statistically independent of feature eigencomponents $\psi_k^2(\mathbf{x}) = \langle\mathbf{u}_k,\boldsymbol{\psi}(\mathbf{x})\rangle^2$ w.r.t. $\mathbf{x}\sim\mathcal{D}$ (Sec. D.2).

**3)** On the other hand, Bordelon & Pehlevan (2021) show that when the features $\boldsymbol{\psi}(\mathbf{x})$ are Gaussian w.r.t. $\mathbf{x}\sim\mathcal{D}$, the SE approximation is exact with $\tau_1 = 1, \tau_2 = -1$.

**4)** Eq. (7) can be derived by considering a generalization of the map (5) and requiring it to be "spectrally expressible". Namely, replace summation over $i$ in (5) by a general linear combination with coefficients $\widehat{R} = (R_{i_1i_2i_3i_4})$ only assuming the symmetries $R_{i_1i_2i_3i_4} = R_{i_2i_1i_3i_4} = R_{i_1i_2i_4i_3}$:

$$\mathbf{\Sigma}_{\widehat{R}}(\mathbf{C}) = \frac{1}{N^2}\sum_{i_1,i_2,i_3,i_4=1}^{N}R_{i_1i_2i_3i_4}\langle\boldsymbol{\psi}(\mathbf{x}_{i_3}),\mathbf{C}\boldsymbol{\psi}(\mathbf{x}_{i_4})\rangle\boldsymbol{\psi}(\mathbf{x}_{i_1})\otimes\boldsymbol{\psi}(\mathbf{x}_{i_2}) - \mathbf{HCH}. \tag{8}$$

Assuming the coefficients $\widehat{R}$ to be fixed and independent of $\mathbf{\Psi},\mathbf{H},\mathbf{C}$, we prove

**Theorem 1.** *For an SGD dynamics* (4) *with noise term* (8)*, the following statements are equivalent:* **(1)** *Spectral measures $\rho_{\mathbf{C}_t}$ occurring during SGD are uniquely determined by the initial spectral measure $\rho_{\mathbf{C}_0}$ and by the eigenvalues $\lambda_k$ of Hessian $\mathbf{H}$;* **(2)** $R_{i_1 i_2 i_3 i_4} = \tau_1 \delta_{i_1 i_2} \delta_{i_3 i_4} - \frac{\tau_2 - 1}{2}(\delta_{i_1 i_3}\delta_{i_2 i_4} + \delta_{i_1 i_4}\delta_{i_2 i_3})$ *for some $\tau_1, \tau_2$, and Eq.* (7) *holds;* **(3)** *The trajectory $\{\mathbf{C}_t\}_{t=0}^{\infty}$ is invariant under transformations $\widetilde{\mathbf{\Psi}} = \mathbf{\Psi}\mathbf{U}^T$ of the feature matrix $\mathbf{\Psi}$ by any orthogonal matrix $\mathbf{U}$.*

**5)** Finally, we observe empirically that SE approximation works well on realistic problems such as MNIST and CIFAR10 for predicting loss evolution (Figs. 1, 9, 10 and Sec. F.4), determination of convergence conditions (Fig. 2) and approximation of original noise term (5) in terms of trace norm (Sec. F.4).

In the remainder we focus on analytical treatment of SGD dynamics under SE approximation (7). Note that parameters $\gamma, \tau_1, \tau_2$ enter evolution equations in combinations $\gamma\tau_1, \gamma\tau_2$. This allows to lighten the notation in the sequel by setting $\tau_1 = 1$ and $\tau_2 = \tau$ without restricting generality.

## 4 REDUCTION TO GENERATING FUNCTIONS

**Gas of rank-1 operators.** Iterations (4) are linear and can be compactly written as $\mathbf{M}_{t+1} = F_t \mathbf{M}_t$ with a linear operator $F_t$. Under SE approximation (7), for all four blocks of $\mathbf{M}_t$ we only need to consider the diagonal components in the eigenbasis of $\mathbf{H}$, e.g. $C_{kk,t} \equiv \langle \mathbf{u}_k, \mathbf{C}_t \mathbf{u}_k \rangle$. Then

$$L(T) = \tfrac{1}{2}\operatorname{Tr}\mathbf{H}\mathbf{C}_T = \tfrac{1}{2}\langle(\begin{smallmatrix}\boldsymbol{\lambda} & 0 \\ 0 & 0\end{smallmatrix}), F_T F_{T-1}\cdots F_1(\begin{smallmatrix}\mathbf{C}_0 & 0 \\ 0 & 0\end{smallmatrix})\rangle. \tag{9}$$

Here $(\begin{smallmatrix}\boldsymbol{\lambda} & 0 \\ 0 & 0\end{smallmatrix}) = ((\begin{smallmatrix}\lambda_1 & 0 \\ 0 & 0\end{smallmatrix}),(\begin{smallmatrix}\lambda_2 & 0 \\ 0 & 0\end{smallmatrix}),\dots), (\begin{smallmatrix}\mathbf{C}_0 & 0 \\ 0 & 0\end{smallmatrix}) = ((\begin{smallmatrix}C_{11,0} & 0 \\ 0 & 0\end{smallmatrix}),(\begin{smallmatrix}C_{22,0} & 0 \\ 0 & 0\end{smallmatrix}),\dots) \in \mathbb{R}^{\mathbb{N}} \otimes \mathbb{R}^{2\times 2}$ are $(2\times2)$-matrix-valued sequences indexed by the eigenvalues $\lambda_k$, and $\langle\cdot,\cdot\rangle$ is the natural scalar product in $l^2 \otimes \mathbb{R}^{2\times2}$. The evolution operator $F_t$ can be represented as the sum $F_t = Q_t + A_t$ of a rank-one operator $Q_t$ (noise term) and an operator $A_t$ (main term) that acts independently at each eigenvalue $\lambda_k$ as a $4 \times 4$ matrix:

$$Q_t = \gamma_t \alpha_t^2 \boldsymbol{\lambda}(\begin{smallmatrix}1 & 1 \\ 1 & 1\end{smallmatrix})\langle(\begin{smallmatrix}\boldsymbol{\lambda} & 0 \\ 0 & 0\end{smallmatrix}),\cdot\rangle, \tag{10}$$

$$A_t = (A_{t,\lambda}), \quad A_{t,\lambda}M = (\begin{smallmatrix}1-\alpha_t\lambda & \beta_t \\ -\alpha_t\lambda & \beta_t\end{smallmatrix})M(\begin{smallmatrix}1-\alpha_t\lambda & \beta_t \\ -\alpha_t\lambda & \beta_t\end{smallmatrix})^T - \tau\gamma_t\alpha_t^2\lambda^2(\begin{smallmatrix}1 & 0 \\ 1 & 0\end{smallmatrix})M(\begin{smallmatrix}1 & 0 \\ 1 & 0\end{smallmatrix})^T, \tag{11}$$

where $\boldsymbol{\lambda} = (\lambda_1, \lambda_2, \dots)^T$. Let us expand the loss by the binomial formula, with $m$ terms $Q_t$ chosen at positions $t_1, \dots, t_m$ and $T - m$ terms $A_t$ at the remaining positions:

$$L(T) = \frac{1}{2}\sum_{m=0}^{T}\sum_{0<t_1<\dots<t_m<T+1} U_{T+1,t_m}U_{t_m,t_{m-1}}\cdots U_{t_2,t_1}V_{t_1}, \tag{12}$$

where

$$U_{t,s} = \langle(\begin{smallmatrix}\boldsymbol{\lambda} & 0 \\ 0 & 0\end{smallmatrix}), A_{t-1}A_{t-2}\cdots A_{s+1}\boldsymbol{\lambda}(\begin{smallmatrix}1 & 1 \\ 1 & 1\end{smallmatrix})\rangle\gamma_s\alpha_s^2, \tag{13}$$

$$V_t = \langle(\begin{smallmatrix}\boldsymbol{\lambda} & 0 \\ 0 & 0\end{smallmatrix}), A_{t-1}A_{t-2}\cdots A_1(\begin{smallmatrix}\mathbf{C}_0 & 0 \\ 0 & 0\end{smallmatrix})\rangle. \tag{14}$$

Expansion (12) has a suggestive interpretation as a partition function of a gas of rank-1 operators interacting with nearest neighbors (via $U_{t,s}$) and the origin (via $V_t$). Interactions via $U_{t,s}$ contain the factor $\gamma$ so that their strength depends on the sampling noise. We expect the model to be in different phases depending on the amount of noise: in the "signal-dominated" regime $L(T)$ is primarily determined by $V_t$ at large $T$, while in the "noise-dominated" regime $L(T)$ is primarily determined by $U_{t,s}$. We show later that such a phase transition indeed occurs for non-strongly convex problems.

**Generating functions.** Expansion (12) allows to compute $L(T)$ iteratively:

$$L(T) = \frac{1}{2}V_{T+1} + \sum_{t_m=1}^{T} U_{T+1,t_m}L(t_m - 1). \tag{15}$$

For constant learning rates and momenta $\alpha_t \equiv \alpha, \beta_t \equiv \beta$ we have $F_t \equiv F$, the value $U_{t,s}$ becomes translation invariant, $U_{t,s} = U_{t-s}$. Then, as we show in Sec. E.1, the loss can be conveniently described by generating functions (or equivalently, Laplace transform):

$$\widetilde{L}(z) = \sum_{t=0}^{\infty} L(t)z^t = \frac{\widetilde{V}(z)/2}{1 - z\widetilde{U}(z)}, \tag{16}$$

where $\widetilde{U}$ and $\widetilde{V}$ are the "noise" and "signal" generating functions:

$$\widetilde{U}(z) = \sum_{t=0}^{\infty} U_{t+1} z^t = \gamma \alpha^2 \langle (\begin{smallmatrix} \boldsymbol{\lambda} & 0 \\ 0 & 0 \end{smallmatrix}), (1-zA)^{-1} \boldsymbol{\lambda} (\begin{smallmatrix} 1 & 1 \\ 1 & 1 \end{smallmatrix}) \rangle \tag{17}$$

$$= \gamma \alpha^2 \sum_k \lambda_k^2 (\beta z + 1)/S(\alpha, \beta, \tau\gamma, \lambda_k, z), \tag{18}$$

$$\widetilde{V}(z) = \sum_{t=0}^{\infty} V_{t+1} z^t = \langle (\begin{smallmatrix} \boldsymbol{\lambda} & 0 \\ 0 & 0 \end{smallmatrix}), (1-zA)^{-1} (\begin{smallmatrix} \mathbf{C}_0 & 0 \\ 0 & 0 \end{smallmatrix}) \rangle \tag{19}$$

$$= \sum_k \lambda_k C_{kk,0} (2\alpha\beta\lambda_k z + \beta^3 z^2 - \beta^2 z - \beta z + 1)/S(\alpha, \beta, \tau\gamma, \lambda_k, z) \tag{20}$$

$$S(\alpha, \beta, \gamma, \lambda, z) = \alpha^2 \beta \gamma \lambda^2 z^2 + \alpha^2 \beta \lambda^2 z^2 + \alpha^2 \gamma \lambda^2 z - \alpha^2 \lambda^2 z - 2\alpha\beta^2 \lambda z^2 - 2\alpha\beta\lambda z^2$$
$$+ 2\alpha\beta\lambda z + 2\alpha\lambda z - \beta^3 z^3 + \beta^3 z^2 + \beta^2 z^2 - \beta^2 z + \beta z^2 - \beta z - z + 1.$$

In the remainder we derive various properties of the loss evolution by analyzing these formulas.

## 5  STABILITY ANALYSIS

Starting from this section we assume a non-strongly-convex scenario with an infinite-dimensional and compact $\mathbf{H}$, i.e., $\lambda_k \to 0$. Also, we assume that $0 < \tau \le 1$ in SE approximations: this simplifies some statements and fits the experiment for practical problems (see Figure 1 and Appendix F).

**Conditions of loss convergence (stability).** First note that as $\lambda_k \to 0$ the "main" components $A_{\lambda_k}$ of the evolution operator $F$ appearing in Eq. (11) have their action in trial matrix $M$ converge to $(\begin{smallmatrix} 1 & \beta \\ 0 & \beta \end{smallmatrix}) M (\begin{smallmatrix} 1 & \beta \\ 0 & \beta \end{smallmatrix})^T$, and from (13) we get $U_t = const \sum_{k=1}^{\infty} \lambda_k^2 (1 + o(1))$, for each $t$. This means that if $\sum_k \lambda_k^2 = \infty$ then each $U_t = \infty$ and loss diverges already at first step ($L(t=1) = \infty$) – we call this effect *immediate divergence*. On the other hand, assuming $\sum_k \lambda_k^2 < \infty$, loss stability can be related to the first positive singularity of the loss generating function $\widetilde{L}(z)$ given by (16). Indeed, let $r_L$ be the convergence radius of power series (16). Since $L(t) \ge 0$, $\widetilde{L}(z)$ must be monotone increasing on $[0, r_L]$ and have a singularity at $z = r_L$. If $r_L < 1$, then $L(t) \nrightarrow 0$ as $t \to \infty$, while if $r_L > 1$, then $L(t) \to 0$ exponentially fast. Thus, large-$t$ loss stability can be characterized by the condition $r_L \ge 1$, which can be further related to the generating function $\widetilde{U}(z)$.

**Proposition 3.** *Let $\beta \in (-1, 1)$, $0 < \alpha < 2(\beta+1)/\lambda_{\max}$ and $\gamma \in [0, 1]$. Then $r_L < 1$ iff $\widetilde{U}(1) > 1$.*

This result is proved by showing that $\widetilde{U}(z), \widetilde{V}(z)$ have no singularities for $z \in (0, 1)$ and that $z\widetilde{U}(z)$ is monotone increasing on $(0, 1)$. The only source of singularity in $\widetilde{L}(z)$ is then the denominator in (16). Note that conditions $\beta \in (-1, 1)$, $0 < \alpha < 2(\beta + 1)/\lambda_{\max}$ exactly specify the convergence region for the non-stochastic problem with $\gamma = 0$ (see (Roy & Shynk, 1990; Tugay & Tanik, 1989) and Appendix H). This is also a necessary condition in the presence of sampling noise since the second term in (4) is a positive semi-definite matrix.

To get a better understanding of convergence condition $\widetilde{U}(1) < 1$ we use Eq. (18) at $z = 1$ and find

$$\widetilde{U}(1) = \sum_k \frac{\lambda_k}{\lambda_k (\tau - \frac{1-\beta}{\gamma(1+\beta)}) + \frac{2(1-\beta)}{\alpha\gamma}} = \frac{\alpha\gamma}{2(1-\beta)} \sum_k \lambda_k (1 + O(\lambda_k)). \tag{21}$$

This shows that $\sum_k \lambda_k = \infty$ iff $\widetilde{U}(1) = \infty > 1$ (regardless of values of $\alpha, \beta, \gamma$), in which case $r_L < 1$ and $L(t)$ diverges exponentially as $t \to \infty$. We refer to this scenario as *eventual divergence*.

Next, consider *effective learning rate* $\alpha_{\text{eff}} \equiv \frac{\alpha}{1-\beta} = \alpha + \alpha\beta + \alpha\beta^2 + \ldots$ which reflects the accumulated effect of previous gradient descent iterations (Tugay & Tanik, 1989; Yuan et al., 2016). We show that condition $\widetilde{U}(1) < 1$ is closely related to a simple bound on $\alpha_{\text{eff}}$ in terms of critical regularization $\lambda_{\text{crit}}$, which is defined only through problem's spectrum $\lambda_k$ and SE parameter $\tau$.

**Proposition 4.** *Retain assumptions of Prop. 3 and define $\lambda_{\text{crit}}$ as the unique positive solution to $\sum_k \frac{\lambda_k}{\tau\lambda_k + \lambda_{\text{crit}}} = 1$. Then, convergence condition $\widetilde{U}(1) < 1$ requires for $\alpha_{\text{eff}}$ to obey*

$$\alpha_{\text{eff}} < \frac{2}{\gamma\lambda_{\text{crit}}}. \tag{22}$$

*Moreover, for fixed $\beta, \gamma$ the convergence condition $\widetilde{U}(1) < 1$ is equivalent to the condition $\alpha_{\mathrm{eff}} < \alpha_{\mathrm{eff}}^{(c)}(\beta, \gamma)$, where the critical values $\alpha_{\mathrm{eff}}^{(c)}(\beta, \gamma)$ are upper bounded by $\frac{2}{\gamma \lambda_{\mathrm{crit}}}$ (by Eq. (22)) and*

$$\frac{2}{\gamma \lambda_{\mathrm{crit}}} / \alpha_{\mathrm{eff}}^{(c)}(\beta, \gamma) - 1 \leq \frac{\lambda_{\mathrm{max}}}{\lambda_{\mathrm{crit}}} \frac{1-\beta}{\gamma(1+\beta)} \overset{\beta \to 1}{=} O(1-\beta). \tag{23}$$

The bound (22) is much simpler to use than the exact condition $\widetilde{U}(1) < 1$, while (23) shows that (22) is tight for practically common case of $\beta$ close to 1. Both conditions $\widetilde{U}(1) < 1$ and (22) agree very well with experiments on MNIST and synthetic data (Fig. 2). Importantly, Eq. (22) implies that, in contrast to the non-stochastic ($\gamma = 0$) GD, effective learning rate in SGD must be *bounded* to prevent divergence. In non-strongly convex problems, accelerated convergence of non-stochastic GD is achieved by gradually adjusting $\beta$ to 1 at a fixed $\alpha$ during optimization (Nemirovskiy & Polyak, 1984). We see that this mechanism is not applicable to mini-batch SGD.

**Exponential divergence.** Our stability analysis above shows that loss diverges at large $t$ when $\widetilde{U}(1) > 1$. This divergence is primarily characterized by the convergence radius $r_L$ determined from the equation $r_L \widetilde{U}(r_L) = 1$ according to (16). Indeed, assuming an asymptotically exponential ansatz $L(t) \sim C e^{t/t_{\mathrm{div}}}$, we immediately get the divergence time scale $t_{\mathrm{div}} = -1/\log r_L$. A closer inspection of $\widetilde{L}(z)$ near $r_L$ also allows to determine the constant $C$ (see appendix E.4):

$$L(t) \sim \frac{\widetilde{V}(r_L)}{2(1 + r_L^2 \widetilde{U}'(r_L))} r_L^{-t}. \tag{24}$$

Note that $r_L$ can be very close to 1, which is naturally observed in *eventual divergent* phase by taking sufficiently small $\alpha$ and/or $\gamma$. In this case the characteristic divergence time $t_{\mathrm{div}}$ is large and we expect divergent behavior to occur only for $t \gtrsim t_{\mathrm{blowup}}$, while for $t \ll t_{\mathrm{blowup}}$ the loss converges roughly with the rate of noiseless model $\widetilde{U}(z) \equiv 0$. As $t_{\mathrm{blowup}}$ should have a meaning of a time moment where loss starts to significantly deviate from noiseless trajectory we define it by equating divergent (24) and noiseless losses. We confirm these effects experimentally in Fig, 3 (right).

## 6 SOLUTIONS FOR POWER-LAW SPECTRAL DISTRIBUTIONS

To get a more detailed picture of the loss evolution let us assume now that the eigenvalues $\lambda_k$ and the second moments $C_{kk,0}$ of initial ($t = 0$) approximation error are subject to large-$k$ power laws

$$\lambda_k = \Lambda k^{-\nu}(1 + o(1)), \quad \sum_{l \geq k} \lambda_l C_{ll,0} = K k^{-\varkappa}(1 + o(1)) \tag{25}$$

with some exponents $\nu, \varkappa > 0$ (also denote $\zeta = \varkappa/\nu$) and coefficients $\Lambda, K$. Such (or similar) power laws are empirically observed in many high-dimensional problems, can be derived theoretically, and are often assumed for theoretical optimization guarantees (see many references in App. A). If $\nu \leq \frac{1}{2}$, then $\sum_k \lambda_k^2 = \infty$ and we are in the "immediate divergence" regime mentioned earlier, and if $\frac{1}{2} < \nu \leq 1$, then $\sum_k \lambda_k = \infty$ and we are in "eventual divergence" regime. Away from these two divergent regimes, we precisely characterize the late time loss asymptotics:

**Theorem 2.** *Assume spectral conditions* (25) *with* $\nu > 1$ *and that parameters* $\alpha, \beta, \gamma$ *are as in Proposition 3 and such that convergence condition* $\widetilde{U}(1) < 1$ *is satisfied. Then*

$$\sum_{t=1}^{T} t L(t) = (1 + o(1)) \begin{cases} \frac{K\Gamma(\zeta+1)}{2(1-\widetilde{U}(1))(2-\zeta)}\left(\frac{2\alpha\Lambda}{1-\beta}\right)^{-\zeta} t^{2-\zeta}, & 2 - \zeta > 1/\nu, \\ \frac{\gamma \widetilde{V}(1)\Gamma(2-1/\nu)}{8(1-\widetilde{U}(1))^2}\left(\frac{2\alpha\Lambda}{1-\beta}\right)^{1/\nu} t^{1/\nu}, & 2 - \zeta < 1/\nu. \end{cases} \tag{26}$$

The idea of the proof is to observe that the generating function for the sequence $t L(t)$ is $\sum_{t=1}^{\infty} t L(t) z^t = z \frac{d}{dz} \widetilde{L}(z)$, where

$$\frac{d}{dz}\widetilde{L}(z) = \frac{\frac{d}{dz}\widetilde{V}(z)}{2(1 - z\widetilde{U}(z))} + \frac{\widetilde{V}(z)\frac{d}{dz}(z\widetilde{U}(z))}{2(1 - z\widetilde{U}(z))^2}. \tag{27}$$

We argue that $r_L = 1$ and apply the Tauberian theorem, relating the asymptotics of the cumulative sums $\sum_{t=1}^{T} t L(t)$ to the singularity of the generating function at $z = r_L = 1$. One can then show

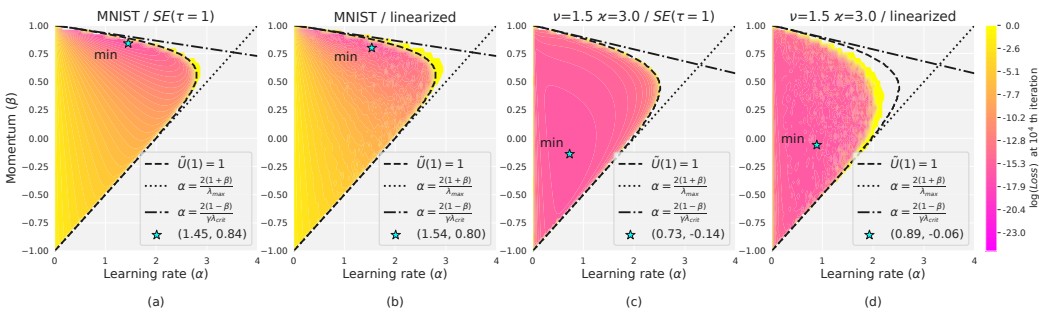

Figure 2: $L(t = 10^4)$ for different learning rates $\alpha$ and momenta $\beta$, in the $SE(\tau = 1)$ and linearized regimes, for MNIST and synthetic data given by Eq. (25) with $\nu = 1.5, \varkappa = 3$, for $b = 10$. Black lines show convergence boundaries from Proposition (3) and Eqs. (21) and (22). MNIST experiment is in the signal-dominated phase and synthetic in the noise-dominated. The stars show $\arg\min_{\alpha,\beta} L$.

using (18), (19) that $\widetilde{U}'(z)$ diverges as $\widetilde{U}'(1 - \varepsilon) \propto \varepsilon^{-\frac{1}{\nu}}$ and $\widetilde{V}'(z)$ diverges as $\widetilde{V}'(1 - \varepsilon) \propto \varepsilon^{\zeta - 2}$ for $\zeta < 2$. Depending on which of the two terms in Eq. (27) has a stronger singularity, we observe two qualitatively different phases (previously known in SGD without momenta (Varre et al., 2021)): the "signal-dominated" phase at $\zeta - 2 < -1/\nu$ and the "noise-dominated" phase at $\zeta - 2 > -1/\nu$. See Figure 3 (left) for the resulting full phase diagram.

In the sequel we assume that not only cumulative sums $\sum_1^T t L(t)$, but also individual terms $t L(t)$ obey power-law asymptotics. Then $L(t) \approx L_{\text{approx}}(t) = C t^{-\xi}$, where, by Eq. (26),

$$L_{\text{approx}}(t) = \begin{cases} C_{\text{signal}} t^{-\zeta} = \frac{K\Gamma(\zeta+1)}{2(1-\widetilde{U}(1))} \left(\frac{2\alpha\Lambda}{1-\beta}\right)^{-\zeta} t^{-\zeta}, & \zeta < 2 - 1/\nu, \quad (28\text{a}) \\ C_{\text{noise}} t^{1/\nu-2} = \frac{\gamma\widetilde{V}(1)\Gamma(2-1/\nu)}{8\nu(1-\widetilde{U}(1))^2} \left(\frac{2\alpha\Lambda}{1-\beta}\right)^{1/\nu} t^{1/\nu-2}, & \zeta > 2 - 1/\nu. \quad (28\text{b}) \end{cases}$$

Note that the asymptotics (28) depend on SGD hyperparameters $(\alpha, \beta, \gamma)$ only through the constants $C_{\text{signal}}(\alpha, \beta, \gamma)$ and $C_{\text{noise}}(\alpha, \beta, \gamma)$, which makes them essential for describing finer details of the loss trajectories in both phases. In the following sections we give a few examples of such analysis.

**Transition between phases.** Asymptotic (28) allows us to go further in understanding the "noise-dominated" phase $\zeta > 2 - 1/\nu$. Using representation (27) and linearity of Laplace transform, we write loss asymptotic as $L(t) \approx C_{\text{signal}} t^{-\zeta} + C_{\text{noise}} t^{-2+\frac{1}{\nu}}$ with two terms given by (28a),(28b). As $C_{\text{noise}}$ vanishes in the noiseless limit $\gamma \to 0$, we expect that for sufficiently small $\gamma$ the signal term here dominates the noise term up to time scale $t_{\text{trans}}$ given by

$$t_{\text{trans}} = \left(\frac{C_{\text{signal}}}{C_{\text{noise}}}\right)^{1/(\zeta-2+1/\nu)} = \left(\left(\frac{1-\beta}{2\alpha\Lambda}\right)^{1/\nu+\zeta} \frac{4\nu K\Gamma(\zeta+1)(1-\widetilde{U}(1))}{\gamma\Gamma(2-1/\nu)\widetilde{V}(1)}\right)^{1/(\zeta-2+1/\nu)}. \quad (29)$$

This conclusion is fully confirmed by our experiments with simulated data, see Fig. 3(center). Note that the time scale $t_{\text{trans}}$ can vary widely depending on $\alpha, \beta, \gamma$; in particular, $t_{\text{trans}}$ might not be reached at all in realistic optimization time if the noise $\gamma$ and effective learning rate $\frac{\alpha}{1-\beta}$ are small.

A similar reasoning can be used to estimate the time $t_{\text{blowup}}$ of transition from early convergence to subsequent divergence (see Fig. 3 (right)). In Sec. E.4 we consider the scenario with $\beta = 0, \gamma = \tau = 1$ and power laws (25) in which $\frac{1}{2} < \nu < 1$ (eventual divergence) and $\zeta < 1$. We show that at small $\alpha$ we have $t_{\text{blowup}} \approx a_* t_{\text{div}} \approx a_* \left(\frac{1}{4\nu}\Gamma(2 - 1/\nu)\Gamma(1/\nu - 1)\right)^{\nu/(\nu-1)} (2\alpha\Lambda)^{1/(\nu-1)}$, where $a_* = a_*(\nu, \zeta)$ is the solution of the equation $\frac{1/\nu-1}{\Gamma(1-\zeta)} a_*^{-\zeta} = e^{a_*}$.

**Hyperparameter optimization.** Observe that the optimal learning ratse $\alpha$ and momenta $\beta$ in Fig. 2 differ significantly between signal-dominated ((a), (b)) and noise-dominated ((c), (d)) phases. To explain this observation, we analyze late time loss asymptotic (28) at a fixed batch size $b$, and show that two phases indeed have distinct patterns of optimal $\alpha, \beta$.

*"Signal-dominated" phase.* First, note that near the divergence boundary $\widetilde{U}(1) = 1$ the loss is high due to denominator $1 - \widetilde{U}(1)$ in (28a). This suggests that at optimal $\alpha, \beta$ the difference $\widetilde{U}(1) - 1$ is of order 1, which allows us to neglect dependence of loss on $\widetilde{U}(1)$ as long as convergence condition $\widetilde{U}(1) < 1$ holds. Then, loss becomes $L(t) \sim (\alpha_{\text{eff}} t)^{-\zeta}$ and it is beneficial to increase $\alpha_{\text{eff}}$. However,

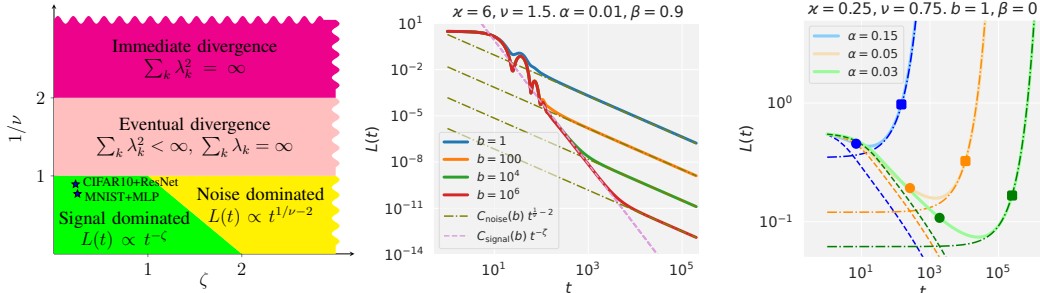

Figure 3: **Left**: Phase diagram for $L(t)$ under power-law spectral conditions (25). The MNIST and CIFAR10 stars are placed according to power-fits in Figs. 1 and 9 (top). **Center**: Transition from signal-dominated to noise-dominated regime during training. Asymptotic power-law lines are calculated from (28a) and (28b). **Right**: Divergent $L(t)$ in the phase $\frac{1}{2} < \nu < 1$: experimental trajectories (solid) with marked points at positions $t = t_{\text{div}}$ (squares) and $t = t_{\text{blowup}}$ (circles), theoretical late-time asymptotics (24) (dash-dotted) and noiseless early time trajectories (dashed).

Figure 4: Theoretical and experimental values of optimal momenta $\beta$ in the range $(\nu, \varkappa) \in [1,3] \times [0,6]$ of power-law models (25). The blue/red background shows the values $\Xi$ given by (30) and characterizing the theoretical sign of $\partial_\beta L_{\text{approx}}(t)|_{\beta=0,\alpha=\alpha_{\text{opt}}}$ at $\tau = \gamma = 1$ in the noisy regime. The four diamonds represent the pairs $(\nu, \varkappa)$ for which we experimentally find optimal $\beta_{\text{opt}}$. The diamond color shows the value of $\beta_{\text{opt}}$. In agreement with Prop. 5, experimental $\beta_{\text{opt}} > 0 (< 0)$ when $\Xi < 0 (> 0)$. The dotted yellow line separates the signal-dominated (below) and the noise-dominated (above) regimes.

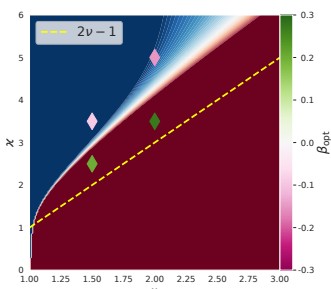

$\alpha_{\text{eff}}$ has an upper bound (23) which becomes tight for sufficiently small $1 - \beta$ (see (23)). Thus, in agreement with Fig. 2 (left), the optimal $\alpha, \beta$ correspond to small $1 - \beta$ and $\alpha_{\text{eff}} \approx \frac{2}{\gamma\lambda_{\text{crit}}}$.

*"Noise-dominated" phase.* Similarly to signal-dominated phase, we see from (28b) that $\widetilde{U}(1)$ can be neglected away from convergence boundary. Then, the loss is $L(t) \sim \widetilde{V}(1)\alpha_{\text{eff}}^{1/\nu} t^{1/\nu-2}$. In contrast to signal-dominated phase, $\alpha_{\text{eff}} = \alpha/(1-\beta)$ appears with the positive power suggesting that high momenta $\beta \approx 1$ are non-optimal. At the same time, it can be shown that $\widetilde{V}(1) \sim \alpha^{-1}$ at small $\alpha$, so it is neither favorable to use small $\alpha$. Thus, in accordance with Fig. 2 (right), the optimal $\alpha, \beta$ should be located in a general position inside the convergence region $\widetilde{U}(1) < 1$.

Additionally, for the problem of optimizing batch-size $b$ at a fixed computational budget $bt$, in Sec. E.5 we derive the previously observed linear scalability of SGD w.r.t. batch size (Ma et al., 2018).

**Positive vs negative momenta.** It is known that in some noisy settings SGD does not benefit from using momentum (Polyak, 1987; Kidambi et al., 2018; Paquette & Paquette, 2021). Empirically, we find that in the signal-dominated regime momentum helps significantly, while in the noise-dominated regime its effect is weaker, and it can even be preferable to set $\beta < 0$ (see Fig. 2). Our asymptotic formula (28) with explicit constants allows us to confirm this theoretically and, moreover, in the case $\tau = \gamma = 1$ give a simple explicit condition under which $\beta < 0$ are preferable. Specifically, we show that if we fix $\alpha$ at the optimal value $\alpha_{\text{opt}}$ at $\beta = 0$, then we can further decrease the loss by increasing or decreasing $\beta$, depending on the regime and a spectral characteristic $\Xi$ defined below:

**Proposition 5** (short version). *Assume asymptotic power laws* (25) *with some exponents $\nu, \varkappa > 0, \zeta = \varkappa/\nu$ and coefficients $\Lambda, K$. Let $\alpha_{\text{opt}} = \arg\min_\alpha L_{\text{approx}}(t)|_{\beta=0}$. Then: I. (signal-dominated case) Let $\zeta < 2 - 1/\nu$ and $0 < \tau \leq 1, 0 \leq \gamma \leq 1$, then $\partial_\beta L_{\text{approx}}(t)|_{\beta=0,\alpha=\alpha_{\text{opt}}} < 0$. II. (noise-dominated case) Let $\zeta > 2 - 1/\nu$ and $\tau = \gamma = 1$, then the sign of $\partial_\beta L_{\text{approx}}(t)|_{\beta=0,\alpha=\alpha_{\text{opt}}}$ equals the sign of the expression*

$$\Xi = \nu \operatorname{Tr}[\mathbf{H}] \operatorname{Tr}[\mathbf{H}\mathbf{C}_0] - (\nu - 1) \operatorname{Tr}[\mathbf{H}^2] \operatorname{Tr}[\mathbf{C}_0]. \tag{30}$$

In Fig. 4 we experimentally validate the $\Xi$ criterion for several pairs $(\nu, \varkappa)$. See details in Sec. E.6.

# 7 DISCUSSION

We can summarize our approach to the theoretical analysis of mini-batch SGD as consisting of 3 key stages: 1) Perform SE approximation; 2) Use it to solve the SGD dynamics and obtain initial spectral expressions (in our case Eqs. (18),(20) combined with (16)); 3) Apply these spectral expressions to study optimization stability, phase transitions, optimal parameters, etc. We have demonstrated this approach to produce various new and nontrivial theoretical results, which are also in good quantitative agreement with experiment on realistic problems such as MNIST and CIFAR10 learned by MLP/ResNet/MobileNet.

In particular, we have shown that, in contrast to full-batch GD, SGD has an effective learning rate limit determined by the spectral properties of the problem and batch size. Also, we have shown that in the "signal-dominated" phase (including, e.g., the MNIST and CIFAR10 tasks) momentum $\beta > 0$ is always beneficial, but in the "noise-dominated" phase a momentum $\beta < 0$ can be preferable.

One natural future research direction is to investigate which properties of data and features $\psi(\mathbf{x})$ are responsible for the accuracy of SE approximation, and how to systematically choose SE parameters $\tau_1, \tau_2$. Second, we were able to solve SE approximation only for the constant learning rate and momentum, and a more general solution might open up a way to a better understanding of accelerated SGD. Third, one can ask how the picture changes for spectral distributions other than power-law.

ACKNOWLEDGMENTS

We thank the anonymous reviewers for several useful comments that allowed us to improve the paper. Most numerical experiments were performed on the computational cluster Zhores (Zacharov et al., 2019). Research was supported by Russian Science Foundation, grant 21-11-00373.

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

## CONTENTS

## A  RELATED WORK

**SGD.**  Stochastic Gradient Descent (SGD) was introduced in Robbins & Monro (1951), and its various versions have been extensively studied since then, see e.g. Kiefer & Wolfowitz (1952); Rumelhart et al. (1986); Bottou & Bousquet (2007); Shamir & Zhang (2013); Jain et al. (2019); Harvey et al. (2019); Moulines & Bach (2011). The version (1) considered in the present paper is motivated by applications to training neural networks and is widely used (LeCun et al., 2015; Goodfellow et al., 2016; Sutskever et al., 2013). Its key feature is the presence of momentum, which is both theoretically known to improve convergence and widespread in practice (Polyak, 1964; 1987; Nesterov, 1983; Sutskever et al., 2013; Shallue et al., 2018).

**Mini-batch SGD and spectrally expressible representation.**  In the present paper we consider the *mini-batch* SGD, meaning that the stochasticity in SGD is specifically associated with the random choice of different batches used to approximate the full loss function at each iteration (not to be confused with SGD modeled as gradient descent with additive noise as e.g. in the gradient Langevin dynamics; see e.g. Wu et al. (2020); Zhu et al. (2018) for some discussion and comparison, as well as our Section G for an illustration in our case of quadratic problems).

We study this SGD in terms of how the second moments of the errors evolve with time. Our Proposition 1 generalizes existing evolution equations (Varre et al., 2021) to SGD with momentum. A serious difficulty in the study of this evolution is the presence of higher order mixing terms coupling different spectral components (the first term on the r.h.s. in our Eq. (5)). In Varre et al. (2021), it was shown that the effects of these terms can be controlled by suitable inequalities, allowing to establish upper bounds on error rates. In contrast, our SE approximation allows us to go beyond upper bounds and, for example, obtain explicit coefficients in the asymptotics of large-$t$ loss evolution without a Gaussian assumption (see Theorem 2).

Our approach is different and is close to the approach of recent paper Bordelon & Pehlevan (2021) where the authors considered a dynamic equation equivalent to our SE approximation with parameter $\tau = -1$ but without momentum. Apart from considering a wider family of dynamic equations, our major advancement compared to Bordelon & Pehlevan (2021) is that we manage to "solve" these dynamic equations using generating functions, and consequently use this solution to obtain a various results concerning stability conditions (Sec. 5) and various SGD characteristics under power-law spectral distributions (Sec. 6).

Related assumptions of Gaussian features have also been used in a recent series of works Goldt et al. (2020a); Loureiro et al. (2021); Goldt et al. (2020b) to analytically predict performance of linear models trained with full- or mini-batch SGD.

**Spectral power laws.**  In Section 6 we adopt very convenient *power law spectral assumptions* (25) on the asymptotic behavior of eigenvalues and target expansion coefficients. Such power laws are empirically observed in many high-dimensional problems (Cui et al., 2021; Bahri et al., 2021; Lee et al., 2020; Canatar et al., 2021; Kopitkov & Indelman, 2020; Dou & Liang, 2020; Atanasov et al., 2021; Bordelon & Pehlevan, 2021; Basri et al., 2020; Bietti, 2021), can be derived theoretically (Basri et al., 2020; Velikanov & Yarotsky, 2021), and closely related conditions are often assumed for theoretical optimization guarantees (Nemirovskiy & Polyak, 1984; Brakhage, 1987; Hanke, 1991; 1996; Gilyazov & Gol'dman, 2013; Varre et al., 2021; Caponnetto & De Vito, 2007; Steinwart et al., 2009; Berthier et al., 2020; Zou et al., 2021; Jin et al., 2021; Bowman & Montufar,

2022). An important aspect of our work is that, in contrast to most other works, we allow the exponent $\zeta$ characterizing initial second moments $C_{kk,0}$ to have values in the interval $(0,1)$. In this case, $\sum_k \lambda_k C_{kk} < \infty$ while $\sum_k C_{kk,0} = \infty$. This effectively means that the target function $y^*(\mathbf{x})$ has a finite euclidean norm, while the norm of the respective optimal weight vector $\mathbf{w}^*$ is infinite (i.e., the loss does not attain a minimum value). This broader assumption allows us, in particular, to correctly describe the loss evolution on MNIST for which $\zeta \approx 0.25$ (see Figures 1 and 3): without assuming $\|\mathbf{w}^*\| = \infty$, the SGD is proved to have error rate $O(t^{-\xi})$ with $\xi \geq 1$ (Varre et al., 2021), which clearly contradicts the experiment (see Figure 1(left)).

**Phase diagram.** It is known that performance of SGD can be decomposed into "bias" and "variance" terms, with the dominant term determining what we call "signal-dominated" or "noise-dominated" regime (Moulines & Bach, 2011; Varre et al., 2021; Varre & Flammarion, 2022). In particular, see Varre et al. (2021) for respective upper bounds for SGD without momentum. The main novelty of our paper compared to these previous results is the analytic framework allowing to obtain large-$t$ loss asymptotics with explicit constants for SGD with momentum (see Theorem 2) and use them to derive various quantitative conclusions related to model stability, phase transitions, and parameter optimization (see Section 6).

**Noisy GD with momentum.** GD with momentum (Heavy Ball) (Polyak, 1964) was extensively studied, but mostly in a noiseless setting or for generic (non-sampling) kinds of noise; see e.g. the monograph Polyak (1987). See Proakis (1974); Roy & Shynk (1990); Tugay & Tanik (1989) for a discussion of stability on quadratic problems. Polyak (1987) analyses various methods with noisy gradients for different kinds of noise and concludes that the fast convergence advantage of Heavy Ball and Conjugate Gradients (compared to GD) are not preserved under noise unless it decreases to 0 as the optimization trajectory approaches the solution. See also Kidambi et al. (2018); Paquette & Paquette (2021) for further comparisons of noisy GD with or without momentum. Our work refines this picture in the case of sampling noise. Our results in Section 6 suggest that including positive momentum in SGD always improves convergence in the signal-dominated phase, but generally not in the noise-dominated phase (where it may even be beneficial to use a negative momentum).

## B DETAILS OF THE SETTING

### B.1 BASICS

In our problem setting we always consider regression tasks of fitting target function $y^\star(\mathbf{x})$ using linear models $f(\mathbf{x}, \mathbf{w})$ of the form

$$f(\mathbf{x}, \mathbf{w}) = \langle \mathbf{w}, \boldsymbol{\psi}(\mathbf{x}) \rangle \tag{31}$$

where $\langle \cdot, \cdot \rangle$ stands for a scalar product and $\boldsymbol{\psi}(\mathbf{x})$ are some (presumably non-linear) features of inputs $\mathbf{x}$. The space $\mathcal{X}$ containing inputs $\mathbf{x}$ is not important in our context. Next, the model will be always trained to minimize a quadratic loss function over training dataset $\mathcal{D}$:

$$L(\mathbf{w}) = \frac{1}{2} \mathbb{E}_{\mathbf{x} \sim \mathcal{D}} (\langle \mathbf{w}, \boldsymbol{\psi}(\mathbf{x}) \rangle - y^\star(\mathbf{x}))^2. \tag{32}$$

Denote by $B_t$ a set of $b = |B_t|$ training samples $\mathbf{x}$ drawn independently and uniformly without replacement from $\mathcal{D}$. Then a single step of SGD with learning rate $\alpha_t$ and momentum parameter $\beta_t$ takes the form

$$\begin{pmatrix} \mathbf{w}_{t+1} \\ \mathbf{v}_{t+1} \end{pmatrix} = \begin{pmatrix} \mathbf{I} - \alpha_t \mathbf{H}(B_t) & \beta_t \mathbf{I} \\ -\alpha_t \mathbf{H}(B_t) & \beta_t \mathbf{I} \end{pmatrix} \begin{pmatrix} \mathbf{w}_t \\ \mathbf{v}_t \end{pmatrix} + \begin{pmatrix} \alpha_t \mathbf{h}(B_t) \\ \alpha_t \mathbf{h}(B_t) \end{pmatrix} \tag{33}$$

$$\mathbf{H}(B_t) \equiv \frac{1}{b} \sum_{\mathbf{x}_i \in B_t} \boldsymbol{\psi}(\mathbf{x}_i) \otimes \boldsymbol{\psi}(\mathbf{x}_i) \tag{34}$$

$$\mathbf{h}(B_t) \equiv \frac{1}{b} \sum_{\mathbf{x}_i \in B_t} \boldsymbol{\psi}(\mathbf{x}_i) y^\star(\mathbf{x}_i) \tag{35}$$

Note that expression (33) of SGD step is equivalent to more compact expression (2) when there exist optimal parameters $\mathbf{w}^\star$ such that $\langle \mathbf{w}^\star, \boldsymbol{\psi}(\mathbf{x}) \rangle = y^\star(\mathbf{x})$. As we have already noted, existence of

optimal parameters with $\|\mathbf{w}^\star\| < \infty$ may not be the case for models relevant to practical problems (e.g. MNIST). Thus we will consider (33) to be the basic form of SGD step in our work. However, the formal usage of (2) instead of (33) always leads to the same conclusions and therefore we focus on the case $\|\mathbf{w}^\star\| < \infty$ in the main paper. In subsection B.3 we consider formulation of SGD in the output space, which allows to treat cases $\|\mathbf{w}^\star\| = \infty$ and $\|\mathbf{w}^\star\| < \infty$ withing the same framework.

It is convenient to distinguish several cases of our problem setting differing by finite/infinite dimensionality $d$ of parameters $\mathbf{w}$, and finite/infinite size $|\mathcal{D}| = N$ of the dataset used for training. In our work we take into account all these cases, as each of them is used in either experimental or theoretical part of the paper.

## B.2 EIGENDECOMPOSITIONS AND OTHER DETAILS

**Case 1:** $d, N < \infty$. This is the simplest case and we will often repeat the respective paragraph of section 2. First, recall that matrix elements $d \times N$ Jacobian matrix $\boldsymbol{\Psi}$ are

$$\Psi_{ij} = \psi_i(\mathbf{x}_j), \ \mathbf{x}_j \in \mathcal{D} \tag{36}$$

We will assume without loss of generality that $\boldsymbol{\Psi}$ has a full rank $r = \min(d, N)$ (the general case reduces to this after a suitable projection). Then by (32) the Hessian of the model is

$$\mathbf{H} = \frac{1}{N} \boldsymbol{\Psi} \boldsymbol{\Psi}^T = \frac{1}{N} \sum_{j=1}^{N} \boldsymbol{\psi}(\mathbf{x}_j) \boldsymbol{\psi}^T(\mathbf{x}_j). \tag{37}$$

Another important matrix is the kernel $K(\mathbf{x}, \mathbf{x}') = \psi(\mathbf{x})^T \psi(\mathbf{x}')$ calculated on the training dataset $\mathcal{D}$

$$\mathbf{K} = \frac{1}{N} \boldsymbol{\Psi}^T \boldsymbol{\Psi} = \frac{1}{N} \left( \sum_{i=1}^{d} \psi_i(\mathbf{x}_j) \psi_i(\mathbf{x}_{j'}) \right)_{j,j'=1}^{N} \tag{38}$$

Next, consider the SVD decomposition of Jacobian $\boldsymbol{\Psi} = \sqrt{N} \mathbf{U} \boldsymbol{\Lambda} \mathbf{V}$ with $\mathbf{U} = (\mathbf{u}_k)_{k=1}^{d}$ being the matrix of left eigenvectors, $\mathbf{V} = (\boldsymbol{\phi}_k^T)_{k=1}^{N}$ being the matrix of right eigenvectors, and rectangular $d \times N$ diagonal matrix with singular values $\boldsymbol{\Lambda}_{kk} = \sqrt{\lambda_k}$, $k = 1 \ldots r$ on the diagonal. The Hessian and kernel matrices are symmetric and have eigendecompositions with shared spectrum of non-zero eigenvalues: $\lambda_k, \mathbf{u}_k$ for $\mathbf{H}$ and $\lambda_k, \mathbf{v}_k$ for $\mathbf{K}$.

Now let us proceed to the target function $y^\star(\mathbf{x})$. As in this paper we focus on the training error (32), we may also restrict ourselves to the target vector $\mathbf{y}^\star$ of target function values at training points: $y_i^\star = \frac{1}{\sqrt{N}} y^\star(\mathbf{x}_i)$, $\mathbf{x}_i \in \mathcal{D}$. The normalization by $\sqrt{N}$ is done here to allow for direct correspondence with the $N = \infty$ case. If $d \leq N$, we recall our assumption of $\boldsymbol{\Psi}$ having full rank and therefore the target function is reachable since the equation $\frac{1}{\sqrt{N}} \mathbf{w}^T \boldsymbol{\Psi} = \mathbf{y}^\star$ always has a solution. However, if $d < N$, there are many possible solutions of $\frac{1}{\sqrt{N}} \mathbf{w}^T \boldsymbol{\Psi} = \mathbf{y}^\star$, and we choose $\mathbf{w}^*$ to be the one with the same projection on $\ker \mathbf{H}$ as the initial model parameters $\mathbf{w}_0$. Later, this choice will guarantee that during SGD dynamic (33) parameter deviation always lies in the image of the Hessian: $\Delta \mathbf{w}_t \equiv \mathbf{w}_t - \mathbf{w}^\star \in \operatorname{Im} \mathbf{H}$. Finally, if $N > d$, the target may by unreachable: $\mathbf{y}^\star \notin \operatorname{Im} \boldsymbol{\Psi^T}$. In this case we consider the decomposition of the output space $\mathbb{R}^N = \operatorname{Im} \mathbf{K} \oplus \ker \mathbf{K}$ and the respective decomposition of target vector $\mathbf{y}^\star = \mathbf{y}_\parallel^\star + \mathbf{y}_\perp^\star$. Then $\mathbf{y}_\parallel^\star$ is always reachable $\mathbf{y}_\parallel^\star \in \operatorname{Im} \boldsymbol{\Psi^T}$ and in the rest we simply consider reachable part of the target $\mathbf{y}_\parallel^\star$ instead of the original target $\mathbf{y}^\star$.

Apart from general finite regression problems, the case $d, N < \infty$ described above applies to linearization of practical neural networks (and other parametric models). Specifically, consider a model $f(\mathbf{w}, \mathbf{x})$ initialized at $\mathbf{w}_0$ with initial predictions $f_0(\mathbf{x}) \equiv f(\mathbf{w}_0, \mathbf{x})$. The training of linearized model $f(\mathbf{w}, \mathbf{x}) = f_0(\mathbf{x}) + \langle \mathbf{w} - \mathbf{w}_0, \nabla_\mathbf{w} f(\mathbf{w}_0, \mathbf{x}) \rangle$ can be mapped to out basic problem setting (31) with replacements $\psi(\mathbf{x}) \longleftrightarrow \nabla_\mathbf{w} f(\mathbf{w}_0, \mathbf{x})$, $\mathbf{w} \longleftrightarrow \mathbf{w} - \mathbf{w}_0$, $y^\star(\mathbf{x}) \longleftrightarrow y^\star(\mathbf{x}) - f_0(\mathbf{x})$.

**Case 2:** $d = \infty, N < \infty$. In this case we assume that model parameters $\mathbf{w}$ and features $\psi(\mathbf{x})$ belong to a Hilbert space $\mathcal{H}$, and $\langle \cdot, \cdot \rangle$ denotes the scalar product in $\mathcal{H}$. For the Hessian we have

$$\mathbf{H} = \frac{1}{N} \sum_{j=1}^{N} \boldsymbol{\psi}(\mathbf{x}_j) \otimes \boldsymbol{\psi}(\mathbf{x}_j), \quad \mathbf{x}_j \in \mathcal{D} \tag{39}$$

We assume that the features $\boldsymbol{\psi}(\mathbf{x}_j)$, $\mathbf{x}_j \in \mathcal{D}$ are linearly independent (as is usually the case in practice), then $\dim \operatorname{Im} \mathbf{H} = N < \infty$. As SGD updates occur in $\operatorname{Im} \mathbf{H}$, we again use space decomposition $\mathcal{H} = \operatorname{Im} \mathbf{H} \oplus \ker \mathbf{H}$ and respective decomposition of parameters $\mathbf{w} = \mathbf{w}_{\parallel} + \mathbf{w}_{\perp}$ and features $\boldsymbol{\psi}(\mathbf{x}) = \boldsymbol{\psi}(\mathbf{x})_{\parallel} + \boldsymbol{\psi}(\mathbf{x})_{\perp}$. As $\boldsymbol{\psi}(\mathbf{x}_j)_{\perp} = 0, \mathbf{x}_j \in \mathcal{D}$ and $\mathbf{w}_{\perp}$ do not affect the loss (32) we can restrict ourselves to $\operatorname{Im} \mathbf{H}$ which bring us to finite dimensional case $d = N < \infty$ fully described above.

The case $d = \infty, N < \infty$ is primarily used for kernel regression problems defined by a kernel function $K(\mathbf{x}, \mathbf{x}')$ which can to be, for example, one of the "classical" kernels (e.g. Gaussian) or Neural Tangent Kernel (NTK) of some infinitely wide neural network. In this case $\mathcal{H}$ is a reproducing kernel Hilbert space (RKHS) of the kernel $K(\mathbf{x}, \mathbf{x}')$, and features $\boldsymbol{\psi}(\mathbf{x})$ are the respective mappings $\boldsymbol{\psi} : \mathcal{X} \to \mathcal{H}$ with the property $K(\mathbf{x}, \mathbf{x}') = \langle \boldsymbol{\psi}(\mathbf{x}), \boldsymbol{\psi}(\mathbf{x}') \rangle$.

**Case 3:** $d, N = \infty$. We are particularly interested in the case $d = N = \infty$ because in practice $N$ is quite large and, as previously discussed, the eigenvalues $\lambda_k$ and target expansion coefficients often are distributed according to asymptotic power laws, which all suggests working in an infinite-dimensional setting. A standard approach to rigorously accommodate $N = \infty$, which we sketch now, is based on Mercer's theorem.

We describe the infinite ($N = \infty$) training dataset by a probability measure $\mu(\mathbf{x})$ so that the loss takes the form

$$L(\mathbf{w}) = \frac{1}{2} \int (\langle \mathbf{w}, \boldsymbol{\psi}(\mathbf{x}) \rangle - y^{\star}(\mathbf{x}))^2 d\mu(\mathbf{x}) = \frac{1}{2} \|\langle \mathbf{w}, \boldsymbol{\psi}(\mathbf{x}) \rangle - y^{\star}(\mathbf{x})\|_{\mathcal{F}}^2, \tag{40}$$

assuming that $\|y^{\star}(\mathbf{x})\|_{\mathcal{F}}^2 < \infty$ for the loss to be well defined. Here in addition to the Hilbert space of parameters $\mathcal{H} \ni \mathbf{w}$ we introduced the Hilbert space $\mathcal{F}$ of square integrable functions $f(\mathbf{x}) : \|f\|_{\mathcal{F}}^2 \equiv \int f(\mathbf{x})^2 d\mu(\mathbf{x}) < \infty$. The Hessian $\mathbf{H}$, and the counterparts of the kernel and feature matrices $\mathcal{K}$ (38) and $\boldsymbol{\Psi}$ (36) are now operators

$$\mathbf{H} : \mathcal{H} \to \mathcal{H}, \quad \mathbf{H} = \int \boldsymbol{\psi}(\mathbf{x}) \otimes \boldsymbol{\psi}(\mathbf{x}) d\mu(\mathbf{x}) \tag{41}$$

$$\mathbf{K} : \mathcal{F} \to \mathcal{F}, \quad \mathbf{K}g(x) = \int K(\mathbf{x}, \mathbf{x}')g(\mathbf{x}') d\mu(\mathbf{x}') \tag{42}$$

$$\boldsymbol{\Psi} : \mathcal{F} \to \mathcal{H}, \quad \boldsymbol{\Psi}g(x) = \int \boldsymbol{\psi}(\mathbf{x}')g(\mathbf{x}') d\mu(\mathbf{x}') \tag{43}$$

Again, due to our interest in convergence of the loss (32) during SGD (33), we project parameters $\mathbf{w}$ to $\mathcal{H} \ominus \ker \mathbf{H}$ and target function $y^{\star}(\mathbf{x})$ to $\mathcal{F} \ominus \ker \mathbf{K}$. After this projection the target is reachable in the sense that $y^{\star}(\mathbf{x}) \in \mathcal{F} \ominus \ker \mathbf{K} = \overline{\operatorname{Im} \boldsymbol{\Psi}^{\dagger}}$. In particular, this means that an optimum such that $\|\langle \mathbf{w}^{\star}, \boldsymbol{\psi}(\mathbf{x}) \rangle - y^{\star}(\mathbf{x})\| = 0$ may not exist, but there is always a sequence $\mathbf{w}_n$ such that $\lim_{n \to \infty} \|\langle \mathbf{w}^{\star}, \boldsymbol{\psi}(\mathbf{x}) \rangle - y^{\star}(\mathbf{x})\| = 0$

Now we proceed to eigendecompositions of $\mathbf{H}$ and $\mathbf{K}$. According to the Mercer's theorem, the restriction of the operator $\mathbf{K}$ given by (42) to $\mathcal{F} \ominus \ker \mathbf{K}$ admits an eigendecomposition of the form

$$\mathbf{K}g(\mathbf{x}) = \sum_{k=1}^{\infty} \lambda_k \phi_k(\mathbf{x}) \int \phi_k(\mathbf{x}')g(\mathbf{x}') d\mu(\mathbf{x}') \tag{44}$$

with $\lambda_k > 0$, $\int \phi_k(\mathbf{x})\phi_l(\mathbf{x}) d\mu(\mathbf{x}) = \delta_{kl}$, and $\{\phi_k(\mathbf{x})\}$ being a complete basis in $\mathcal{F} \ominus \ker \mathbf{K}$. The latter allows to decompose features as

$$\boldsymbol{\psi}(\mathbf{x}) = \sum_k \rho_k \mathbf{u}_k \phi_k(\mathbf{x}) \tag{45}$$

with $\|\mathbf{u}_k\|^2 = 1$. Substituting (45) into (42) gives $\langle \mathbf{u_k}, \mathbf{u_l} \rangle = \delta_{kl}$ and $\rho_k^2 = \lambda_k$. Substituting then (45) into (41) gives

$$\mathbf{H} = \sum_{k=1}^{\infty} \lambda_k \mathbf{u}_k \otimes \mathbf{u}_k, \tag{46}$$

which also makes $\{\mathbf{u}_k\}$ an orthonormal basis in $\mathcal{H} \ominus \ker \mathbf{H}$.

Under mild regularity assumptions, one can show (König, 2013) that the eigenvalues $\lambda_k$ converge to 0, and we will assume this in the sequel. Note that if $\lambda_k \to 0$, then the range of the operator $\mathbf{\Psi}^\dagger$ is not closed so that, exactly as pointed out above, for some $y^*(\mathbf{x}) \in \mathcal{F} \ominus \ker \mathbf{K}$ there are no exact finite-norm minimizers $\mathbf{w}^* \in \mathcal{H}$ satisfying $\langle \mathbf{w}^\star, \boldsymbol{\psi}(\mathbf{x}) \rangle = y^\star(\mathbf{x})$ for $\mu$-almost all $\mathbf{x}$.

### B.3 OUTPUT AND PARAMETER SPACES

Recall that we only assume $\int y^\star(\mathbf{x})^2 d\mu(\mathbf{x}) < \infty$ but do not guarantee existence of the optimum $\mathbf{w}^\star$ in the case $N = d = \infty$. This means that considering SGD dynamics directly in the space of model outputs $f(\mathbf{x}) = \langle \mathbf{w}, \boldsymbol{\psi}(\mathbf{x}) \rangle$ may be advantageous when $\mathbf{w}^\star$ is not available. Moreover, it will allow to completely bypass construction of parameter space when original problem is formulated in terms of kernel $K(\mathbf{x}, \mathbf{x}')$ (e.g. for an infinitely wide neural network in the NTK regime). First, let us rewrite SGD recursion in the output space by taking the scalar product of (33) with $\boldsymbol{\psi}(\mathbf{x})$:

$$\begin{pmatrix} f_{t+1}(\mathbf{x}) \\ v_{t+1}(\mathbf{x}) \end{pmatrix} = \begin{pmatrix} \mathbf{I} - \alpha_t \mathbf{K}(B_t) & \beta_t \mathbf{I} \\ -\alpha_t \mathbf{K}(B_t) & \beta_t \mathbf{I} \end{pmatrix} \begin{pmatrix} f_t(\mathbf{x}) \\ v_t(\mathbf{x}) \end{pmatrix} + \begin{pmatrix} \alpha_t \mathbf{K}(B_t) y^\star(\mathbf{x}) \\ \alpha_t \mathbf{K}(B_t) y^\star(\mathbf{x}) \end{pmatrix} \tag{47}$$

$$\mathbf{K}(B_t) g(\mathbf{x}) = \frac{1}{b} \sum_{\mathbf{x}_i \in B_t} K(\mathbf{x}, \mathbf{x}_i) g(\mathbf{x}_i) \tag{48}$$

Here $v_t(\mathbf{x}) = \langle \mathbf{v}_t, \boldsymbol{\psi}(\mathbf{x}) \rangle = f_t(\mathbf{x}) - f_{t+1}(\mathbf{x})$. Considering the approximation error $\Delta f(\mathbf{x}) \equiv f(\mathbf{x}) - y^\star(\mathbf{x})$ we get

$$\begin{pmatrix} \Delta f_{t+1}(\mathbf{x}) \\ v_{t+1}(\mathbf{x}) \end{pmatrix} = \begin{pmatrix} \mathbf{I} - \alpha_t \mathbf{K}(B_t) & \beta_t \mathbf{I} \\ -\alpha_t \mathbf{K}(B_t) & \beta_t \mathbf{I} \end{pmatrix} \begin{pmatrix} \Delta f_t(\mathbf{x}) \\ v_t(\mathbf{x}) \end{pmatrix} \tag{49}$$

As (47) is equivalent to (33), we also get that (49) is equivalent to (2) when $\mathbf{w}^\star$ is available. Thus (49) can be considered as the most general and convenient form of writing an SGD iteration.

Similarly to second moments (3) defined for parameter space, we define second moments in the output space:

$$\begin{aligned} C^{\text{out}}(\mathbf{x}, \mathbf{x}') &\equiv \mathbb{E}[\Delta f(\mathbf{x}) \Delta f(\mathbf{x}')] \\ J^{\text{out}}(\mathbf{x}, \mathbf{x}') &\equiv \mathbb{E}[\Delta f(\mathbf{x}) v(\mathbf{x}')] \\ V^{\text{out}}(\mathbf{x}, \mathbf{x}') &\equiv \mathbb{E}[v(\mathbf{x}) v(\mathbf{x}')] \\ \mathbf{M}^{\text{out}}(\mathbf{x}, \mathbf{x}') &\equiv \begin{pmatrix} C^{\text{out}}(\mathbf{x}, \mathbf{x}') & J^{\text{out}}(\mathbf{x}, \mathbf{x}') \\ J^{\text{out}}(\mathbf{x}', \mathbf{x}) & V^{\text{out}}(\mathbf{x}, \mathbf{x}') \end{pmatrix}, \end{aligned} \tag{50}$$

where the superscript $^{\text{out}}$ indicates the output space (to avoid confusion with the respective moment counterparts defined in the parameter space). The same moments can be written in the eigenbasis $\{\phi(\mathbf{x})\}$ of the operator $\mathbf{K}$ according to the rule

$$G_{kl}^{\text{out}} = \int \phi_k(\mathbf{x}) G^{\text{out}}(\mathbf{x}, \mathbf{x}') \phi_l(\mathbf{x}) d\mu(\mathbf{x}), \tag{51}$$

where $G$ stands for either $C$, $F$ or $V$. When $\mathbf{w}^\star$ is available, the moments $C_{kl}^{\text{out}}$ in the output space are related to the moments $C_{kl}$ in the parameter space simply by

$$C_{kl}^{\text{out}} = \sqrt{\lambda_k \lambda_l} C_{kl}. \tag{52}$$

In particular,

$$C_{kk}^{\text{out}} = \lambda_k C_{kk} \tag{53}$$

and

$$\sum_k \lambda_k C_{kk} = \sum_k C_{kk}^{\text{out}} = \mathbb{E}[\|\Delta f\|^2] < \infty. \tag{54}$$

Even when $\mathbf{w}^\star$ is not available we can still define $C_{kl}$ by Eq. (52), but in this case $\sum_k C_{kk} = \infty$.

## C DYNAMICS OF SECOND MOMENTS

The purpose of this section is to prove proposition 1. However, as discussed in section B.3, the description in terms of moments in output space (50) is more widely applicable than description in terms of parameter moments (3). Thus we first prove an analogue of proposition 1 for output space moments. For convinience, let us denote the moments (50) as $\mathbf{C}^{\text{out}}, \mathbf{J}^{\text{out}}, \mathbf{V}^{\text{out}} \in \mathcal{F} \otimes \mathcal{F}$.

**Proposition 6.** *Consider SGD (49) with learning rates $\alpha_t$, momentum $\beta_t$, batch size $b$ and random uniform choice of the batch $B_t$. Then the update of second moments (50) is*

$$
\begin{pmatrix} \mathbf{C}_{t+1}^{out} & \mathbf{J}_{t+1}^{out} \\ (\mathbf{J}_{t+1}^{out})^\dagger & \mathbf{V}_{t+1}^{out} \end{pmatrix} = \begin{pmatrix} \mathbf{I} - \alpha_t \mathbf{K} & \beta_t \mathbf{I} \\ -\alpha_t \mathbf{K} & \beta_t \mathbf{I} \end{pmatrix} \begin{pmatrix} \mathbf{C}_t^{out} & \mathbf{J}_t^{out} \\ (\mathbf{J}_t^{out})^\dagger & \mathbf{V}_t^{out} \end{pmatrix} \begin{pmatrix} \mathbf{I} - \alpha_t \mathbf{K} & \beta_t \mathbf{I} \\ -\alpha_t \mathbf{K} & \beta_t \mathbf{I} \end{pmatrix}^T
$$
$$
+ \gamma \alpha_t^2 \begin{pmatrix} \boldsymbol{\Sigma}_t^{out} & \boldsymbol{\Sigma}_t^{out} \\ \boldsymbol{\Sigma}_t^{out} & \boldsymbol{\Sigma}_t^{out} \end{pmatrix}
\tag{55}
$$

*The second term represents covariance of function gradients induced mini-batch sampling noise and is given by*

*1) for finite dataset $\mathcal{D} = \{\mathbf{x}_i\}_{i=1}^N$:*

$$
\boldsymbol{\Sigma}_t^{out}(\mathbf{x}, \mathbf{x}') = \frac{1}{N} \sum_{\mathbf{x}_i \in \mathcal{D}} K(\mathbf{x}, \mathbf{x}_i) C_t^{out}(\mathbf{x}_i, \mathbf{x}_i) K(\mathbf{x}_i, \mathbf{x}') - \mathbf{K} \mathbf{C}_t^{out} \mathbf{K}(\mathbf{x}, \mathbf{x}'),
\tag{56}
$$

*and the amplitude $\gamma = \frac{N-b}{(N-1)b}$;*

*2) for infinite dataset $\mathcal{D}$ with density $d\mu(\mathbf{x})$:*

$$
\boldsymbol{\Sigma}_t^{out}(\mathbf{x}, \mathbf{x}') = \int K(\mathbf{x}, \mathbf{x}'') C_t^{out}(\mathbf{x}'', \mathbf{x}'') K(\mathbf{x}'', \mathbf{x}') d\mu(\mathbf{x}'') - \mathbf{K} \mathbf{C}_t^{out} \mathbf{K}(\mathbf{x}, \mathbf{x}'),
\tag{57}
$$

*and the amplitude $\gamma = \frac{1}{b}$.*

*Proof.* First, we take expression of single SGD (49) and use it to express $M_{t+1}^{out}$ through $M_t^{out}$.

$$
M_{t+1}^{out}(\mathbf{x}, \mathbf{x}') = \mathbb{E} \left[ \begin{pmatrix} \Delta f_{t+1}(\mathbf{x}) \Delta f_{t+1}(\mathbf{x}') & \Delta f_{t+1}(\mathbf{x}) v_{t+1}(\mathbf{x}') \\ v_{t+1}(\mathbf{x}) \Delta f_{t+1}(\mathbf{x}') & v_{t+1}(\mathbf{x}) v_{t+1}(\mathbf{x}') \end{pmatrix} \right]
$$
$$
= \mathbb{E} \left[ \begin{pmatrix} \mathbf{I} - \alpha_t \mathbf{K}(B_t) & \beta_t \mathbf{I} \\ -\alpha_t \mathbf{K}(B_t) & \beta_t \mathbf{I} \end{pmatrix} \begin{pmatrix} \Delta f_t(\mathbf{x}) \\ v_t(\mathbf{x}) \end{pmatrix} \begin{pmatrix} \Delta f_t(\mathbf{x}') & v_t(\mathbf{x}') \end{pmatrix} \begin{pmatrix} \mathbf{I} - \alpha_t \mathbf{K}(B_t) & \beta_t \mathbf{I} \\ -\alpha_t \mathbf{K}(B_t) & \beta_t \mathbf{I} \end{pmatrix}^T \right]
$$
$$
\overset{(1)}{=} \mathbb{E} \left[ \begin{pmatrix} \mathbf{I} - \alpha_t \mathbf{K}(B_t) & \beta_t \mathbf{I} \\ -\alpha_t \mathbf{K}(B_t) & \beta_t \mathbf{I} \end{pmatrix} \begin{pmatrix} \mathbf{C}_t^{out} & \mathbf{J}_t^{out} \\ (\mathbf{J}_t^{out})^\dagger & \mathbf{V}_t^{out} \end{pmatrix} \begin{pmatrix} \mathbf{I} - \alpha_t \mathbf{K}(B_t) & \beta_t \mathbf{I} \\ -\alpha_t \mathbf{K}(B_t) & \beta_t \mathbf{I} \end{pmatrix}^T \right]
$$
$$
\overset{(2)}{=} \begin{pmatrix} \mathbf{I} - \alpha_t \mathbf{K} & \beta_t \mathbf{I} \\ -\alpha_t \mathbf{K} & \beta_t \mathbf{I} \end{pmatrix} \begin{pmatrix} \mathbf{C}_t^{out} & \mathbf{J}_t^{out} \\ (\mathbf{J}_t^{out})^\dagger & \mathbf{V}_t^{out} \end{pmatrix} \begin{pmatrix} \mathbf{I} - \alpha_t \mathbf{K} & \beta_t \mathbf{I} \\ -\alpha_t \mathbf{K} & \beta_t \mathbf{I} \end{pmatrix}^T
$$
$$
+ \begin{pmatrix} \mathbb{E} \left[ \delta \mathbf{K}(B_t) \mathbf{C}^{out} \delta \mathbf{K}(B_t) \right] & \mathbb{E} \left[ \delta \mathbf{K}(B_t) \mathbf{C}^{out} \delta \mathbf{K}(B_t) \right] \\ \mathbb{E} \left[ \delta \mathbf{K}(B_t) \mathbf{C}^{out} \delta \mathbf{K}(B_t) \right] & \mathbb{E} \left[ \delta \mathbf{K}(B_t) \mathbf{C}^{out} \delta \mathbf{K}(B_t) \right] \end{pmatrix}
\tag{58}
$$

Here in (1) we used that $\Delta f_t, v_t$ are independent from $B_t$ and therefore the average factorizes into product of averages. In (2) we introduced notation $\delta \mathbf{K}(B_t) = \mathbf{K}(B_t) - \mathbf{K}$ and used that $\mathbb{E}[\delta \mathbf{K}(B_t)] = 0$.

Now we proceed to the calculation of the average from the last line in (58):

$$
\gamma \boldsymbol{\Sigma}_t^{out} \equiv \mathbb{E} \left[ \delta \mathbf{K}(B_t) \mathbf{C}_t^{out} \delta \mathbf{K}(B_t) \right]
\tag{59}
$$

*1)* Finite $\mathcal{D} = \{\mathbf{x}_i\}_{i=1}^N$.

Denoting for convenience function values calculated at $\mathbf{x}_i, \mathbf{x}_j$ with subscripts $_{ij}$, and omitting time index $t$, we get

$$
\begin{aligned}
\gamma\left(\mathbf{\Sigma}^{\text{out}}\right)_{ij} =& \mathbb{E}\left[\left(\delta\mathbf{K}(B)\mathbf{C}^{\text{out}}\delta\mathbf{K}(B)\right)_{ij}\right] \\
=& \mathbb{E}\left[\left(\frac{1}{b}\sum_{i'\in B}-\frac{1}{N}\sum_{i'}\right)\left(\frac{1}{b}\sum_{j'\in B}-\frac{1}{N}\sum_{j'}\right)K_{ii'}K_{jj'}C^{\text{out}}_{i'j'}\right] \\
=& \mathbb{E}\left[\left(\frac{1}{b^2}\sum_{i',j'\in B}-\frac{1}{N^2}\sum_{i',j'}\right)K_{ii'}K_{jj'}C^{\text{out}}_{i'j'}\right] \\
\overset{(1)}{=}& \left(\left(\frac{1}{b^2}\frac{b}{N}-\frac{1}{N^2}\right)\sum_{i'=j'}+\left(\frac{1}{|B|^2}\frac{b(b-1)}{N(N-1)}-\frac{1}{N^2}\right)\sum_{i'\neq j'}\right)K_{ii'}K_{jj'}C^{\text{out}}_{i'j'} \\
=& \left(\frac{N-b}{b}\sum_{i'=j'}-\frac{N-b}{b(N-1)}\sum_{i'\neq j'}\right)\frac{1}{N^2}K_{ii'}K_{jj'}C^{\text{out}}_{i'j'} \\
=& \frac{N-b}{(N-1)b}\sum_{i'j'}(N\delta_{i'j'}-1)\frac{1}{N^2}K_{ii'}K_{jj'}C^{\text{out}}_{i'j'}
\end{aligned}
\tag{60}
$$

Here unspecified sums run over dataset $\mathcal{D}$. In (1) we used that the fraction of batches $B$ which contain two indices $i'\neq j'$ is $\binom{N-2}{|B|-2}/\binom{N}{|B|}=\frac{|B|(|B-1|)}{N(N-1)}$ and the fraction of batches containing index $j'$ is $\binom{N-1}{|B|-1}/\binom{N}{|B|}=\frac{|B|}{N}$. Taking $\gamma=\frac{N-b}{(N-1)b}$ we get (56)

*2) Infinite $\mathcal{D}$ with density $d\mu(\mathbf{x})$.* Proceeding similarly to (60) we get

$$
\begin{aligned}
\mathbf{\Sigma}^{\text{out}}(\mathbf{x},\mathbf{x}') =& \mathbb{E}\left[\mathbf{K}(B)\mathbf{C}^{\text{out}}\mathbf{K}(B)\right](\mathbf{x},\mathbf{x}')-\mathbf{K}\mathbf{C}^{\text{out}}\mathbf{K}(\mathbf{x},\mathbf{x}') \\
=& \frac{b}{b^2}\int K(\mathbf{x},\mathbf{x}'')C^{\text{out}}(\mathbf{x}'',\mathbf{x}'')K(\mathbf{x}'',\mathbf{x}')d\mu(\mathbf{x}'') \\
& +\frac{b(b-1)}{b^2}\mathbf{K}\mathbf{C}^{\text{out}}\mathbf{K}(\mathbf{x},\mathbf{x}')-\mathbf{K}\mathbf{C}^{\text{out}}\mathbf{K}(\mathbf{x},\mathbf{x}') \\
=& \frac{1}{b}\left(\int K(\mathbf{x},\mathbf{x}'')C^{\text{out}}(\mathbf{x}'',\mathbf{x}'')K(\mathbf{x}'',\mathbf{x}')d\mu(\mathbf{x}'')-\mathbf{K}\mathbf{C}^{\text{out}}\mathbf{K}(\mathbf{x},\mathbf{x}')\right)
\end{aligned}
\tag{61}
$$

which gives (57) after setting $\gamma=\frac{1}{b}$. $\qquad\square$

Let us write second moments dynamics for both parameter space (4) and output space (55) in eigenbasises $\{\mathbf{u}_k\}$ and $\{\phi_k(\mathbf{x})\}$ using decompositions (45) and (44). For parameter space we get

$$
\begin{aligned}
\begin{pmatrix} C_{kl,t+1} & J_{kl,t+1} \\ J_{lk,t+1} & V_{kl,t+1} \end{pmatrix} =& \begin{pmatrix} 1-\alpha_t\lambda_k & \beta_t \\ -\alpha_t\lambda_k & \beta_t \end{pmatrix}\begin{pmatrix} C_{kl,t} & J_{kl,t} \\ J_{lk,t} & V_{kl,t} \end{pmatrix}\begin{pmatrix} 1-\alpha_t\lambda_l & \beta_t \\ -\alpha_t\lambda_l & \beta_t \end{pmatrix}^T \\
& +\gamma\alpha_t^2\sqrt{\lambda_k\lambda_l}\left(\sum_{k'l'}\sqrt{\lambda_{k'}\lambda_{l'}}C_{k'l',t}\int\phi_k(\mathbf{x})\phi_l(\mathbf{x})\phi_{k'}(\mathbf{x})\phi_{l'}(\mathbf{x})d\mu(\mathbf{x})-C_{kl,t}\right)\begin{pmatrix} 1 & 1 \\ 1 & 1 \end{pmatrix}
\end{aligned}
\tag{62}
$$

And for output space

$$
\begin{aligned}
\begin{pmatrix} C^{\text{out}}_{kl,t+1} & J^{\text{out}}_{kl,t+1} \\ J^{\text{out}}_{lk,t+1} & V^{\text{out}}_{kl,t+1} \end{pmatrix} =& \begin{pmatrix} 1-\alpha_t\lambda_k & \beta_t \\ -\alpha_t\lambda_k & \beta_t \end{pmatrix}\begin{pmatrix} C^{\text{out}}_{kl,t} & J^{\text{out}}_{kl,t} \\ J^{\text{out}}_{lk,t} & V^{\text{out}}_{kl,t} \end{pmatrix}\begin{pmatrix} 1-\alpha_t\lambda_l & \beta_t \\ -\alpha_t\lambda_l & \beta_t \end{pmatrix}^T \\
& +\gamma\alpha_t^2\lambda_k\lambda_l\left(\sum_{k'l'}C^{\text{out}}_{k'l',t}\int\phi_k(\mathbf{x})\phi_l(\mathbf{x})\phi_{k'}(\mathbf{x})\phi_{l'}(\mathbf{x})d\mu(\mathbf{x})-C^{\text{out}}_{kl,t}\right)\begin{pmatrix} 1 & 1 \\ 1 & 1 \end{pmatrix}
\end{aligned}
\tag{63}
$$

Now we are ready to prove proposition 1

**Proposition 1.** *Consider SGD* (1) *with learning rates* $\alpha_t$, *momentum* $\beta_t$, *batch size* $b$ *and random uniform choice of the batch* $B_t$. *Then the update of second moments* (3) *is*

$$\begin{pmatrix} \mathbf{C}_{t+1} & \mathbf{J}_{t+1} \\ \mathbf{J}_{t+1}^\dagger & \mathbf{V}_{t+1} \end{pmatrix} = \begin{pmatrix} \mathbf{I} - \alpha_t\mathbf{H} & \beta_t\mathbf{I} \\ -\alpha_t\mathbf{H} & \beta_t\mathbf{I} \end{pmatrix} \begin{pmatrix} \mathbf{C}_t & \mathbf{J}_t \\ \mathbf{J}_t^\dagger & \mathbf{V}_t \end{pmatrix} \begin{pmatrix} \mathbf{I} - \alpha_t\mathbf{H} & \beta_t\mathbf{I} \\ -\alpha_t\mathbf{H} & \beta_t\mathbf{I} \end{pmatrix}^T + \gamma\alpha_t^2 \begin{pmatrix} \mathbf{\Sigma}(\mathbf{C}_t) & \mathbf{\Sigma}(\mathbf{C}_t) \\ \mathbf{\Sigma}(\mathbf{C}_t) & \mathbf{\Sigma}(\mathbf{C}_t) \end{pmatrix}$$

$$(4)$$

*Here* $\gamma\mathbf{\Sigma}(\mathbf{C}_t)$ *is the covariance of gradient noise due to mini-batch sampling, with* $\gamma = \frac{N-b}{(N-1)b}$ *and*

$$\mathbf{\Sigma}(\mathbf{C}) = \frac{1}{N}\sum_{i=1}^N \langle\boldsymbol{\psi}(\mathbf{x}_i), \mathbf{C}\boldsymbol{\psi}(\mathbf{x}_i)\rangle\boldsymbol{\psi}(\mathbf{x}_i)\otimes\boldsymbol{\psi}(\mathbf{x}_i) - \mathbf{HCH}. \tag{5}$$

*Proof.* Note that in the eigenbasis $\{\phi_k(\mathbf{x})\}$ of $\mathbf{K}$ and eigenbasis $\{\mathbf{u}_k\}$ of $\mathbf{H}$ output and parameter second moments are connected as

$$\sqrt{\lambda_k\lambda_l}M_{kl} = M_{kl}^{\text{out}} \tag{64}$$

Using this connection rule we see that parameter dynamics (62) in eigenbasis of $\mathbf{H}$ is equivalent to output space second moments dynamics (63) in the eigenbasis of $\mathbf{K}$. Finally, (63) is equivalent to (55) proved in 6. □

For completeness, let us also write a formula of SE approximation in output space:

$$\begin{pmatrix} C_{kk,n+1}^{\text{out}} & J_{kk,n+1}^{\text{out}} \\ J_{kk,n+1}^{\text{out}} & V_{kk,n+1}^{\text{out}} \end{pmatrix} = \begin{pmatrix} 1 - \alpha_t\lambda_k & \beta_t \\ -\alpha_t\lambda_k & \beta_t \end{pmatrix} \begin{pmatrix} C_{kk,t}^{\text{out}} & J_{kk,t}^{\text{out}} \\ J_{kk,t}^{\text{out}} & V_{kk,t}^{\text{out}} \end{pmatrix} \begin{pmatrix} 1 - \alpha_t\lambda_k & \beta_t \\ -\alpha_t\lambda_k & \beta_t \end{pmatrix}^T$$

$$+ \gamma(\alpha_t\lambda_k)^2\left(\tau_1\sum_l C_{ll,t}^{\text{out}} - \tau_2 C_{kk,t}\right)\begin{pmatrix} 1 & 1 \\ 1 & 1 \end{pmatrix}. \tag{65}$$

# D  SE APPROXIMATION

## D.1  PRESENCE OF NON-SPECTRAL DETAILS IN THE GENERAL CASE.

In this section we show that for a mini-batch SGD noise term $\mathbf{\Sigma}$ given by (5) one cannot exactly describe respective noise spectral components $\Sigma_{kk} = \langle\mathbf{u}_k, \mathbf{\Sigma}\mathbf{u}_k\rangle$ in terms of only spectral distributions: eigenvalues $\lambda_k$ and second moments components $C_{kk}$. In turn, for a SGD dynamics (4) this would imply that spectral distributions are not sufficient to reconstruct $\rho_{\mathbf{C}_{t+1}}$ from $\rho_{\mathbf{C}_t}$. The following proposition characterizes non-spectral variability of $\Sigma_{kk}$ in a stronger sense: when not only $\lambda_k, C_{kk}$ are fixed, but the full Hessian $\mathbf{H}$ and second moments $\mathbf{C}$.

**Proposition 7.** *Consider a problem with fixed Hessian* $\mathbf{H}$ *and second moments* $\mathbf{C}$, *and a finite dataset size* $N < \infty$. *Then, all noise spectral component* $\Sigma_{kk}$ *are bounded as*

$$\Sigma_{kk} \le (N-1)\lambda_k \operatorname{Tr}[\mathbf{HC}] \tag{66}$$

*Next, any chosen component* $\Sigma_{kk}$ *can take any value in the interval* $[A, B]$ *(or* $[B, A]$ *if* $B < A$*) where endpoints* $A, B$ *are given by*

$$A = \lambda_k(\operatorname{Tr}[\mathbf{HC}] - \lambda_k C_{kk}), \quad B = (N-1)\lambda_k^2 C_{kk}. \tag{67}$$

Let us first discuss this result. Intuitively, the value $A$ corresponds to the case where the angle between different $\boldsymbol{\psi}(\mathbf{x}_i)$ and $\boldsymbol{\psi}(\mathbf{x}_j)$ is small, and therefore different rank-one terms in $\mathbf{H} = \frac{1}{N}\sum_{i=1}^n \boldsymbol{\psi}(\mathbf{x}_i)\otimes\boldsymbol{\psi}(\mathbf{x}_i)$ greatly overlap with each other. The value $B$ corresponds to the opposite case where all $\boldsymbol{\psi}(\mathbf{x}_i)$ are orthogonal.

Now we characterize the magnitude of non-spectral variations. For this, from SE approximation (7) we can take $\lambda_k \operatorname{Tr}[\mathbf{HC}]$ as a natural scale of $\Sigma_{kk}$. Then, bound (66) shows that relative to this natural scale, the magnitude of non-spectral variations of $\Sigma_{kk}$ is bounded by $N$. The values $A, B$ in (67) actually show that this magnitude is approximately achieved for a typical scenario with large dataset size $N$ and the most contributing term to the trace $\operatorname{Tr}[\mathbf{HC}]$ having the same order as the full trace: $\lambda_{k_*}C_{k_*k_*} \sim \operatorname{Tr}[\mathbf{HC}]$. Indeed, in this case $B \sim N\lambda_k^2 C_{kk} \sim N\lambda_k \operatorname{Tr}[\mathbf{HC}]$, $A \lesssim \lambda_k \operatorname{Tr}[\mathbf{HC}]$ and the ratio $B/A \gtrsim N$.

*Proof of Proposition 7.* Recall SVD decomposition (see section B) of the Jacobian $\boldsymbol{\Psi} = \sqrt{N} \mathbf{U} \boldsymbol{\Lambda} \mathbf{V}$ with $\mathbf{U} = (\mathbf{u}_k)_{k=1}^d$ being the matrix of left eigenvectors, $\mathbf{V} = (\boldsymbol{\phi}_k^T)_{k=1}^N$. Also, in the proof $k$ is always fixed and denote an index for each we want to analyze $\Sigma_{kk}$. Then, spectral component $\Sigma_{kk}$ can be written as

$$\Sigma_{kk} = \lambda_k \left[ N \sum_{i=1}^N \phi_{k,i}^2 \sum_{k_1,k_2} \phi_{k_1,i} \phi_{k_2,i} \sqrt{\lambda_{k_1} \lambda_{k_2}} C_{k_1 k_2} \right] - \lambda_k^2 C_{kk} \tag{68}$$

From representation (68) we observe that $\mathrm{Tr}[\mathbf{H}^{-1} \boldsymbol{\Sigma}] = (N-1) \mathrm{Tr}[\mathbf{H} \mathbf{C}]$ by using orthogonality of both columns and rows of $\mathbf{V} = (\boldsymbol{\phi}_k^T)_{k=1}^N$. Then, since $\boldsymbol{\Sigma}$ is positive semi-definite, we have $\lambda_k^{-1} \Sigma_{kk} < \mathrm{Tr}[\mathbf{H}^{-1} \boldsymbol{\Sigma}]$ and bound (66) follows.

The key idea of the proof for the second part of the proposition is that after fixing $\mathbf{H}$ and $\mathbf{C}$, we fixed $\mathbf{U}$ and $\boldsymbol{\Lambda}$ but $\mathbf{V}$ can be an arbitrary orthogonal matrix. Now we construct two specific examples of $\mathbf{V}$ so that $\Sigma_{kk}$ attains values $A$ and $B$ given by (67).

To get the $A$ value, for all $i$ take $\phi_{k,i} = \frac{1}{\sqrt{N}}$. Substituting this into (68) and using orthogonality $\sum_i \phi_{k_1,i} \phi_{k_2,i} = \delta_{k_1 k_2}$ gives

$$A = \lambda_k \left[ \sum_{k_1 k_2} \delta_{k_1 k_2} \sqrt{\lambda_{k_1} \lambda_{k_2}} C_{k_1 k_2} \right] - \lambda_k^2 C_{kk} = \lambda_k (\mathrm{Tr}[\mathbf{H} \mathbf{C}] - \lambda_k C_{kk}) \tag{69}$$

To get the $B$ value, we set $\mathbf{V}$ to the identity matrix. Then we have

$$B = \lambda_k \left[ N \sum_{i=1}^N \delta_{ki} \sum_{k_1,k_2} \delta_{k_1 i} \delta_{k_2 i} \sqrt{\lambda_{k_1} \lambda_{k_2}} C_{k_1 k_2} \right] - \lambda_k^2 C_{kk} = N \lambda_k^2 C_{kk} - \lambda_k^2 C_{kk} \tag{70}$$

Finally, we demonstrate that $\Sigma_{kk}$ can take any value between $A$ and $B$ by showing that two examples can be continuously deformed from one to another. Indeed, in the first example we specified only single row of $\mathbf{V}$, and the rest of the matrix can be chosen so that $\det \mathbf{V} = 1$. As the special orthogonal group $SO(N)$ is path connected, the required continuous deformation is guaranteed to exist. $\square$

## D.2 SE APPROXIMATION FOR UNCORRELATED LOSSES AND FEATURES

Recall expression (68) for the noise-spectral component $\Sigma_{kk}$. From SVD decomposition of Jacobian $\boldsymbol{\Psi} = \sqrt{N} \mathbf{U} \boldsymbol{\Lambda} \mathbf{V}$ we get $\psi_k^2(\mathbf{x}_i) \equiv \langle \mathbf{u}_k, \boldsymbol{\psi}(\mathbf{x}_i) \rangle^2 = N \lambda_k \phi_{k,i}^2$. Using again SVD of the Jacobian and definition $C_{k_1 k_2} = \langle \mathbf{u}_{k_1}, \mathbf{C} \mathbf{u}_{k_2} \rangle$, we get

$$\sum_{k_1,k_2} \phi_{k_1,i} \phi_{k_2,i} \sqrt{\lambda_{k_1} \lambda_{k_2}} C_{k_1 k_2} = \langle \boldsymbol{\psi}(\mathbf{x}_i), \mathbf{C} \boldsymbol{\psi}(\mathbf{x}_i) \rangle = \mathbb{E} \left[ |(\mathbf{w} - \mathbf{w}^*)^T \boldsymbol{\psi}(\mathbf{x}_i)|^2 \right]$$
$$= \mathbb{E} \left[ |f(\mathbf{w}, \mathbf{x}_i) - f^*(\mathbf{x}_i)|^2 \right] \equiv 2L(\mathbf{x}_i) \tag{71}$$

Now we can rewrite (68) as

$$\Sigma_{kk} = \frac{1}{N} \sum_{i=1}^N \psi_k^2(\mathbf{x}_i) 2L(\mathbf{x}_i) - \lambda_k^2 C_{kk} = \mathbb{E}_{\mathbf{x} \sim \mathcal{D}} \left[ \psi_k^2(\mathbf{x}) 2L(\mathbf{x}) \right] - \lambda_k^2 C_{kk} \tag{72}$$

If $L(\mathbf{x})$ and $\psi_k^2(\mathbf{x})$ are statistically independent w.r.t. $\mathbf{x} \sim \mathcal{D}$, the expectation factorizes into the product of expectations $\mathbb{E}_{\mathbf{x} \sim \mathcal{D}} [\psi_k^2(\mathbf{x})] = \lambda_k$ and $\mathbb{E}_{\mathbf{x} \sim \mathcal{D}} [2L(\mathbf{x})] = \mathrm{Tr}[\mathbf{H} \mathbf{C}]$. In this case

$$\Sigma_{kk} = \mathbb{E}_{\mathbf{x} \sim \mathcal{D}} \left[ \psi_k^2(\mathbf{x}) 2L(\mathbf{x}) \right] - \lambda_k^2 C_{kk} = \lambda_k \mathrm{Tr}[\mathbf{H} \mathbf{C}] - \lambda_k^2 C_{kk} \tag{73}$$

which is exactly SE approximation with $\tau_1 = \tau_2 = 1$.

### D.3 DYNAMICS OF SPECTRAL MEASURES

In Sec. 3 we mention that SE approximation allows to reconstruct the trajectories of observables $\text{Tr}[\phi(\mathbf{H})\mathbf{C}_t]$ from initial state $\mathbf{C}_0$. The way to do it is, of course, through considering dynamics of spectral measures, which we explicitly write below. Denote, for simplicity, $\rho_{\mathbf{J}_t} \equiv \rho_{\frac{1}{2}(\mathbf{J}_t + \mathbf{J}_t^\dagger)}$.

**Proposition 8.** *Assume SE approximation* (7) *holds for all* $\mathbf{C}_t$ *during SGD dynamics* (4). *Then spectral measures of all second moment matrices* $\rho_{\mathbf{C}_t}, \rho_{\mathbf{J}_t}, \rho_{\mathbf{V}_t}$ *can be written as*

$$\begin{pmatrix} \rho_{\mathbf{C}_t}(d\lambda) \\ \rho_{\mathbf{J}_t}(d\lambda) \\ \rho_{\mathbf{V}_t}(d\lambda) \end{pmatrix} = \mathbf{G}_t(\lambda) \begin{pmatrix} \rho_{\mathbf{C}_0}(d\lambda) \\ \rho_{\mathbf{J}_0}(d\lambda) \\ \rho_{\mathbf{V}_0}(d\lambda) \end{pmatrix} + \mathbf{s}_t(\lambda)\rho_{\mathbf{H}}(d\lambda) \tag{74}$$

*where* $\mathbf{G}_t(\lambda)$ *is* $\lambda$ *dependent* $3 \times 3$ *matrix, and* $\mathbf{s}_t(\lambda)$ *is* $\lambda$ *dependent 3-dimensional vector.*

*During SGD iterations* $\mathbf{G}_t(\lambda)$ *and* $\mathbf{s}_t(\lambda)$ *are transformed as*

$$\mathbf{G}_{t+1}(\lambda) = \begin{pmatrix} (1 - \alpha_t\lambda)^2 - \gamma\tau_2(\alpha_t\lambda)^2 & 2\beta_t(1 - \alpha_t\lambda) & \beta_t^2 \\ -\alpha_t\lambda(1 - \alpha_t\lambda) - \gamma\tau_2(\alpha_t\lambda)^2 & 1 - 2\beta_t\alpha_t\lambda & \beta_t^2 \\ (\alpha_t\lambda)^2 - \gamma\tau_2(\alpha_t\lambda)^2 & -2\beta_t\alpha_t\lambda & \beta_t^2 \end{pmatrix} \mathbf{G}_t(\lambda) \tag{75}$$

$$\begin{aligned} \mathbf{s}_{t+1}(\lambda) = {}& \begin{pmatrix} (1 - \alpha_t\lambda)^2 - \gamma\tau_2(\alpha_t\lambda)^2 & 2\beta_t(1 - \alpha_t\lambda) & \beta_t^2 \\ -\alpha_t\lambda(1 - \alpha_t\lambda) - \gamma\tau_2(\alpha_t\lambda)^2 & 1 - 2\beta_t\alpha_t\lambda & \beta_t^2 \\ (\alpha_t\lambda)^2 - \gamma\tau_2(\alpha_t\lambda)^2 & -2\beta_t\alpha_t\lambda & \beta_t^2 \end{pmatrix} \mathbf{s}_t(\lambda) \\ &+ \gamma\tau_1\alpha_t^2 \int \lambda' s_{t,1}(\lambda)\rho_{\mathbf{H}}(d\lambda) \begin{pmatrix} 1 \\ 1 \\ 1 \end{pmatrix} \\ &+ \gamma\tau_1\alpha_t^2 \int \lambda' \Big[ G_{t,11}(\lambda)\rho_{\mathbf{C}_0}(d\lambda') + G_{t,12}(\lambda)\rho_{\mathbf{J}_0}(d\lambda')G_{t,13}(\lambda)\rho_{\mathbf{V}_0}(d\lambda') \Big] \begin{pmatrix} 1 \\ 1 \\ 1 \end{pmatrix} \end{aligned} \tag{76}$$

*Proof.* Let us rewrite dynamics of second moments (4) as linear transformation of on $(\mathbf{C}_t, \mathbf{J}_t, \mathbf{V}_t)$ and then take traces $\text{Tr}[\phi(\mathbf{H})\#]$ of both sides with arbitrary test function $\phi(\lambda)$. Since arbitrariness of $\phi(\lambda)$ in the obtained relation implies the same relation but for spectral measures instead of traces

$$\begin{pmatrix} \rho_{\mathbf{C}_{t+1}}(d\lambda) \\ \rho_{\mathbf{J}_{t+1}}(d\lambda) \\ \rho_{\mathbf{V}_{t+1}}(d\lambda) \end{pmatrix} = \begin{pmatrix} (1 - \alpha_t\lambda)^2 & 2\beta_t(1 - \alpha_t\lambda) & \beta_t^2 \\ -\alpha_t\lambda(1 - \alpha_t\lambda) & 1 - 2\beta_t\alpha_t\lambda & \beta_t^2 \\ (\alpha_t\lambda)^2 & -2\beta_t\alpha_t\lambda & \beta_t^2 \end{pmatrix} \begin{pmatrix} \rho_{\mathbf{C}_t}(d\lambda) \\ \rho_{\mathbf{J}_t}(d\lambda) \\ \rho_{\mathbf{V}_t}(d\lambda) \end{pmatrix} + \gamma\alpha_t^2 \begin{pmatrix} \rho_{\mathbf{\Sigma}_t} \\ \rho_{\mathbf{\Sigma}_t} \\ \rho_{\mathbf{\Sigma}_t} \end{pmatrix} \tag{77}$$

Now we use SE approximation (7) to express $\rho_{\mathbf{\Sigma}_t}$ through $\rho_{\mathbf{C}_t}$ and $\rho_{\mathbf{H}}$. The result is

$$\begin{aligned} \begin{pmatrix} \rho_{\mathbf{C}_{t+1}}(d\lambda) \\ \rho_{\mathbf{J}_{t+1}}(d\lambda) \\ \rho_{\mathbf{V}_{t+1}}(d\lambda) \end{pmatrix} = {}& \begin{pmatrix} (1 - \alpha_t\lambda)^2 - \gamma\tau_2(\alpha_t\lambda)^2 & 2\beta_t(1 - \alpha_t\lambda) & \beta_t^2 \\ -\alpha_t\lambda(1 - \alpha_t\lambda) - \gamma\tau_2(\alpha_t\lambda)^2 & 1 - 2\beta_t\alpha_t\lambda & \beta_t^2 \\ (\alpha_t\lambda)^2 - \gamma\tau_2(\alpha_t\lambda)^2 & -2\beta_t\alpha_t\lambda & \beta_t^2 \end{pmatrix} \begin{pmatrix} \rho_{\mathbf{C}_t}(d\lambda) \\ \rho_{\mathbf{J}_t}(d\lambda) \\ \rho_{\mathbf{V}_t}(d\lambda) \end{pmatrix} \\ &+ \gamma\tau_1\alpha_t^2 \int \lambda'\rho_{\mathbf{C}_t}(d\lambda') \begin{pmatrix} \rho_{\mathbf{H}}(d\lambda) \\ \rho_{\mathbf{H}}(d\lambda) \\ \rho_{\mathbf{H}}(d\lambda) \end{pmatrix} \end{aligned} \tag{78}$$

Now we use obtained form of recursion of spectral measures prove representation (74) and transition formulas (75),(76). We proceed by induction. At $t = 0$ representation (74) indeed holds with $\mathbf{G}_0(\lambda) = \mathbf{I}$ and $\mathbf{s}_0(\lambda) = 0$. Now assume that (74) holds for iteration $t$ and substitute it into (78).

The result is

$$
\begin{aligned}
\begin{pmatrix} \rho_{\mathbf{C}_{t+1}}(d\lambda) \\ \rho_{\mathbf{J}_{t+1}}(d\lambda) \\ \rho_{\mathbf{V}_{t+1}}(d\lambda) \end{pmatrix} &= \begin{pmatrix} (1-\alpha_t\lambda)^2 - \gamma\tau_2(\alpha_t\lambda)^2 & 2\beta_t(1-\alpha_t\lambda) & \beta_t^2 \\ -\alpha_t\lambda(1-\alpha_t\lambda) - \gamma\tau_2(\alpha_t\lambda)^2 & 1-2\beta_t\alpha_t\lambda & \beta_t^2 \\ (\alpha_t\lambda)^2 - \gamma\tau_2(\alpha_t\lambda)^2 & -2\beta_t\alpha_t\lambda & \beta_t^2 \end{pmatrix} \mathbf{G}_t(\lambda) \begin{pmatrix} \rho_{\mathbf{C}_0}(d\lambda) \\ \rho_{\mathbf{J}_0}(d\lambda) \\ \rho_{\mathbf{V}_0}(d\lambda) \end{pmatrix} \\
&+ \begin{pmatrix} (1-\alpha_t\lambda)^2 - \gamma\tau_2(\alpha_t\lambda)^2 & 2\beta_t(1-\alpha_t\lambda) & \beta_t^2 \\ -\alpha_t\lambda(1-\alpha_t\lambda) - \gamma\tau_2(\alpha_t\lambda)^2 & 1-2\beta_t\alpha_t\lambda & \beta_t^2 \\ (\alpha_t\lambda)^2 - \gamma\tau_2(\alpha_t\lambda)^2 & -2\beta_t\alpha_t\lambda & \beta_t^2 \end{pmatrix} \mathbf{s}_t(\lambda)\rho_{\mathbf{H}}(d\lambda) \\
&+ \gamma\tau_1\alpha_t^2 \int \lambda' \Big[ G_{t,11}(\lambda)\rho_{\mathbf{C}_0}(d\lambda') + G_{t,12}(\lambda)\rho_{\mathbf{J}_0}(d\lambda')G_{t,13}(\lambda)\rho_{\mathbf{V}_0}(d\lambda') \Big] \begin{pmatrix} \rho_{\mathbf{H}}(d\lambda) \\ \rho_{\mathbf{H}}(d\lambda) \\ \rho_{\mathbf{H}}(d\lambda) \end{pmatrix} \\
&+ \gamma\tau_1\alpha_t^2 \int \lambda' s_{t,1}(\lambda')\rho_{\mathbf{H}}(d\lambda') \begin{pmatrix} \rho_{\mathbf{H}}(d\lambda) \\ \rho_{\mathbf{H}}(d\lambda) \\ \rho_{\mathbf{H}}(d\lambda) \end{pmatrix}
\end{aligned}
\tag{79}
$$

Now we simply observe that right-hand side of (79) has exactly the form (74) with $\mathbf{G}_{t+1}(\lambda), \mathbf{s}_{t+1}(\lambda)$ expressed through $\mathbf{G}_t(\lambda), \mathbf{s}_t(\lambda)$ according to (75),(76).

$\square$

Note that updates of matrix $\mathbf{G}_t(\lambda)$ and vector $\mathbf{s}_t(\lambda)$ in representation (74) depend only on learning algorithm parameters $\alpha_t, \beta_t$, combinations $\gamma\tau_1, \gamma\tau_2$, and non-zero eigenvalues of Hessian $\mathbf{H}$. Also, we stress that although in this proposition we considered all possible initial conditions with $\mathbf{J}_0 \neq 0, \mathbf{V} \neq 0$, in all other parts of the paper we focus on the case $\mathbf{J}_0 = \mathbf{V} = 0$.

### D.4   THEOREM 1

In this section we provide full version of theorem 1 and its proof, and also elaborate on how spectral measures $\rho_{\mathbf{C}_t}$ change during training. We keep the notations from previous sections, in particular the model Jacobian is $\mathbf{\Psi}$, the Hessian $\mathbf{H} = \frac{1}{N}\mathbf{\Psi}\mathbf{\Psi}^T$, the kernel matrix $\mathbf{K} = \frac{1}{N}\mathbf{\Psi}^T\mathbf{\Psi}$, and output space second moments $\mathbf{C}^{\text{out}} = \mathbf{\Psi}^T\mathbf{C}\mathbf{\Psi}$. However, for simplicity, in this section we consider only finite training datasets $N < \infty$ and features dimension $d \geq N$ (including $d = \infty$). The extension of the results to all values of $d, N$ seems to be rather technical and we expect it to lead to effectively the same conclusions.

We start with an analogue of theorem 1 but for the isolated noise term (8) containing the tensor $\widehat{R}$. It turns out that the SE approximation is closely related to what we call the *SE family* of tensors $\widehat{R}$:

$$
R_{i_1i_2i_3i_4} = \tau_1\delta_{i_1i_2}\delta_{i_3i_4} - \frac{\tau_2-1}{2}(\delta_{i_1i_3}\delta_{i_2i_4} + \delta_{i_1i_4}\delta_{i_2i_3})
\tag{80}
$$

It is easy to check that the respective $\mathbf{\Sigma}(\widehat{R})$ satisfies not only (7), but also a stronger version

$$
\mathbf{\Sigma}_{\widehat{R}}(\mathbf{C}) = \tau_1\mathbf{H}\operatorname{Tr}[\mathbf{HC}] - (\tau_2)\mathbf{HCH}
\tag{81}
$$

Let's focus only on the $\widehat{R}$-dependent term in (8), and for convenience rewrite it in terms $\mathbf{\Psi}$ as

$$
\mathbf{\Sigma}'_{\widehat{R}}(\mathbf{C}) = \frac{1}{N^2} \sum_{i_1,i_2,i_3,i_4=1}^{N} R_{i_1i_2i_3i_4}\langle \mathbf{\Psi}_{i_3}, \mathbf{C}\mathbf{\Psi}_{i_4}\rangle \mathbf{\Psi}_{i_1} \otimes \mathbf{\Psi}_{i_2},
\tag{82}
$$

where $\mathbf{\Psi}_i = \psi(\mathbf{x}_i)$ denotes the i'th column of $\mathbf{\Psi}$. Then we have

**Proposition 9.** *Consider a fixed tensor $\widehat{R}$, and let operators $\mathbf{\Sigma}', \mathbf{C}$ be related to each over as $\mathbf{\Sigma}' = \mathbf{\Sigma}'_{\widehat{R}}(\mathbf{C})$. Then, the following 3 statements are equivalent:*

1. *For any Jacobian $\mathbf{\Psi}$ and initial state $\mathbf{C}$, spectral measure $\rho_{\mathbf{\Sigma}'}$ depends on $\mathbf{C}$ only through its spectral measure $\rho_{\mathbf{C}}$, and on $\mathbf{\Psi}$ only through non-zero eigenvalues $\lambda_k$ of Hessian $\mathbf{H}$.*

2. *Tensor $\hat{R}$ belongs to SE family (80).*

3. *For any Jacobian $\mathbf{\Psi}$ and initial state $\mathbf{C}$, the operator $\mathbf{\Sigma}'$ does not change by the replacement of $\mathbf{\Psi}$ with $\widetilde{\mathbf{\Psi}} = \mathbf{\Psi}\mathbf{U}^T$ in (82) for any orthogonal matrix $\mathbf{U}$.*

*Proof.* **From (2) to (3).** Substituting SE form (80) of tensor $\widehat{R}$ into (82) we get

$$\mathbf{\Sigma}' = \tau_1 \mathbf{H} \operatorname{Tr}[\mathbf{HC}] + \tau_2 \mathbf{HCH} \tag{83}$$

Then statement **(3)** follows from invariance of Hessian under orthogonal transformations of the Jacobian

$$\widetilde{\mathbf{H}} = \frac{1}{N}\widetilde{\mathbf{\Psi}}\widetilde{\mathbf{\Psi}}^T = \frac{1}{N}\mathbf{\Psi}\mathbf{U}^T\mathbf{U}\mathbf{\Psi}^T = \mathbf{H} \tag{84}$$

**From (3) to (2).** Denote $\widetilde{\mathbf{\Sigma}}'$ the result of (82) but rotated Jacobian $\widetilde{\mathbf{\Psi}} = \mathbf{\Psi}\mathbf{U}^T$ in the right-hand side. It is convenient to represent $\widetilde{\mathbf{\Sigma}}'$ as

$$\widetilde{\mathbf{\Sigma}}' = \frac{1}{N^2} \sum_{i_1,i_2,i_3,i_4=1}^{N} \widetilde{R}_{i_1 i_2 i_3 i_4} C^{\text{out}}_{i_3 i_4} \mathbf{\Psi}_{i_1} \otimes \mathbf{\Psi}_{i_2} \tag{85}$$

where $\widehat{\widetilde{R}}$ is the rotated tensor $\widehat{R}$ with coordinates $\widetilde{R}_{i_1 i_2 i_3 i_4} = U_{j_1 i_1} U_{j_2 i_2} U_{j_3 i_3} U_{j_4 i_4} R_{j_1 j_2 j_3 j_4}$. As statement **(3)** implies $\widetilde{\mathbf{\Sigma}}' = \mathbf{\Sigma}'$, we get

$$0 = \widetilde{\mathbf{\Sigma}}' - \mathbf{\Sigma}' = \frac{1}{N^2} \sum_{i_1,i_2,i_3,i_4=1}^{N} \delta R_{i_1 i_2 i_3 i_4} C^{\text{out}}_{i_3 i_4} \mathbf{\Psi}_{i_1} \otimes \mathbf{\Psi}_{i_2} \tag{86}$$

where the tensor $\delta\widehat{R} = \widehat{\widetilde{R}} - \widehat{R}$ is the difference between original and "rotated" tensor $\widehat{R}$. Note that $\widehat{R}$, $\widehat{\widetilde{R}}$ and therefore $\delta\widehat{R} = \widehat{\widetilde{R}} - \widehat{R}$ are symmetric w.r.t. permutation of the first two and the last two indices. Taking Jacobian with the full rank $\operatorname{rank}(\mathbf{\Psi}) = N$ and exploiting permutation symmetry $i_1 \leftrightarrow i_2$ of $\delta\widehat{R}$ we notice that (86) implies

$$\sum_{i_3,i_4=1}^{N} \delta R_{i_1 i_2 i_3 i_4} C^{\text{out}}_{i_3 i_4} = 0 \tag{87}$$

Next, we observe that for full rank $\mathbf{\Psi}$ we can choose $\mathbf{C}$ such that $\mathbf{C}^{\text{out}}$ is an arbitrary positive-definite matrix. Then, permutation symmetry $i_3 \leftrightarrow i_4$ and arbitrariness of $\mathbf{C}^{\text{out}}$ implies $\delta\widehat{R} = 0$. As $\mathbf{U}$ was arbitrary, we see that

$$U_{j_1 1_1} U_{j_2 i_2} U_{j_3 i_3} U_{j_4 i_4} R_{j_1 j_2 j_3 j_4} = R_{i_1 i_2 i_3 i_4}, \quad \forall \mathbf{U} \in \mathrm{O}(N), \tag{88}$$

where $\mathrm{O}(N)$ is the group of orthogonal matrices of size $N$. According to Weyl Weyl (1939) (see Jeffreys (1973) for compact reference), the equality above implies that $\widehat{R}$ is isotropic - expressed as a sum of products of dirac deltas $\delta_{ii'}$ and possibly one anti-symmetric symbol $\epsilon_{j_1 j_2 \ldots j_N}$. In our case we can not have $\epsilon$ tensor: for $N > 4$ it has higher rank $N > \operatorname{rank}(\hat{R})$; for $N = 4$ it does not satisfy symmetries of $\widehat{R}$; for $N = 3$ it cannot be present because 3 is odd; for $N = 2$ all three possible combinations $\delta_{i_1 i_2}\epsilon_{i_3 i_4}, \delta_{i_3 i_4}\epsilon_{i_1 i_2}, \delta_{i_1 i_4}\epsilon_{i_2 i_3}$ again do not satisfy permutation symmetries $i_1 \leftrightarrow i_2$ and $i_3 \leftrightarrow i_4$. Then we left with 3 possible combinations of Dirac deltas

$$\mathbf{R}_{i_1 i_2 i_3 i_4} = c_1 \delta_{i_1 i_2} \delta_{i_3 i_4} + c_2 \delta_{i_1 i_3} \delta_{i_2 i_4} + c_3 \delta_{i_1 i_4} \delta_{i_2 i_3} \tag{89}$$

Again, from symmetry considerations we get $c_2 = c_3$, and finally recover (80) by setting $c_1 = \tau_1$ and $c_2 = c_3 = \frac{1}{2}\tau_2$.

**From (2) to (1)**. We take arbitrary test function $\phi(\lambda)$ and calculate its average with spectral measure $\rho_{\mathbf{\Sigma}'}$

$$\int \phi(\lambda)\rho_{\mathbf{\Sigma}'}(d\lambda) = \operatorname{Tr}[\phi(\mathbf{H})(\tau_1 \mathbf{H} \operatorname{Tr}[\mathbf{HC}] + \tau_2 \mathbf{HCH})]$$

$$= \tau_2 \operatorname{Tr}[\phi(\mathbf{H})\mathbf{H}^2\mathbf{C}] + \tau_1 \operatorname{Tr}[\mathbf{CH}] \operatorname{Tr}[\phi(\mathbf{H})\mathbf{H}] \tag{90}$$

$$= \int \phi(\lambda)\tau_2\lambda^2 \rho_{\mathbf{C}}(d\lambda) + \tau_1 \operatorname{Tr}[\phi(\mathbf{H})\mathbf{H}] \int \lambda\rho_{\mathbf{C}}(d\lambda)$$

As $\phi(\lambda)$ is arbitrary, we obtain

$$\rho_{\boldsymbol{\Sigma}'}(d\lambda) = \tau_2\lambda^2\rho_{\mathbf{C}}(d\lambda) + \Big(\tau_1\int\lambda'\rho_{\mathbf{C}}(d\lambda')\Big)\rho_{\mathbf{H}}(d\lambda) \tag{91}$$

As the right-hand side depends only on the initial spectral measure $\rho_{\mathbf{C}}(d\lambda)$ and the Hessian spectral measure $\rho_{\mathbf{H}}(d\lambda)$, statement **(1)** follows from the fact that $\rho_{\mathbf{H}}(d\lambda)$ depends only on non-zero eigenvalues of $\mathbf{H}$.

**From (1) to (2).** We take arbitrary $\mathbf{U}\in\mathrm{O}(N)$ and follow the notation of the **From (3) to (2)** proof. As we already noted, original and rotated Jacobians have the same hessian $\widetilde{\mathbf{H}}=\mathbf{H}$, and therefore eigenvalues $\lambda_k$. Then, as we do not change $\mathbf{C}$ with the rotation of Jacobian, statement **(3)** implies that spectral measure of $\boldsymbol{\Sigma}'$ and $\widetilde{\boldsymbol{\Sigma}}'$ are the same: $\rho_{\boldsymbol{\Sigma}'} = \rho_{\widetilde{\boldsymbol{\Sigma}}'}$. For an arbitrary test function $\phi(\lambda)$ we have

$$\begin{aligned}
0 &= \int\phi(\lambda)(\rho_{\widetilde{\boldsymbol{\Sigma}}'}(d\lambda) - \rho_{\boldsymbol{\Sigma}'}(d\lambda))\\
&= \frac{1}{N^2}\mathrm{Tr}[\phi(\mathbf{H})\sum_{i_1i_2}(\delta\widehat{R}\mathbf{C}^{\mathrm{out}})_{i_1i_2}\boldsymbol{\Psi}_{i_1}\otimes\boldsymbol{\Psi}_{i_2}]\\
&= \frac{1}{N^2}\sum_{i_1i_2}\boldsymbol{\Psi}_{i_1}^T\phi(\mathbf{H})\boldsymbol{\Psi}_{i_2}(\delta\widehat{R}\mathbf{C}^{\mathrm{out}})_{i_1i_2}\\
&= \mathrm{Tr}\Big[\phi(\mathbf{K})\mathbf{K}(\delta\widehat{R}\mathbf{C}^{\mathrm{out}})\Big] = 0
\end{aligned} \tag{92}$$

Let us now fix some arbitrary output space matrix $\mathbf{C}_*^{\mathrm{out}}$. As $\boldsymbol{\Psi}$ and $\mathbf{C}$ were arbitrary, we can choose them so that we get arbitrary $\mathbf{K}$ but $\mathbf{C}^{\mathrm{out}} = \mathbf{C}_*^{\mathrm{out}}$. It means that in the last line in (92) we can consider $\mathbf{C}^{\mathrm{out}}$ to be fixed, but $\mathbf{K},\phi$ to be arbitrary. Then (92) implies that $\delta\widehat{R}\mathbf{C}^{\mathrm{out}} = 0$ and we can repeat respective steps of **From (3) to (2)** proof to get the statement **(2)**. $\qquad\square$

With proposition 9 as a basis, we are ready to state and prove the full version of theorem 1.

**Theorem 3.** *Consider SGD dynamics* (4) *with noise term* (8)*, initial condition* $\mathbf{v}_0 = 0$ *and* $\gamma, \alpha_0 > 0$. *If tensor* $\widehat{R}$ *is fixed, the following statements are equivalent:*

1. *For any Jacobian* $\boldsymbol{\Psi}$ *and initial state* $\mathbf{C}_0$*, spectral measures* $\rho_{\mathbf{C}_t}$ *occurring during SGD are uniquely determined by the initial spectral measure* $\rho_{\mathbf{C}_0}$*, the eigenvalues* $\lambda_k$ *of Hessian* $\mathbf{H}$*, and SGD parameters* $\{\alpha_t, \beta_t\}_{t=0}^\infty$.

2. *Tensor* $\widehat{R}$ *is given by* $R_{i_1i_2i_3i_4} = \tau_1\delta_{i_1i_2}\delta_{i_3i_4} - \frac{\tau_2-1}{2}(\delta_{i_1i_3}\delta_{i_2i_4} + \delta_{i_1i_4}\delta_{i_2i_3})$ *for some* $\tau_1, \tau_2$*, and Eq.* (7) *holds.*

3. *For any Jacobian* $\boldsymbol{\Psi}$ *and initial state* $\mathbf{C}_0$*, the trajectory* $\{\mathbf{C}_t\}_{t=0}^\infty$ *is invariant under transformations* $\widetilde{\boldsymbol{\Psi}} = \boldsymbol{\Psi}\mathbf{U}^T$ *of the feature matrix* $\boldsymbol{\Psi}$ *by any* $\mathbf{U}\in\mathrm{O}(N)$.

*Proof.* **From (1) to (2).** Let us consider the first iteration of SGD. The initial condition $\mathbf{v}_0 = 0$ implies $\mathbf{J}_0 = \mathbf{V}_0 = 0$. Also, denote for simplicity $\boldsymbol{\Sigma}_t = \boldsymbol{\Sigma}_{\widetilde{R}}(\mathbf{C}_t)$. Then, according to (4) we have

$$\mathbf{C}_1 = (\mathbf{I} - \alpha_0\mathbf{H})\mathbf{C}_0(\mathbf{I} - \alpha_0\mathbf{H}) + \gamma\alpha_0^2\boldsymbol{\Sigma}_0 \tag{93}$$

Respective spectral measure is

$$\rho_{\mathbf{C}_1} = (1-\alpha_0\lambda)^2\rho_{\mathbf{C}_0} + \gamma\alpha_0^2\rho_{\boldsymbol{\Sigma}_0} = \Big((1-\alpha_0\lambda)^2 - \gamma\alpha_0^2\lambda^2\Big)\rho_{\mathbf{C}_0} + \gamma\alpha_0^2\rho_{\boldsymbol{\Sigma}_0'} \tag{94}$$

Then, according to statement **(1)**, we get that $\rho_{\boldsymbol{\Sigma}_0'}$ is uniquely determined by $\rho_{\mathbf{C}_0}$ and eigenvalues of $\mathbf{H}$, and therefore $\widehat{R}$ is given by (80) according to equivalence of statements **(1)** and **(2)** of proposition 9.

**From (3) to (2).** Considering first iteration of SGD as in the previous step, we get that $\boldsymbol{\Sigma}_0'$ is invariant under orthogonal transformations of Jacobian $\boldsymbol{\Psi}$. Then, statement **(2)** follows from equivalence of statements **(3)** and **(2)** of proposition 9.

**From (2) to (3).** We proceed by induction. First, notice that $\mathbf{C}_0, \mathbf{J}_0, \mathbf{V}_0$ are (trivially) invariant under orthogonal transformations of Jacobian $\widetilde{\mathbf{\Psi}} = \mathbf{\Psi}$.

To make an induction step, assume that $\mathbf{C}_t, \mathbf{J}_t, \mathbf{V}_t$ are invariant under orthogonal transformations of Jacobian. Then, according to equivalence of statements **(3)** and **(2)** of proposition 9 and invariance of Hessian $\mathbf{H}$ under orthogonal transformations of Jacobian $\mathbf{\Psi}$ we get that $\mathbf{\Sigma}_t$ is also invariant. As first term in (4) depend on $\mathbf{\Psi}$ only through $\mathbf{H}$ we get that the whole right-hand side of (4) is invariant under orthogonal transformations of Jacobian, and so are $\mathbf{C}_{t+1}, \mathbf{J}_{t+1}, \mathbf{V}_{t+1}$.

**From (2) to (1)** As noted in the statement **(2)**, SE approximation (7) is satisfied. It means that we can apply proposition 8 with initial conditions $\rho_{\mathbf{J}_0} = \rho_{\mathbf{V}_0} = 0$. Then, according to representation (74) and update rules (75),(76), spectral measures $\rho_{\mathbf{C}_t}$ are uniquely determined by the initial spectral measure $\rho_{\mathbf{C}_0}$, the eigenvalues $\lambda_k$ of Hessian $\mathbf{H}$, SGD parameters $\{\alpha_t, \beta_t\}_{t=0}^{\infty}$, and parameters $\gamma, \tau_1, \tau_2$. Recalling that $\gamma, \tau_1, \tau_2$ are considered to be fixed in statement **(1)** finishes the proof. $\qquad\square$

### D.5 TRANSLATION INVARIANT MODELS

The purpose of this section is to prove Proposition 2.

**Proposition 2.** *Consider a problem with regular grid dataset $\mathcal{D} = \{\mathbf{x_i} = (\frac{2\pi i_1}{N_1}, \ldots, \frac{2\pi i_d}{N_d})\}$, $\quad \mathbf{i} \in (\mathbb{Z}/N_1\mathbb{Z}) \times \cdots \times (\mathbb{Z}/N_d\mathbb{Z})$ on the d-dimensional torus $\mathbb{T}^d = (\mathbb{R}/2\pi\mathbb{Z})^d$ and with translation invariant kernel $K(\mathbf{x}, \mathbf{x}') = K(\mathbf{x} - \mathbf{x}')$. Then (7) holds exactly with $\tau_1 = 1, \tau_2 = 1$.*

*Proof.* Statement (7) with $\tau_1 = 1, \tau_2 = 1$ is equivalent to

$$\operatorname{Tr}(\phi(\mathbf{H})\mathbf{\Sigma}) = \operatorname{Tr}(\mathbf{H}\mathbf{C})\operatorname{Tr}(\mathbf{H}\phi(\mathbf{H})) - \operatorname{Tr}(\mathbf{H}^2\phi(\mathbf{H})\mathbf{C}), \quad \forall\phi. \tag{95}$$

We will be proving this latter statement. Denote the grid-indexing set $(\mathbb{Z}/N_1\mathbb{Z}) \times \cdots \times (\mathbb{Z}/N_d\mathbb{Z})$ by $\mathbb{Z}/\mathbf{N}\mathbb{Z}$. Thanks to translation invariance of the data set and kernel, the eigenvectors $\mathbf{u}_k$ of $\mathbf{H}$ have an explicit representation in the $\mathbf{x}$-domain in terms of Fourier modes. Specifically, let $\mathbf{K}$ be the $N \times N$ kernel matrix with matrix elements

$$K_{\mathbf{ij}} = K_{\mathbf{i-j}} = K_{\mathbf{j-i}} = \frac{1}{N}\langle \psi(\mathbf{x_i}), \psi(\mathbf{x_j}) \rangle, \quad \mathbf{i}, \mathbf{j} \in \mathbb{Z}/\mathbf{N}\mathbb{Z}. \tag{96}$$

Consider the functions $\phi_{\mathbf{k}} : \mathcal{D} \to \mathbb{C}$ given by

$$\phi_{\mathbf{k}}(\mathbf{x_i}) = N^{-1/2}e^{\sqrt{-1}\mathbf{k}\cdot\mathbf{x_i}} \equiv \phi_{\mathbf{ki}}, \quad \mathbf{k} \in \mathbb{Z}/\mathbf{N}\mathbb{Z}. \tag{97}$$

These functions form an orthonormal basis in $\mathbb{C}^{\mathcal{D}}$ and diagonalize the matrix $\mathbf{K}$:

$$K_{\mathbf{ij}} = \sum_{\mathbf{k}\in\mathbb{Z}/\mathbf{N}\mathbb{Z}} \lambda_{\mathbf{k}}\phi_{\mathbf{ki}}\overline{\phi}_{\mathbf{kj}} \tag{98}$$

with

$$\lambda_{\mathbf{k}} = \sum_{\mathbf{i}\in\mathbb{Z}/\mathbf{N}\mathbb{Z}} K_{\mathbf{i}}\phi_{\mathbf{ki}}. \tag{99}$$

Operator $\mathbf{H}$ is unitarily equivalent to $\mathbf{K}$ up to its nullspace. Assuming that the Hilbert space of $\mathbf{H}$ is complexified, the respective normalized eigenvectors $\mathbf{u_k}$ of $\mathbf{H}$ can be written as

$$\mathbf{u_k} = (N\lambda_{\mathbf{k}})^{-1/2} \sum_{\mathbf{i}\in\mathbb{Z}/\mathbf{N}\mathbb{Z}} \psi(\mathbf{x_i})\phi_{\mathbf{ki}}, \quad \mathbf{H}\mathbf{u_k} = \lambda_{\mathbf{k}}\mathbf{u_k}. \tag{100}$$

Assume for the moment that the problem is nondegenerate in the sense that all $\lambda_{\mathbf{k}} > 0$ so that all the vectors $\mathbf{u_k}$ are well-defined. It follows in particular that the indices $k$ of the eigenvalues $\lambda_k$ can be identified with $\mathbf{k} \in \mathbb{Z}/\mathbf{N}\mathbb{Z}$. By inverting relation (100),

$$\psi(\mathbf{x_i}) = \sum_{\mathbf{k}\in\mathbb{Z}/\mathbf{N}\mathbb{Z}} (N\lambda_{\mathbf{k}})^{1/2}\mathbf{u_k}\overline{\phi}_{\mathbf{ki}}. \tag{101}$$

We can then find the diagonal elements of $\boldsymbol{\Sigma}$ using Eq. (5) and Eqs. (97), (101):

$$\Sigma_{\mathbf{kk}} = \Big(\frac{1}{N}\sum_{i=1}^{N}\langle\boldsymbol{\psi}(\mathbf{x}_i), \mathbf{C}\boldsymbol{\psi}(\mathbf{x}_i)\rangle\boldsymbol{\psi}(\mathbf{x}_i)\otimes\boldsymbol{\psi}(\mathbf{x}_i) - \mathbf{HCH}\Big)_{\mathbf{kk}} \tag{102}$$

$$= N\sum_{\mathbf{i}}\lambda_{\mathbf{k}}\phi_{\mathbf{ki}}\overline{\phi}_{\mathbf{ki}}\sum_{\mathbf{k'},\mathbf{l'}}\lambda_{\mathbf{k'}}^{1/2}C_{\mathbf{k'l'}}\lambda_{\mathbf{l'}}^{1/2}\overline{\phi}_{\mathbf{k'i}}\phi_{\mathbf{l'i}} - \lambda_{\mathbf{k}}^2 C_{\mathbf{kk}} \tag{103}$$

$$= \lambda_{\mathbf{k}}\sum_{\mathbf{k'},\mathbf{l'}}\lambda_{\mathbf{k'}}^{1/2}\lambda_{\mathbf{l'}}^{1/2}C_{\mathbf{k'l'}}\sum_{\mathbf{i}}\overline{\phi}_{\mathbf{k'i}}\phi_{\mathbf{l'i}} - \lambda_{\mathbf{k}}^2 C_{\mathbf{kk}} \tag{104}$$

$$= \lambda_{\mathbf{k}}\sum_{\mathbf{l}}\lambda_{\mathbf{l}}C_{\mathbf{ll}} - \lambda_{\mathbf{k}}^2 C_{\mathbf{kk}}. \tag{105}$$

It follows that

$$\operatorname{Tr}(\phi(\mathbf{H})\boldsymbol{\Sigma}) = \sum_{\mathbf{k}}\phi(\lambda_{\mathbf{k}})\Sigma_{\mathbf{kk}} \tag{106}$$

$$= \sum_{\mathbf{k}}\phi(\lambda_{\mathbf{k}})\lambda_{\mathbf{k}}\sum_{\mathbf{l}}\lambda_{\mathbf{l}}C_{\mathbf{ll}} - \sum_{\mathbf{k}}\phi(\lambda_{\mathbf{k}})\lambda_{\mathbf{k}}^2 C_{\mathbf{kk}} \tag{107}$$

$$= \operatorname{Tr}(\mathbf{H}\phi(\mathbf{H}))\operatorname{Tr}(\mathbf{HC}) - \operatorname{Tr}(\mathbf{H}^2\phi(\mathbf{H})\mathbf{C}), \tag{108}$$

which is the desired Eq. (95).

Now we comment on the degenerate case, when $\lambda_{\mathbf{k}} = 0$ for some $\mathbf{k}$. This occurs if there is linear dependence between some $\psi(\mathbf{x_i})$, i.e. $\dim\operatorname{Ran}(\mathbf{H}) < N$. The vectors $\mathbf{u_k}$ given by (100) with $\lambda_{\mathbf{k}} \neq 0$ form a basis in $\operatorname{Ran}(\mathbf{H})$. Inverse relation (101) remains valid with arbitrary vectors $\mathbf{u_k}$ for $\mathbf{k}$ such that $\lambda_{\mathbf{k}} = 0$ (this can be seen, for example, by lifting the degeneracy of the problem with a regularization $\varepsilon$ and then letting $\varepsilon \to 0$). As a result, Eq. (105) remains valid for $\mathbf{k}$ such that $\lambda_{\mathbf{k}} > 0$. On the other hand, if $\mathbf{u}_k$ is an eigenvector corresponding to the eigenvalue $\lambda_k = 0$, i.e. $\mathbf{u}_k \in \ker\mathbf{H}$, then $\mathbf{u}_k$ is orthogonal to $\psi(\mathbf{x_i})$ for all $\mathbf{i}$, and then $\Sigma_{\mathbf{kk}} = 0$ for such $k$. Thus, formula (105) remains valid in this case too, and so Eq. (108) holds true. $\qquad\square$

## E  GENERATING FUNCTIONS AND THEIR APPLICATIONS

### E.1  REDUCTION TO GENERATING FUNCTIONS

We clarify some details of derivations in Section 4.

**Gas of rank-1 operators.**  Let us first discuss SE condition (7). Recall that we assumed $\mathbf{H}$ to be diagonalized by the eigenbasis $\mathbf{u}_k$ with eigenvalues $\lambda_k$. If the eigenvalues of $\mathbf{H}$ are non-degenerate, SE condition (7) can be equivalently written as a relation between the diagonal matrix elements $\Sigma_{kk}, C_{kk}$ of the matrices $\boldsymbol{\Sigma}, \mathbf{C}$:

$$\Sigma_{kk} = \lambda_k\sum_l\lambda_l C_{ll} - \tau\lambda_k^2 C_{kk} \quad \forall k \tag{109}$$

(where we have incorporated the convention $\tau_1 = 1, \tau_2 = \tau$). More generally, if several $\lambda_k$ may be equal to some $\lambda$ and so the choice of eigenvectors $\mathbf{u}_k$ is not unique, conditions (109) should be replaced by weaker conditions (invariant w.r.t. this choice):

$$\sum_{k:\lambda_k=\lambda}\Sigma_{kk} = \sum_{k:\lambda_k=\lambda}\lambda\sum_l\lambda_l C_{ll} - \tau\lambda^2\sum_{k:\lambda_k=\lambda}C_{kk}, \quad \forall\lambda\in\operatorname{spec}(\mathbf{H}). \tag{110}$$

Note, however, that representation (109) can be used without any restriction of generality if we are interested in the observables $\operatorname{Tr}[\phi(\mathbf{H})\mathbf{C}_t]$ (in particular, $L(t)$) along the optimization trajectory (5): by Proposition 8 such quantities are uniquely determined by initial conditions and the learning rates of SGD once the SE approximation is assumed.

Representations (9) – (11) now follow directly from the evolution law (5), by inspecting the evolution of diagonal elements in submatrices $\mathbf{C}_t, \mathbf{J}_t, \mathbf{V}_t$ using Eq. (109).

**Genarting functions.** Eq. (16) follows from a relation between generating functions obtained using Eq. (15):

$$\widetilde{L}(z) = \sum_{t=0}^{\infty} L(t) z^t \tag{111}$$

$$= \sum_{T=0}^{\infty} \left( \frac{1}{2} V_{T+1} + \sum_{t_m=1}^{T} U_{T+1-t_m} L(t_m - 1) \right) z^T \tag{112}$$

$$= \frac{1}{2} \sum_{T=0}^{\infty} V_{T+1} z^T + \sum_{t=0}^{\infty} \sum_{s=0}^{\infty} U_{t+1} L(s) z^{t+s+1} \tag{113}$$

$$= \frac{1}{2} \widetilde{V}(z) + z \widetilde{U}(z) \widetilde{L}(z), \tag{114}$$

where we have used the substitution $T - t_m = t, t_m - 1 = s$.

The resolvent formulas in Eqs. (17), (19) follow from the definitions of $U_t, V_s$ (Eqs. (13), (14)).

We explain now the expansions (18), (20) of the noise and signal generating functions $\widetilde{U}(z), \widetilde{V}(z)$ as series of scalar rational functions. Consider first expansion (18) for $\widetilde{U}(z)$. The terms of this expansion correspond to the eigenvalues $\lambda_k$ of **H**. For each eigenvalue, the respective term is, by Eq. (17),

$$\lambda^2 \langle \left( \begin{smallmatrix} 1 & 0 \\ 0 & 0 \end{smallmatrix} \right), (1 - z A_\lambda)^{-1} \left( \begin{smallmatrix} 1 & 1 \\ 1 & 1 \end{smallmatrix} \right) \rangle, \tag{115}$$

where $A_\lambda$ is the linear operator acting on the space of $2 \times 2$ matrices and given by Eq. (11). The scalar product in (115) can be computed as the component $x_{11}$ of the solution of the $4 \times 4$ linear system

$$(1 - z A_\lambda) \left( \begin{smallmatrix} x_{11} & x_{12} \\ x_{21} & x_{22} \end{smallmatrix} \right) = \left( \begin{smallmatrix} 1 & 1 \\ 1 & 1 \end{smallmatrix} \right). \tag{116}$$

Since all the components of the $4 \times 4$ matrix of the operator $1 - z A_\lambda$ are polynomial in the parameters $\alpha, \beta, \gamma, \tau, \lambda, z$, the resulting $x_{11}$ is a rational function of these parameters. The explicit form of $x_{11}$ is tedious to derive by hand, but it can be easily obtained with the help of a computer algebra system such as Sympy (Meurer et al., 2017). We provide the respective Jupyter notebook in the supplementary material.[2]

The terms of $\widetilde{V}$ are obtained similarly, but by solving the equation

$$(1 - z A_\lambda) \left( \begin{smallmatrix} x_{11} & x_{12} \\ x_{21} & x_{22} \end{smallmatrix} \right) = \left( \begin{smallmatrix} 1 & 0 \\ 0 & 0 \end{smallmatrix} \right) \tag{117}$$

instead of Eq. (116).

### E.2 CONVERGENCE CONDITIONS

**Proof of Proposition 3.** Note first that Proposition 3 is implied by the following lemma:

**Lemma 1.** *Assume (as in Proposition (3)) that $\beta \in (-1, 1)$, $0 < \alpha < 2(\beta+1)/\lambda_{\max}$ and $\gamma \in [0, 1]$. Then*

1. *$S(\alpha, \beta, \gamma, \lambda, z) > 0$ for all $z \in (0, 1)$;*

2. *$\widetilde{U}(z), \widetilde{V}(z)$ are analytic in an open neighborhood of the interval $(0, 1)$ and have no singularities there;*

3. *$z\widetilde{U}(z)$ is monotone increasing for all $z \in (0, 1)$.*

Indeed, by statement 2) and representation (16), a singularity of $\widetilde{L}(z) = \widetilde{V}(z)/(2(1 - z\widetilde{U}(z)))$ on the interval $(0, 1)$ can only be a pole resulting from a zero of the denominator $1 - z\widetilde{U}(z)$. By statement 3), such a zero is present if and only if $\widetilde{U}(z) > 1$, as claimed. It remains to prove the lemma.

---

[2] https://colab.research.google.com/drive/1eai1apCMCeLLbGIC7vfjbXNOKZ3CaIbw?usp=sharing

*Proof of Lemma 1.*

1) Our proof of statement 1 is computer-assisted. Note first that this statement can be written as the condition $W = \varnothing$ for a semi-algebraic set $W \subset \mathbb{R}^4$ defined by several polynomial inequalities with integer coefficients:

$$W = \{\exists (a, \beta, \gamma, z) \in \mathbb{R}^4 : (-1 < \beta < 1) \wedge (0 < a < 2(\beta + 1)) \tag{118}$$
$$\wedge (0 \le \gamma \le 1) \wedge (0 < z < 1) \wedge (S(a, \beta, \gamma, 1, z) \le 0)\}, \tag{119}$$

where we have introduced the variable $a = \alpha\lambda$. By Tarski-Seidenberg theorem, the emptyness of a semi-algebraic set in $\mathbb{R}^n$ is an algorithmically decidable problem (solvable by sequential elimination of existential quantifiers, via repeated standard operations such as differentiation, evaluation of polynomials, division of polynomials with remainder, etc., see e.g. (Basu, 2014)). This procedure is implemented in some computer algebra systems, e.g. in REDUCE/Redlog (Dolzmann & Sturm, 1997). We used this system to verify that $W = \varnothing$:

```
1: rlset reals$
2: S := a**2*b*g*z**2 + a**2*b*z**2 + a**2*g*z - a**2*z - 2*a*b**2*z**2 -
2*a*b*z**2 + 2*a*b*z + 2*a*z - b**3*z**3 + b**3*z**2 + b**2*z**2 - b**2*z
+ b*z**2 - b*z - z + 1$
3: W := ex({a,b,g,z}, -1<b and b<1 and 0<a and a<2*(b+1) and 0<=g and g<=1
and 0<z and z<1 and S<=0)$
4: rlqe W;
false
```

2) Statement 2 follows from statement 1 and the fact that $S(\alpha, \beta, \gamma, \lambda, z) \to (1-z)(1-\beta z)(1-\beta^2 z)$ as $\lambda \to 0$. Indeed, since the series $\sum_k \lambda_k^2$ and $\sum_k \lambda_k C_{kk,0}$ converge, singularities in $\widetilde{U}, \widetilde{V}$ outside the points $z = 1, \beta^{-1}, \beta^{-2}$ can only be poles of some of the terms of expansions (18), (20) associated with zeros of $S$, which are excluded by statement 1.

3) Our proof of statement 3 is also computer-assisted and similar to the proof of statement 1. First we compute

$$\tfrac{d}{dz}(z\widetilde{U}(z)) = \gamma\alpha^2 \sum_k \lambda_k^2 \frac{R(\alpha, \beta, \lambda_k, z)}{S^2(\alpha, \beta, \gamma\tau, \lambda_k, z)}, \tag{120}$$

where

$$R(\alpha, \beta, \lambda_k, z) = -2\alpha^2\beta\lambda_k^2 z^2 + 4\alpha\beta^2\lambda_k z^2 + 4\alpha\beta\lambda_k z^2 + \beta^4 z^4 + 2\beta^3 z^3 - 2\beta^3 z^2 - 2\beta^2 z^2 - 2\beta z^2 + 2\beta z + 1. \tag{121}$$

By statement 1, it is sufficient to check that $R \ge 0$ on the domain of interest. This is done by verifying the emptyness of the set

$$W_1 = \{\exists (a, \beta, z) \in \mathbb{R}^3 : (-1 < \beta < 1) \wedge (0 < a < 2(\beta + 1)) \tag{122}$$
$$\wedge (0 < z < 1) \wedge (R(a, \beta, 1, z) < 0)\} \tag{123}$$

in REDUCE/Redlog:

```
1: rlset reals$
2: R1 := -2*a**2*b*z**2 + 4*a*b**2*z**2 + 4*a*b*z**2 + b**4*z**4
+ 2*b**3*z**3 - 2*b**3*z**2 - 2*b**2*z**2 - 2*b*z**2 + 2*b*z + 1$
3: W1 := ex({a,b,z}, -1<b and b<1 and 0<a and a<2*(b+1) and 0<z and z<1
and R1<0)$
4: rlqe W1;
false
```

$\square$

We provide experimental verifications of statements 1 and 2 in the accompanying Jupyter notebook.

**Proof of Proposition 4.**   Recall representation (21)

$$\widetilde{U}(1) = \sum_k \frac{\lambda_k}{\lambda_k\big(\tau - \frac{1-\beta}{\gamma(1+\beta)}\big) + \frac{2(1-\beta)}{\alpha\gamma}} \tag{124}$$

and rewrite it as $\widetilde{U}(1) = f(\frac{2(1-\beta)}{\alpha\gamma}, Y)$, $Y = \frac{1-\beta}{\gamma(1+\beta)}$ where $f(x,y)$ is defined as

$$f(x,y) = \sum_k \frac{\lambda_k}{\lambda_k(\tau - y) + x} \tag{125}$$

Note that due to assumptions $\alpha\lambda_{\max} < 2(\beta + 1)$ and $\tau > 0$, the denominator in (124) is always positive. Then, from (124) we see that $\widetilde{U}(1)$ is monotonically increasing function of $\alpha$, and therefore at fixed $\beta, \gamma$ the convergence condition $\widetilde{U}(1) < 1$ can be written as $\alpha < \alpha_c$, or equivalently $\gamma\alpha_{\mathrm{eff}}/2 < 1/x_c$, where $x_c$ is the largest positive solution of $f(x, Y) = 1$. Also, original definition of $\lambda_{\mathrm{crit}}$ in 4 can be formulated in terms of $f$ is a unique positive solution of $f(\lambda_{\mathrm{crit}}, 0) = 1$. Now observe that $f(x, Y) > f(x, 0)$ for $x > x_c$, and $f(x, 0)$ is monotonically decreasing functions of $x$ for $x > 0$. These observations immediately imply that $x_c > \lambda_{\mathrm{crit}}$ and thus the statement (22).

Now we obtain the estimate (23) by abusing the fact $f(\lambda_{\mathrm{crit}}, 0) = f(x_c, Y) = 1$ we explicitly calculate the difference $f(\lambda_{\mathrm{crit}}, 0) - f(x_c, Y)$ and use it to bound $x_c - \lambda_{\mathrm{crit}}$

$$0 = f(\lambda_{\mathrm{crit}}, 0) - f(x_c, Y) = \sum_k \frac{\lambda_k\big(x_c - \lambda_{\mathrm{crit}} - \lambda_k Y\big)}{\big(\lambda_k(\tau - Y) + x_c\big)\big(\tau\lambda_k + \lambda_{\mathrm{crit}}\big)} \tag{126}$$

$$\sum_k \frac{\lambda_k(x_c - \lambda_{\mathrm{crit}})}{\big(\lambda_k(\tau - Y) + x_c\big)\big(\tau\lambda_k + \lambda_{\mathrm{crit}}\big)} = Y\sum_k \frac{\lambda_k^2}{\big(\lambda_k(\tau - Y) + x_c\big)\big(\tau\lambda_k + \lambda_{\mathrm{crit}}\big)} \tag{127}$$

$$\tag{128}$$

Next we bound l.h.s. from below with $\frac{x_c - \lambda_{\mathrm{crit}}}{\tau\lambda_m a x + \lambda_{\mathrm{crit}}}$ (by bounding denominator with $\lambda_{\mathrm{crit}}$ and equating the rest of the series to $f(x_c, Y)$). To bound the sum in r.h.s. we note that it can be expressed as $\sum_k a_k b_k$ were $\sum_k a_k = f(x_c, Y)$ and $\sum_k b_k = f(\lambda_{\mathrm{crit}}, 0)$ . Then, the upper bound $\sum_k a_k b_k < b_0 = \frac{\lambda_{\max}}{\tau\lambda_{\max} + \lambda_{\mathrm{crit}}}$ is constructed by noting that redistribution of the mass $a_k \to \delta_{k0}$ yields the maximum r.h.s. for fixed $b_k$ and under constraint $a_k \geq 0$, $\sum_k a_k \leq 1$. Thus we arrive at

$$\frac{x_c - \lambda_{\mathrm{crit}}}{\tau\lambda_{\max} + \lambda_{\mathrm{crit}}} \leq Y\frac{\lambda_{\max}}{\tau\lambda_{\max} + \lambda_{\mathrm{crit}}} \tag{129}$$

$$x_c - \lambda_{\mathrm{crit}} \leq \frac{1 - \beta}{\gamma(1 + \beta)}\lambda_{\max} \tag{130}$$

Noting that $\alpha_{\mathrm{eff}}^c(\beta, \gamma) = \frac{2}{\gamma x_c}$ we get (23) as

$$|\alpha_{\mathrm{eff}}^c(\beta, \gamma) - \frac{2}{\gamma\lambda_{\mathrm{crit}}}|/\alpha_{\mathrm{eff}}^c(\beta, \gamma) = \frac{x_c - \lambda_{\mathrm{crit}}}{\lambda_{\mathrm{crit}}} \leq \frac{\lambda_{\max}}{\lambda_{\mathrm{crit}}}\frac{1 - \beta}{\gamma(1 + \beta)} \tag{131}$$

### E.3   LOSS ASYMPTOTIC UNDER POWER-LAWS

The main purpose of this section is to prove theorem 2. First, note that if $\lambda_k \to 0$ and $C_{kk,0} > 0$ for infinitely many $k$ (non-strongly-convex problems, including power-laws (25)), then $r_L \leq 1$, thus excluding the possibility of exponential loss convergence. Indeed, $S(\alpha, \beta, \gamma, \lambda, z) \xrightarrow{\lambda \to 0} (1 - z)(1 - \beta z)(1 - \beta^2 z)$. It follows that for small $\lambda_k$ the respective terms in expansion (18) have three poles $z_{k,0}, z_{k,1}, z_{k,2}$ converging to $1, \beta^{-1}, \beta^{-2}$, respectively. By Eq. (16), $\widetilde{L}(z)$ has a removable singularity at $z_{k,0}$, moreover $\widetilde{L}(z_{k,0}) < 0$ if $C_{kk,0} > 0$ and $k$ is large enough, meaning that $r_L < z_{k,0}$. We then get $r_L \leq 1$ by taking the limit in $k$.

As $r_L \geq 1$ for convergent trajectories and $r_L \leq 1$ for non-strongly-convex problems, we conclude that $r_L = 1$. Then, we formulate a lemma allowing to calculate the $\varepsilon \to +0$, $\varepsilon = 1 - z$ asymptotics of the generating functions (18) and (20).

**Lemma 2.** *Suppose that $S_k$ and $C_k$ are two sequences such that $S_k = k^{-\nu}(1 + o(1))$ and $F_k = \sum_{l=k}^{\infty} C_l = k^{-\varkappa}(1 + o(1))$ as $k \to \infty$, with some constants $\nu, \varkappa > 0$. Also denote $\zeta = \frac{\varkappa}{\nu}$. Then*

*1.*

$$\sum_{l=k}^{\infty} C_l S_l = \frac{\zeta}{\zeta + 1} k^{-\varkappa - \nu}(1 + o(1)), \quad k \to \infty. \tag{132}$$

*2. Assume $\zeta < n$ for some $n > 0$, then*

$$\sum_{k=1}^{\infty} \frac{C_k}{(\varepsilon + S_k)^n} = \frac{\Gamma(\zeta + 1)\Gamma(n - \zeta)}{\Gamma(n)} \varepsilon^{\zeta - n}(1 + o(1)), \quad \varepsilon \to 0 + . \tag{133}$$

*Proof.*

**Part 1.** Fix $k$ and consider a discrete measure $\mu_k$ with atoms located at points $x_{l,k} = (l/k)^{-\nu}$:

$$\mu_k = k^{\varkappa} \sum_{l=k}^{\infty} C_l \delta_{x_{l,k}} \tag{134}$$

By assumption on $F_k$ as $k \to \infty$, the cumulative distribution function of $\mu_k$ converges to cumulative distribution function of measure $\mu(dx) = \mathbf{1}_{[0,1]} dx^{\zeta}$, i.e. the measure $\mu_k$ weakly converges to $\mu$.

Next, consider the function $f_k(x)$ defined at points $x_{l,k}$ by $f_k(x_{l,k}) = k^{\nu} S_l$, with $f_k(0) = 0$, linearly interpolated between the points $x_{l,k}$, and with $f(x > 1) = f(x_{k,k})$. Then $f_k(x)$ converges uniformly on $[0, 1]$ to $f(x) = x$ as $k \to \infty$.

The above two observations on $\mu_k$ and $f_k$ give

$$\lim_{k \to \infty} k^{\varkappa + \nu} \sum_{l=k}^{\infty} C_l S_l = \lim_{k \to \infty} \int_0^1 f_k(x) \mu_k(dx) = \int_0^1 x \zeta x^{\zeta - 1} dx = \frac{\zeta}{\zeta + 1} \tag{135}$$

**Part 2.** For any $a > 0$ let $k_{a,\varepsilon} = \lfloor (\varepsilon a)^{-1/\nu} \rfloor$. Rescale the sum of interest and divide it into two parts separated by index $k_{a,\varepsilon}$:

$$\varepsilon^{n-\zeta} \sum_{k=1}^{\infty} \frac{C_k}{(\varepsilon + S_k)^n} = \varepsilon^{n-\zeta} \sum_{k=1}^{k_{a,\varepsilon}} + \varepsilon^{n-\zeta} \sum_{k=k_{a,\varepsilon}+1}^{\infty} =: I_1(a, \varepsilon) + I_2(a, \varepsilon). \tag{136}$$

We will show that

$$\limsup_{\varepsilon \to 0+} I_1(a, \varepsilon) = O(a^{\zeta - n}), \tag{137}$$

$$\lim_{\varepsilon \to 0+} I_2(a, \varepsilon) = \int_0^a \frac{dx^{\zeta}}{(1 + x)^n} =: J(a). \tag{138}$$

These two fact will imply the statement of the lemma since $\zeta - n < 0$ and

$$\lim_{a \to +\infty} J(a) = \int_0^{\infty} \frac{\zeta x^{\zeta - 1}}{(1 + x)^n} dx \tag{139}$$

$$\overset{1+x=1/t}{=} \zeta \int_0^1 t^{n - \zeta - 1}(1 - t)^{\zeta - 1} dt \tag{140}$$

$$= \zeta B(n - \zeta, \zeta). \tag{141}$$

To prove Eq. (137), use summation by parts and the hypotheses on $S_k, F_k$:

$$\varepsilon^{n-\zeta} \sum_{k=1}^{k_{a,\varepsilon}} \frac{C_k}{(\varepsilon + S_k)^n} = \varepsilon^{n-\zeta} O\Big(\sum_{k=1}^{k_{a,\varepsilon}} C_k k^{\nu n}\Big) \tag{142}$$

$$= \varepsilon^{n-\zeta} O\Big(\sum_{k=1}^{k_{a,\varepsilon}} (F_k - F_{k+1}) k^{\nu n}\Big) \tag{143}$$

$$= \varepsilon^{n-\zeta} O\Big(F_1 - F_{k_{a,\varepsilon}+1} k_{a,\varepsilon}^{\nu n} + \sum_{k=2}^{k_{a,\varepsilon}} F_k (k^{\nu n} - (k-1)^{\nu n})\Big) \tag{144}$$

$$= \varepsilon^{n-\zeta} O(k_{a,\varepsilon}^{\nu n - \varkappa}) \tag{145}$$

$$= O(a^{\zeta-n}). \tag{146}$$

To prove Eq. (138), it is convenient to represent the sum $I_2(a, \varepsilon)$ as an integral over discrete measure $\mu_{a,\varepsilon}$ with atoms $x_{k,\varepsilon} = k^{-\nu}/\varepsilon$:

$$I_2(a, \varepsilon) = \int f_\varepsilon d\mu_{a,\varepsilon}(x), \tag{147}$$

$$\mu_{a,\varepsilon} = \varepsilon^{-\zeta} \sum_{k=k_{a,\varepsilon}+1}^{\infty} C_k \delta_{x_{k,\varepsilon}}, \tag{148}$$

$$f_\varepsilon(x_{k,\varepsilon}) = (1 + S_k/\varepsilon)^{-n}. \tag{149}$$

Again, by assumption on $F_k$, as $\varepsilon \to 0+$, the cumulative distribution functions of $\mu_{a,\varepsilon}$ converge to the cumulative distribution function of the measure $\mu_a(dx) = \mathbf{1}_{[0,a]} dx^\zeta$, i.e., the measures $\mu_{a,\varepsilon}$ weakly converge to $\mu_a$. At the same time, the functions $f_\varepsilon(x)$ converge to $(1+x)^{-n}$ uniformly on $[0, a]$. This implies desired convergence (138). □

Now we are ready to proceed with the proof of theorem 2.

**Theorem 2.** *Assume spectral conditions (25) with $\nu > 1$ and that parameters $\alpha, \beta, \gamma$ are as in Proposition 3 and such that convergence condition $\widetilde{U}(1) < 1$ is satisfied. Then*

$$\sum_{t=1}^{T} tL(t) = (1 + o(1)) \begin{cases} \frac{K\Gamma(\zeta+1)}{2(1-\widetilde{U}(1))(2-\zeta)} \left(\frac{2\alpha\Lambda}{1-\beta}\right)^{-\zeta} t^{2-\zeta}, & 2 - \zeta > 1/\nu, \\ \frac{\gamma\widetilde{V}(1)\Gamma(2-1/\nu)}{8(1-\widetilde{U}(1))^2} \left(\frac{2\alpha\Lambda}{1-\beta}\right)^{1/\nu} t^{1/\nu}, & 2 - \zeta < 1/\nu. \end{cases} \tag{26}$$

*Proof.* As asymptotic of partial sums $\sum_{t=1}^{T} tL(t)$ are associated with asymptotic of $\widetilde{L}'(1 - \varepsilon)$ at $\varepsilon \to +0$ we recall its expression (27)

$$\frac{d}{dz} \widetilde{L}(z) = \frac{\frac{d}{dz} \widetilde{V}(z)}{2(1 - z\widetilde{U}(z))} + \frac{\widetilde{V}(z) \frac{d}{dz}(z\widetilde{U}(z))}{2(1 - z\widetilde{U}(z))^2} \tag{150}$$

Full expressions for generating functions (18) and (20) can be significantly simplified in the limit $\varepsilon \to +0$ and $\lambda_k \to 0$

$$\widetilde{U}(1 - \varepsilon) = \gamma \sum_{k=1}^{\infty} \frac{\left(\frac{\alpha\lambda_k}{1-\beta}\right)^2}{\varepsilon + 2\frac{\alpha\lambda_k}{1-\beta}} (1 + O(\lambda_k) + O(\varepsilon)) \tag{151}$$

$$\widetilde{V}(1 - \varepsilon) = \sum_{k=1}^{\infty} \frac{\lambda_k C_{kk}}{\varepsilon + 2\frac{\alpha\lambda_k}{1-\beta}} (1 + O(\lambda_k) + O(\varepsilon)) \tag{152}$$

Differentiating we get

$$\widetilde{U}'(1-\varepsilon) = \gamma \sum_{k=1}^{\infty} \frac{\left(\frac{\alpha\lambda_k}{1-\beta}\right)^2}{\left(\varepsilon + 2\frac{\alpha\lambda_k}{1-\beta}\right)^2}(1 + O(\lambda_k) + O(\varepsilon)) \tag{153}$$

$$\widetilde{V}'(1-\varepsilon) = \sum_{k=1}^{\infty} \frac{\lambda_k C_{kk}}{\left(\varepsilon + 2\frac{\alpha\lambda_k}{1-\beta}\right)^2}(1 + O(\lambda_k) + O(\varepsilon)) \tag{154}$$

Let's make a couple of observations. From asymptotic expressions above one can show that

$$\widetilde{U}(1-\varepsilon) \sim \sum_{k=1}^{\infty} \frac{\lambda_k^2}{\varepsilon + \lambda_k} \sim \sum_{k=1}^{\varepsilon^{-\frac{1}{\nu}}} \lambda_k \sim 1 \tag{155}$$

$$\widetilde{U}'(1-\varepsilon) \sim \sum_{k=1}^{\infty} \frac{\lambda_k^2}{(\varepsilon + \lambda_k)^2} \sim \sum_{k=1}^{\varepsilon^{-\frac{1}{\nu}}} 1 \sim \varepsilon^{-\frac{1}{\nu}} \tag{156}$$

$$\widetilde{V}(1-\varepsilon) \sim \sum_{k=1}^{\infty} \frac{\lambda_k C_{kk}}{\varepsilon + \lambda_k} \sim \sum_{k=1}^{\varepsilon^{-\frac{1}{\nu}}} C_{kk} \sim \begin{cases} 1, & \zeta > 1 \\ \varepsilon^{\zeta-1}, & \zeta < 1 \end{cases} \tag{157}$$

$$\widetilde{V}'(1-\varepsilon) \sim \sum_{k=1}^{\infty} \frac{\lambda_k C_{kk}}{(\varepsilon + \lambda_k)^2} \sim \sum_{k=1}^{\varepsilon^{-\frac{1}{\nu}}} \frac{C_{kk}}{\lambda_k} \sim \begin{cases} 1, & \zeta > 2 \\ \varepsilon^{\zeta-2}, & \zeta < 2 \end{cases} \tag{158}$$

Here the sign $\sim$ means an asymptotic ($\varepsilon \to 0$) equality up to a multiplicative constant. Using these asymptotic relations we see that for $\widetilde{L}'(1-\varepsilon)$ leading asymptotic terms are given by either $\widetilde{U}'(1-\varepsilon)$ when $\zeta > 2 - \frac{1}{\nu}$, or by $\widetilde{V}'(1-\varepsilon)$ when $\zeta < 2 - \frac{1}{\nu}$.

Now we set for simplicity $\Lambda = K = 1$ and apply lemma 2 to (153) and (154) to get more refined asymptotic expressions. For $V'(1-\varepsilon)$ we assume $\zeta < 2$ because $V'(1-\varepsilon) = O(1)$ for $\zeta > 2$ and refined asymptotic is not needed.

$$\begin{aligned}
\widetilde{V}'(1-\varepsilon) &= \sum_{k=1}^{\infty} \frac{\lambda_k C_{kk}(1 + O(\lambda_k) + O(\varepsilon))}{\left(\varepsilon + 2\frac{\alpha\lambda_k}{1-\beta}\right)^2} \\
&= (1 + O(\varepsilon)) \sum_{k=1}^{\infty} \frac{\lambda_k C_{kk}}{\left(\varepsilon + 2\frac{\alpha\lambda_k}{1-\beta}\right)^2} + \sum_{k=1}^{\infty} \frac{O(C_{kk}\lambda_k^2)}{\left(\varepsilon + 2\frac{\alpha\lambda_k}{1-\beta}\right)^2} \\
&\overset{(1)}{=} \left(\frac{2\alpha}{1-\beta}\right)^{-2} \sum_{k=1}^{\infty} \frac{C_{kk}\lambda_k}{\left(\frac{\varepsilon(1-\beta)}{2\alpha} + \lambda_k\right)^2} + O(\varepsilon^{\min(0,\zeta-1)}) \\
&\overset{(1)}{=} \Gamma(\zeta+1)\Gamma(2-\zeta)\left(\frac{2\alpha}{1-\beta}\right)^{-\zeta} \varepsilon^{\zeta-2}(1 + o(1))
\end{aligned} \tag{159}$$

Here in (1) we first used first part of lemma 2 to get $\sum_{l \geq k} C_{ll}\lambda_l^2 = \frac{\zeta}{\zeta+1}k^{-\varkappa-\nu}(1 + o(1))$. Then for $1 < \zeta < 2$ the second sum is finite (see (158)) and for $\zeta < 1$ second part of lemma 2 allows to bound the sum as $O(\varepsilon^{\zeta-1})$. In (2) we applied second part of lemma 2 to the first sum in (1).

For $U'(1 - \varepsilon)$ we have

$$\widetilde{U}'(1-\varepsilon) = \gamma \sum_{k=1}^{\infty} \frac{\left(\frac{\alpha\lambda_k}{1-\beta}\right)^2 (1 + O(\lambda_k) + O(\varepsilon))}{\left(\varepsilon + 2\frac{\alpha\lambda_k}{1-\beta}\right)^2}$$

$$\stackrel{(1)}{=} \frac{\gamma}{4}(1 + O(\varepsilon)) \sum_{k=1}^{\infty} \frac{\lambda_k^2}{\left(\frac{\varepsilon(1-\beta)}{2\alpha} + \lambda_k\right)^2} + \frac{\gamma}{4} \sum_{k=1}^{\infty} \frac{O(\lambda_k^3)}{\left(\frac{\varepsilon(1-\beta)}{2\alpha} + \lambda_k\right)^2} \quad (160)$$

$$\stackrel{(2)}{=} \frac{\gamma}{4\nu}\Gamma(2 - \frac{1}{\nu})\Gamma(\frac{1}{\nu})\left(\frac{2\alpha}{1-\beta}\right)^{\frac{1}{\nu}} \varepsilon^{-\frac{1}{\nu}}(1 + o(1))$$

In (1) the second sum can be bounded as $O(1)$ after recalling that $\nu > 1$. In (2) we again applied second part of lemma 2 with $C_k = \lambda_k^2$ and $n = 2$ since $\sum_{l\geq k} \lambda_k^2 = \frac{1}{2\nu-1}k^{-2\nu+1}(1 + o(1))$. To get the coefficient in (2) we used $\frac{\Gamma(3-1/\nu)}{2\nu-1} = \frac{\Gamma(2-1/\nu)}{\nu}$.

Substituting (160) and (159) into (150) we get

$$\widetilde{L}'(1-\varepsilon) = \frac{C_{V'}}{2(1 - \widetilde{U}(1))}\varepsilon^{-\max(0,2-\zeta)}(1 + o(1)) + \frac{\widetilde{V}(1)C_{U'}}{2(1 - \widetilde{U}(1))^2}\varepsilon^{-\frac{1}{\nu}}(1 + o(1)) \quad (161)$$

where $C_{V'}$ and $C_{U'}$ are constants in the asymptotics (159) and (160). Picking the leading term depending on the sign of $\zeta - (2 - \frac{1}{\nu})$ we obtain final expressions

$$\widetilde{L}'(1-\varepsilon) = \begin{cases} \frac{\Gamma(\zeta+1)\Gamma(2-\zeta)}{2(1-\widetilde{U}(1))}\left(\frac{2\alpha}{1-\beta}\right)^{-\zeta}\varepsilon^{\zeta-2}(1 + o(1)), & \zeta < 2 - \frac{1}{\nu} \\ \gamma\frac{\widetilde{V}(1)\Gamma(2-\frac{1}{\nu})\Gamma(\frac{1}{\nu})}{8\nu(1-\widetilde{U}(1))^2}\left(\frac{2\alpha}{1-\beta}\right)^{\frac{1}{\nu}}\varepsilon^{-\frac{1}{\nu}}(1 + o(1)), & \zeta < 2 - \frac{1}{\nu} \end{cases} \quad (162)$$

The last step is to apply Tauberian theorem (Feller (1971), p. 445) which states that if generating function $\widetilde{F}(z)$ of a sequence $F(t)$ has asymptotic $\widetilde{F}(1-\varepsilon) = C\varepsilon^{-\rho}(1+o(1))$, then the partial sums have the asymptotic $\sum_{t=1}^{T} F(t) = \frac{C}{\Gamma(\rho+1)}T^{\rho}(1+o(1))$. Applying this theorem to $F(t) = tL(t)$ and its generating function given by (162) we get precisely (26) with $\Lambda = K = 1$. To restore the values of constants $\Lambda, K$ recall that the part of the loss coming from signal ($\widetilde{V}(z)$) is simply proportional to $K$, and $\Lambda$ always goes in combination $\alpha\Lambda$. $\qquad\square$

Finally, let us show how (26) can be used to obtain asymptotics (28a),(28b) for the loss values $L(t)$ themselves. Assuming that $L(t) = Ct^{-\xi}, \xi < 2$, we get

$$\sum_{t=1}^{T} tL(t) = C \sum_{t=1}^{T} t^{-\xi+1}(1 + o(1)) \approx C \int_{1}^{T} t^{-\xi+1}(1 + o(1)) = \frac{C}{2 - \xi}T^{2-\xi}(1 + o(1)) \quad (163)$$

Comparing this with (26), we find the constants $C$ appearing in (28a),(28b).

### E.4 EXPONENTIAL DIVERGENCE OF THE LOSS AND CONVERGENCE-DIVERGENCE TRANSITION

**Convergence radius $r_L$ and exponential divergence rate.** By Proposition 3, the convergence radius $r_L < 1$ if and only if $\widetilde{U}(1) > 1$, and in the case $\widetilde{U}(1) > 1$ it is determined by the condition

$$1 = r_l\widetilde{U}(r_L). \quad (164)$$

If $r_L < 1$, loss exponentially diverges as $t \to \infty$:

**Proposition 10.** *Under assumption of Proposition 3, suppose that $r_L < 1$. Then*

$$\sum_{t=0}^{T} L(t)r_L^t = \frac{\widetilde{V}(r_L)}{2(1 + r_L^2\widetilde{U}'(r_L))}T(1 + o(1)), \quad (T \to \infty). \quad (165)$$

*Proof.* Consider the modified generating function $\widetilde{L}_1(z) = \widetilde{L}(r_L z)$, which is the generating function for the sequence $L(t)r_L^t$ :

$$\widetilde{L}_1(z) = \sum_{t=0}^{\infty} z^t L(t) r_L^t. \tag{166}$$

The convergence radius for $\widetilde{L}_1$ is 1. We can also write

$$\widetilde{L}_1(1-\epsilon) = \frac{\widetilde{V}(r_L(1-\epsilon))}{2(1 - r_L(1-\epsilon)\widetilde{U}(r_L(1-\epsilon)))} \tag{167}$$

$$= \frac{\widetilde{V}(r_L)}{2(1 + r_L^2 \widetilde{U}'(r_L))\epsilon}(1 + o(1)), \quad (\epsilon \to 0). \tag{168}$$

The statement of the proposition then follows by the Tauberian theorem. $\qquad\square$

If we assume that

$$L(t) \sim C r_L^{-t} \tag{169}$$

with some constant $C$, then this constant is fixed by Eq. (165):

$$L(t) \sim \frac{\widetilde{V}(r_L)}{2(1 + r_L^2 \widetilde{U}'(r_L))} r_L^{-t}. \tag{170}$$

**"Eventual divergence" phase with small $\alpha$.** Recall that we characterized the "eventual divergence" phase by the condition $\widetilde{U}(1) = \infty$. In this phase we necessarily have $r_L < 1$. However, $r_L$ can be very close to 1 (and thus "divergence delayed") – for example if the learning rate $\alpha$ is very small. In this scenario we can derive an asymptotic expression for the difference $\epsilon_* = 1 - r_L$.

In the sequel, we assume for simplicity that $\beta = 0$ and $\tau = \gamma = 1$, and that the eigenvalues and initial covariance coefficients $C_{kk,0}$ satisfy power law relations (25) with $\frac{1}{2} < \nu < 1$ (corresponding to the "eventual divergence" phase).

**Proposition 11.** *As $\alpha \to 0$,*

$$\epsilon_* \equiv 1 - r_L = \left(\frac{1}{4\nu}\Gamma(2 - 1/\nu)\Gamma(1/\nu - 1)\right)^{\nu/(1-\nu)} (2\alpha\Lambda)^{1/(1-\nu)}(1 + o(1)). \tag{171}$$

*Proof.* With $\beta = 0$ and $\tau = \gamma = 1$, we have by Eq. (18)

$$\widetilde{U}(1 - \epsilon) = \alpha^2 \sum_k \frac{\lambda_k^2}{2\alpha\lambda_k(1-\epsilon) + \epsilon} \tag{172}$$

$$= \frac{\alpha}{2\Lambda} \sum_k \frac{\lambda_k^2/(1 - 2\alpha\lambda_k)}{\lambda_k/(\Lambda(1 - 2\alpha\lambda_k)) + \epsilon/(2\alpha\Lambda)}. \tag{173}$$

For any $\alpha$, we can now apply Lemma 2(2) with $\varkappa = 2\nu - 1$, $\zeta = 2 - 1/\nu$ and $n = 1$ and get

$$\widetilde{U}(1 - \epsilon) = \frac{\alpha\Lambda}{2(2\nu - 1)}\Gamma(3 - 1/\nu)\Gamma(1/\nu - 1)\left(\frac{\epsilon}{2\alpha\Lambda}\right)^{1-1/\nu}(1 + o(1)) \tag{174}$$

$$= \frac{1}{4\nu}\Gamma(2 - 1/\nu)\Gamma(1/\nu - 1)(2\alpha\Lambda)^{1/\nu}\epsilon^{1-1/\nu}(1 + o(1)), \quad (\epsilon \to 0). \tag{175}$$

It is easy to see that $o(1)$ here is uniform for small $\alpha$. The desired $\epsilon_* = \epsilon_*(\alpha)$ is determined from the equation $1 = (1 - \epsilon_*)\widetilde{U}(1 - \epsilon_*)$. As $\alpha \to 0$, we clearly have $\epsilon_* = \epsilon_*(\alpha) \to 0$. It follows that

$$1 = \frac{1}{4\nu}\Gamma(2 - 1/\nu)\Gamma(1/\nu - 1)(2\alpha\Lambda)^{1/\nu}\epsilon_*(\alpha)^{1-1/\nu}(1 + o(1)), \quad (\alpha \to 0). \tag{176}$$

This implies desired expression (171). $\qquad\square$

We can similarly use Lemma 2 to also estimate the derivative $\widetilde{U}'(1 - \epsilon)$ :

$$\widetilde{U}'(1 - \epsilon) = \alpha^2 \sum_k \frac{\lambda_k^2 (1 - 2\alpha\lambda_k)}{(2\alpha\lambda_k(1 - \epsilon_*) + \epsilon)^2} \tag{177}$$

$$= \frac{1}{4\Lambda^2} \sum_k \frac{\lambda_k^2/(1 - 2\alpha\lambda_k)}{(\lambda_k/(\Lambda(1 - 2\alpha\lambda_k)) + \epsilon/(2\alpha\Lambda))^2} \tag{178}$$

$$= \frac{1}{4(2\nu - 1)}\Gamma(3 - 1/\nu)\Gamma(1/\nu)\left(\frac{\epsilon}{2\alpha\Lambda}\right)^{-1/\nu}(1 + o(1)) \tag{179}$$

$$= \frac{1}{4\nu}\Gamma(2 - 1/\nu)\Gamma(1/\nu)(2\alpha\Lambda)^{1/\nu}\epsilon^{-1/\nu}(1 + o(1)) \tag{180}$$

$$= \frac{1/\nu - 1}{\epsilon}\widetilde{U}(1 - \epsilon)(1 + o(1)), \quad (\epsilon \to 0) \tag{181}$$

(this result can also formally be obtained by differentiating Eq. (174) in $\epsilon$). In particular, for $\epsilon = \epsilon_*$ we get

$$\widetilde{U}'(1 - \epsilon_*) = \frac{1/\nu - 1}{\epsilon_* r_L}(1 + o(1)), \quad (\alpha \to 0). \tag{182}$$

**"Early convergent" regime.** If $r_L < 1$, then, as discussed above, loss eventually diverges exponentially. However, in general, we expect this divergence to occur only after some interval of convergence (see Figure 3(right)). We will give the loss asymptotic for this "early convergent" phase and give a rough estimate of the time scale at which the transition between the two regimes occurs. We retain the assumptions $\beta = 0$ and $\gamma = \tau = 1$ from the previous paragraph, and assume power laws (25) with $\frac{1}{2} < \nu < 1$ and $\zeta < 1$.

We start with finding the loss asymptotic in the early convergent phase. First we argue that this phase can be approximately described by the loss generating function

$$\widetilde{L}(z) \approx \frac{1}{2}\widetilde{V}(z), \tag{183}$$

i.e. generating function (16) in which we approximate the denominator by 1:

$$1 - z\widetilde{U}(z) \approx 1. \tag{184}$$

This approximation can be justified for $z < r_L$ by noting that, by Eq. (174),

$$\widetilde{U}(1 - \epsilon) \approx \left(\frac{\epsilon}{\epsilon_*}\right)^{1 - 1/\nu}\widetilde{U}(1 - \epsilon_*) = \left(\frac{\epsilon}{\epsilon_*}\right)^{1 - 1/\nu} \tag{185}$$

with $1 - 1/\nu < 0$. We remark that approximations (184) and (183) essentially mean that we ignore terms containing factors $U_{t,s}$ associated with noise in expansion (12).

If relation (183) held for all $z \in (0, 1)$, then we would have an asymptotic convergence of the loss for all $t$. Indeed, under our assumptions,

$$\widetilde{V}(1 - \epsilon) = \sum_{k=1}^{\infty} \frac{\lambda_k C_{kk,0}}{2\alpha\lambda_k(1 - \epsilon) + \epsilon} \tag{186}$$

and $\widetilde{V}(1 - \epsilon) \to \infty$ as $\epsilon \to 0$ (since $\zeta < 1$). Applying again Lemma 2, we find

$$\widetilde{V}(1 - \epsilon) = \frac{1}{2\alpha\Lambda} \sum_k \frac{\lambda_k C_{kk,0}/(1 - 2\alpha\lambda_k)}{\lambda_k/(\Lambda(1 - 2\alpha\lambda_k)) + \epsilon/(2\alpha\Lambda)} \tag{187}$$

$$= \frac{K}{2\alpha\Lambda}\Gamma(\zeta + 1)\Gamma(1 - \zeta)\left(\frac{\epsilon}{2\alpha\Lambda}\right)^{\zeta - 1}(1 + o(1)) \tag{188}$$

$$= \frac{K}{2}\Gamma(\zeta + 1)\Gamma(1 - \zeta)(2\alpha\Lambda)^{-\zeta}\epsilon^{\zeta - 1}(1 + o(1)), \quad (\epsilon \to 0). \tag{189}$$

The Tauberian theorem then implies that

$$\sum_{t=0}^{T} L(t) = \frac{K}{2\Gamma(2-\zeta)}\Gamma(\zeta+1)\Gamma(1-\zeta)(2\alpha\Lambda)^{-\zeta}T^{1-\zeta}(1+o(1)) \tag{190}$$

$$= \frac{K}{2(1-\zeta)}\Gamma(\zeta+1)(2\alpha\Lambda)^{-\zeta}T^{1-\zeta}(1+o(1)), \quad (\epsilon \to 0). \tag{191}$$

Assuming a power-law form

$$L(t) \sim Ct^{-\zeta}, \tag{192}$$

we then find

$$L(t) \sim \frac{K}{2}\Gamma(\zeta+1)(2\alpha\Lambda)^{-\zeta}t^{-\zeta}. \tag{193}$$

This expression agrees with "signal-dominated" loss asymptotic (28a) up to the factor $(1-\widetilde{U}(1))^{-1}$ (missing because of our approximation (184)).

**Transition from convergence to divergence.** Of course, the obtained asymptotic (193) does not hold for very large $t$, since representation (183) breaks down at $z \gtrsim r_L$, and the loss trajectory switches to exponential divergence. We can roughly estimate the switching time $t_{\text{blowup}}$ by equating the asymptotic expressions for $L(t)$ in the convergent and divergent phases:

$$\frac{K}{2}\Gamma(\zeta+1)(2\alpha\Lambda)^{-\zeta}t_{\text{blowup}}^{-\zeta} \approx \frac{\widetilde{V}(r_L)}{2(1+r_L^2\widetilde{U}'(r_L))}r_L^{-t_{\text{blowup}}}. \tag{194}$$

This is a transcendental equation not explicitly solvable in $t_{\text{blowup}}$. We simplify it using a number of approximations. Using approximation (189) for $\widetilde{V}(r_L)$ and the fact that $\epsilon_*$ is small, we can simplify this equation to

$$\frac{1+r_L^2\widetilde{U}'(1-\epsilon_*)}{\Gamma(1-\zeta)}\epsilon_*^{1-\zeta}t_{\text{blowup}}^{-\zeta} \approx (1-\epsilon_*)^{-t_{\text{blowup}}} \approx e^{\epsilon_*t_{\text{blowup}}}. \tag{195}$$

Next, using Eq. (182), we can approximate the l.h.s. by

$$\frac{1+r_L^2\widetilde{U}'(1-\epsilon_*)}{\Gamma(1-\zeta)}\epsilon_*^{1-\zeta}t_{\text{blowup}}^{-\zeta} \approx \frac{1/\nu-1}{\Gamma(1-\zeta)}\epsilon_*^{-\zeta}t_{\text{blowup}}^{-\zeta}. \tag{196}$$

It follows that Eq. (195) can be approximated as

$$\frac{1/\nu-1}{\Gamma(1-\zeta)}(\epsilon_*t_{\text{blowup}})^{-\zeta} = e^{\epsilon_*t_{\text{blowup}}}. \tag{197}$$

Thus, we can write

$$t_{\text{blowup}} \approx \frac{a_*}{\epsilon_*}, \tag{198}$$

where $a_*$ is the solution of the equation

$$\frac{1/\nu-1}{\Gamma(1-\zeta)}a_*^{-\zeta} = e^{a_*}. \tag{199}$$

If $t_{\text{div}} = -1/\log r_L$ denotes the characteristic time scale of exponential divergence, then at small $\alpha$ we have $t_{\text{div}} \approx 1/\epsilon_*$, so the characteristic transition time $t_{\text{blowup}}$ is related to $t_{\text{div}}$ by the simple relation

$$t_{\text{blowup}} \approx a_*t_{\text{div}}. \tag{200}$$

## E.5 BUDGET OPTIMIZATION

It is known that for SGD with not too large batches, there is approximately linear scalability of learning w.r.t. batch size, in the sense that to reach a given loss value with an increased batch size the number of iterations should be decreased proportionally (Ma et al., 2018; Shallue et al., 2018). This law is naturally no longer valid at very large batch sizes, since in the limit of infinitely large batch size SGD just converges to the non-stochastic GD, so that further increasing batch size when

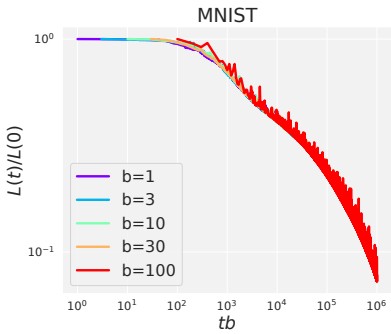

Figure 5: Representation of loss $L(t)$ obtained using different batch sizes $b$ as (approximately) a function of the computation budged $bt$ (regardless of how it is factorized over $b$ and $t$) for MNIST. Momentum is fixed at $\beta = 0$ and the ratio $\alpha/b$ is kept the same for all curves.

it it already large brings little benefit (the effect of "diminishing returns"). Figure 5 shows that this effect is also reproduced in our setting.

Let us now show that this effect should be observed in the signal-dominated phase if for each batch size $b$, learning rate and momentum are chosen optimally. In the respective part of Sec. 6 we showed that loss can be approximately written as $L(t) \sim (\alpha_{\mathrm{eff}} t)^{-\zeta}$; the optimal learning rate and momenta are located near the convergence boundary and the momenta is close to 1. Since the effective learning rate is bounded $\alpha_{\mathrm{eff}} < \frac{2}{\gamma \lambda_{\mathrm{crit}}}$ (see eq. (22)) and the bound is tight for the $beta$ close to 1 (see Eq. (23)), we in the optimal loss value we can approximate $\alpha_{\mathrm{eff}}$ with $\frac{2}{\gamma \lambda_{\mathrm{crit}}}$. This leads us to the optimal loss $L(t) \approx C'(t/\gamma)^{-\xi}$. Replacing $\gamma$ with $\frac{1}{b}$ (which is very accurate for large dataset sizes $N$, see prop. 1), we get $L(t) \approx C'(bt)^{-\xi}$ which is exactly linear scalability of SGD with batch size.

### E.6 Positive vs. negative momenta

Recall that our analysis of the effect of momentum is based on the approximations (28a), (28b) for $L(t)$ at large $t$:

$$L_{\mathrm{approx}}(t) = \begin{cases} \frac{K\Gamma(\zeta+1)}{2(1-\widetilde{U}(1))}\left(\frac{2\alpha\Lambda}{1-\beta}\right)^{-\zeta}t^{-\zeta}, & \zeta < 2 - 1/\nu, \quad (201a) \\ \frac{\gamma\widetilde{V}(1)\Gamma(2-1/\nu)}{8\nu(1-\widetilde{U}(1))^2}\left(\frac{2\alpha\Lambda}{1-\beta}\right)^{1/\nu}t^{1/\nu-2}, & \zeta > 2 - 1/\nu. \quad (201b) \end{cases}$$

We now formulate and prove the full version of Proposition 5 describing how $L_{\mathrm{approx}}(t)$ is affected by changing the momentum from $\beta = 0$ at the optimal $\alpha$. Note that $t$ enters expression $L_{\mathrm{approx}}(t)$ only through the factor $t^{-\zeta}$ or $t^{1/\nu-2}$. As a result, the statements below hold in the same form for all $t > 0$.

**Proposition 12** (full version of Proposition 5). *Suppose that the eigenvalues $\lambda_k$ and initial moments $C_{kk,0}$ are subject to large-$k$ power law asymptotics (25) with some exponents $\nu, \varkappa > 0, \zeta = \varkappa/\nu$ and coefficients $\Lambda, K$. Denote $\alpha_{\mathrm{opt}} = \arg\min_\alpha L_{\mathrm{approx}}(t)|_{\beta=0}$.*

*I. (signal-dominated phase) Let $\zeta < 2 - 1/\nu$ and $0 < \tau \leq 1, 0 \leq \gamma \leq 1$. Then*

1. *At $\beta = 0$ and as a function of $\alpha$, the function $L_{\mathrm{approx}}(t)$ has a unique minimum in the interval $(0, \alpha_{\max})$, where $\alpha_{\max}$ is the critical stability value defined by the condition $\widetilde{U}(z = 1, \alpha = \alpha_{\max}, \beta = 0) = 1$. Accordingly, $\alpha_{\mathrm{opt}} = \arg\min_\alpha L_{\mathrm{approx}}(t)|_{\beta=0}$ is well-defined and unique.*

2. *$\partial_\beta L_{\mathrm{approx}}(t)|_{\beta=0,\alpha=\alpha_{\mathrm{opt}}} < 0$.*

*II. (noise-dominated phase) Let $\zeta > 2 - 1/\nu$ and Let $\tau = \gamma = 1$. Then*

1. *$\alpha_{\mathrm{opt}} = \frac{2(\nu-1)}{3\nu-1}(\mathrm{Tr}[\mathbf{H}])^{-1}$;*

2. *The sign of $\partial_\beta L_{\mathrm{approx}}(t)|_{\beta=0,\alpha=\alpha_{\mathrm{opt}}}$ equals the sign of the expression*

$$\Xi = \nu \operatorname{Tr}[\mathbf{H}] \operatorname{Tr}[\mathbf{H}\mathbf{C}_0] - (\nu - 1) \operatorname{Tr}[\mathbf{H}^2] \operatorname{Tr}[\mathbf{C}_0]. \tag{202}$$

Before giving the proof, we make a few remarks.

1. The "noise-diminated" part of our result is proved under much more restrictive assumptions ($\tau = \gamma = 1$) than the "signal-dominated" part ($0 < \tau \le 1, 0 \le \gamma \le 1$). The reason is that the effect of $\beta$ is much more subtle in the noise-dominated case, and is described by relatively cumbersome formulas. Setting $\tau = \gamma = 1$ allows us to substantially simplify these formulas. In contrast, improvement from positive $\beta$ in the signal-dominated case appears to be a general and robust phenomenon.

2. It can be observed from the proof in the noise-dominated case that the second, negative term in Eq. (202) results from the "signal" generating function $\widetilde{U}(1)$ in Eq. (201b), while the first, positive term results from the "noise" generating function $\widetilde{V}(1)$. A similar observation can be made in the signal-dominated case, but Eq. (201a) only includes the "signal" generating function $\widetilde{U}(1)$, so the counterpart of expression $\Xi$ in this case is always negative.

3. If $\tau = \gamma = 1$, then in the signal-dominated case $\alpha_{\mathrm{opt}}$ has the simple form $\alpha_{\mathrm{opt}} = \frac{2\zeta}{\zeta+1}(\operatorname{Tr}[\mathbf{H}])^{-1}$.

4. If $\tau = \gamma = 1$, then the critical stability value $\alpha_{\mathrm{max}}$ at which $\widetilde{U}(z = 1, \alpha, \beta = 0) = 1$ has the simple form $\alpha_{\mathrm{max}} = 2(\operatorname{Tr}[\mathbf{H}])^{-1}$. In particular, we see from the explicit formulas for $\alpha_{\mathrm{opt}}$ that in both signal-dominated and noise-dominated regime $\alpha_{\mathrm{opt}} < \alpha_{\mathrm{max}}$, as expected.

*Proof of Proposition 12.*

Let $\widetilde{U}_1, \widetilde{V}_1$ denote the values $\widetilde{U}(z = 1), \widetilde{V}(z = 1)$. Recall from Eqs. (18), (20) that

$$\widetilde{U}_1(\alpha, \beta) = \sum_k \frac{\gamma\alpha\lambda_k(\beta + 1)}{\alpha\beta\tau\gamma\lambda_k + \alpha\beta\lambda_k + \alpha\tau\gamma\lambda_k - \alpha\lambda_k - 2\beta^2 + 2}, \tag{203}$$

$$\widetilde{V}_1(\alpha, \beta) = \sum_k \frac{C_{kk,0}(2\alpha\beta\lambda_k + \beta^3 - \beta^2 - \beta + 1)}{\alpha(\alpha\beta\tau\gamma\lambda_k + \alpha\beta\lambda_k + \alpha\tau\gamma\lambda_k - \alpha\lambda_k - 2\beta^2 + 2)}. \tag{204}$$

As a preliminary step, note that at $\beta = 0$ we have:

$$\widetilde{U}_1(\alpha, 0) = \sum_k \frac{\gamma\alpha\lambda_k}{\alpha\lambda_k(\tau\gamma - 1) + 2}, \tag{205}$$

$$\partial_\alpha \widetilde{U}_1(\alpha, 0) = \sum_k \frac{\gamma\lambda_k}{\alpha\lambda_k(\tau\gamma - 1) + 2} - \frac{\gamma\alpha\lambda_k \cdot \lambda_k(\tau\gamma - 1)}{(\alpha\lambda_k(\tau\gamma - 1) + 2)^2} \tag{206}$$

$$= \sum_k \frac{2\gamma\lambda_k}{(\alpha\lambda_k(\tau\gamma - 1) + 2)^2}, \tag{207}$$

$$\partial_\beta \widetilde{U}_1(\alpha, 0) = \sum_k \frac{\gamma\alpha\lambda_k}{\alpha\lambda_k(\tau\gamma - 1) + 2} - \frac{\gamma\alpha\lambda_k \cdot \alpha\lambda_k(\tau\gamma + 1)}{(\alpha\lambda_k(\tau\gamma - 1) + 2)^2} \tag{208}$$

$$= \sum_k \frac{-2\gamma\alpha^2\lambda_k^2 + 2\gamma\alpha\lambda_k}{(\alpha\lambda_k(\tau\gamma - 1) + 2)^2} \tag{209}$$

$$= \alpha\partial_\alpha \widetilde{U}_1(\alpha, 0) - \sum_k \frac{2\gamma\alpha^2\lambda_k^2}{(\alpha\lambda_k(\tau\gamma - 1) + 2)^2}. \tag{210}$$

By our assumptions, $\tau\gamma - 1 \le 0$. As a result, the function $\widetilde{U}_1(\alpha, 0)$ is monotone increasing in $\alpha$, and there is a unique critical stability value $\alpha_{\mathrm{max}}$ at which $\widetilde{U}_1(\alpha_{\mathrm{max}}, 0) = 1$ and $L_{\mathrm{approx}}(t)$ becomes infinite by Eq. (201).

I. (**signal-dominated phase**)

1. It is convenient to consider the logarithmic derivative of $L_{\text{approx}}(t)|_{\beta=0}$. By Eq. (201a),

$$\partial_\alpha \ln L_{\text{approx}}(t)|_{\beta=0} \tag{211}$$

$$= \alpha^\zeta \partial_\alpha \alpha^{-\zeta} + (1 - \widetilde{U}_1(\alpha, 0))^{-1} \partial_\alpha \widetilde{U}_1(\alpha, 0) \tag{212}$$

$$= \Big[ (1 - \sum_k \frac{\gamma \alpha \lambda_k}{\alpha \lambda_k (\tau\gamma - 1) + 2}) \frac{-\zeta}{\alpha} + \sum_k \frac{2\gamma \lambda_k}{(\alpha \lambda_k (\tau\gamma - 1) + 2)^2} \Big] \tag{213}$$

$$\times (1 - \widetilde{U}_1(\alpha, 0))^{-1} \tag{214}$$

$$= \Big[ \zeta \sum_k \frac{\gamma \alpha \lambda_k}{\alpha \lambda_k (\tau\gamma - 1) + 2} + \sum_k \frac{2\gamma \alpha \lambda_k}{(\alpha \lambda_k (\tau\gamma - 1) + 2)^2} - \zeta \Big] \tag{215}$$

$$\times \alpha^{-1} (1 - \widetilde{U}_1(\alpha, 0))^{-1}. \tag{216}$$

Note that the last expression in square brackets is monotone increasing in $\alpha$, because $\tau\gamma \leq 1$. If $\alpha = 0$, this expression is negative. On the other hand, at $\alpha = \alpha_{\max}$ such that $\widetilde{U}_1(\alpha_{\max}, 0) = 1$ this expression is positive. It follows that there is a unique point $\alpha_{\text{opt}} \in (0, \alpha_{\max})$ where

$$\partial_\alpha L_{\text{approx}}(t)|_{\alpha=\alpha_{\text{opt}}, \beta=0} = 0 \tag{217}$$

and $L_{\text{approx}}(t)|_{\beta=0}$ attains its minimum.

2. Now observe that

$$\partial_\beta \ln L_{\text{approx}}(t)|_{\beta=0} \tag{218}$$

$$= [(1 - \beta)^{-\zeta} \partial_\beta (1 - \beta)^\zeta + (1 - \widetilde{U}_1(\alpha, 0))^{-1} \partial_\beta \widetilde{U}_1(\alpha, 0)]|_{\beta=0} \tag{219}$$

$$= \Big[ (1 - \widetilde{U}_1(\alpha, 0))(-\zeta) + \alpha \partial_\alpha \widetilde{U}_1(\alpha, 0) - \sum_k \frac{2\gamma \alpha^2 \lambda_k^2}{(\alpha \lambda_k (\tau\gamma - 1) + 2)^2} \Big] \tag{220}$$

$$\times (1 - \widetilde{U}_1(\alpha, 0))^{-1} \tag{221}$$

$$= \alpha \partial_\alpha \ln L_{\text{approx}}(t) - \Big[ \sum_k \frac{2\gamma \alpha^2 \lambda_k^2}{(\alpha \lambda_k (\tau\gamma - 1) + 2)^2} \Big] (1 - \widetilde{U}_1(\alpha, 0))^{-1}. \tag{222}$$

At $\alpha = \alpha_{\max}$ the first term in Eq. (222) vanishes, and the remaining term is negative, proving that

$$\partial_\beta L_{\text{approx}}(t)|_{\beta=0, \alpha=\alpha_{\text{opt}}} < 0. \tag{223}$$

II. (**noise-dominated phase**) At $\tau = \gamma = 1$, formulas (205)-(210) admit a simpler form:

$$\widetilde{U}_1(\alpha, 0) = \frac{\alpha}{2} \sum_k \lambda_k = \frac{\alpha}{2} \text{Tr}[\mathbf{H}], \tag{224}$$

$$\partial_\alpha \widetilde{U}_1(\alpha, 0) = \frac{1}{2} \sum_k \lambda_k = \frac{1}{2} \text{Tr}[\mathbf{H}], \tag{225}$$

$$\partial_\beta \widetilde{U}_1(\alpha, 0) = \frac{\alpha}{2} \sum_k \lambda_k (1 - \alpha \lambda_k) \tag{226}$$

$$= \alpha \partial_\alpha \widetilde{U}_1(\alpha, 0) - \frac{\alpha^2}{2} \text{Tr}[\mathbf{H}^2]. \tag{227}$$

Snce Eq. (201b) for the noise-dominated phase involves $\widetilde{V}_1$, we will also need similar formulas for this expression. At $\beta = 0$ we have

$$\widetilde{V}_1(\alpha, 0) = \frac{1}{2\alpha} \sum_k C_{kk,0} = \frac{1}{2\alpha} \text{Tr}[\mathbf{C}_0], \tag{228}$$

$$\partial_\alpha \widetilde{V}_1(\alpha, 0) = -\frac{1}{2\alpha^2} \sum_k C_{kk,0} = -\frac{1}{2\alpha^2} \text{Tr}[\mathbf{C}_0], \tag{229}$$

$$\partial_\beta \widetilde{V}_1(\alpha, 0) = \sum_k \Big( \frac{C_{kk,0}(2\alpha \lambda_k - 1)}{2\alpha} - \frac{C_{kk,0} \lambda_k}{2} \Big) \tag{230}$$

$$= \frac{1}{2} \text{Tr}[\mathbf{H} \mathbf{C}_0] - \frac{1}{2\alpha} \text{Tr}[\mathbf{C}_0]. \tag{231}$$

1. Differentiating Eq. (201b),

$$\partial_\alpha \ln L_{\text{approx}}(t)|_{\beta=0} \tag{232}$$

$$= \alpha^{-1/\nu}\partial_\alpha\alpha^{1/\nu} + \widetilde{V}_1^{-1}(\alpha,0)\partial_\alpha\widetilde{V}_1(\alpha,0) + 2(1-\widetilde{U}_1(\alpha,0))^{-1}\partial_\alpha\widetilde{U}_1(\alpha,0) \tag{233}$$

$$= \frac{1}{\alpha\nu} - \frac{1}{\alpha} + 2\Big(1 - \frac{\alpha}{2}\operatorname{Tr}[\mathbf{H}]\Big)^{-1}\frac{1}{2}\operatorname{Tr}[\mathbf{H}] \tag{234}$$

$$= \Big[\Big(\frac{1}{\alpha\nu} - \frac{1}{\alpha}\Big)\Big(1 - \frac{\alpha}{2}\operatorname{Tr}[\mathbf{H}]\Big) + \operatorname{Tr}[\mathbf{H}]\Big]\Big(1 - \frac{\alpha}{2}\operatorname{Tr}[\mathbf{H}]\Big)^{-1} \tag{235}$$

$$= \Big[\Big(\frac{1}{\nu} - 1\Big) + \alpha\Big(\frac{3}{2} - \frac{1}{2\nu}\Big)\operatorname{Tr}[\mathbf{H}]\Big]\alpha^{-1}\Big(1 - \frac{\alpha}{2}\operatorname{Tr}[\mathbf{H}]\Big)^{-1}. \tag{236}$$

The expression in brackets is monotone increasing in $\alpha$ and vanishes at

$$\alpha_{\text{opt}} = \frac{2(\nu-1)}{3\nu-1}(\operatorname{Tr}[\mathbf{H}])^{-1}, \tag{237}$$

proving statement 1.

2. We have

$$\partial_\beta \ln L_{\text{approx}}(t)|_{\beta=0} \tag{238}$$

$$= \Big[(1-\beta)^{1/\nu}\partial_\beta(1-\beta)^{-1/\nu} + \widetilde{V}_1^{-1}(\alpha,0)\partial_\beta\widetilde{V}_1(\alpha,0) + 2(1-\widetilde{U}_1(\alpha,0))^{-1}\partial_\beta\widetilde{U}_1(\alpha,0)\Big]\Big|_{\beta=0}$$

$$= \frac{1}{\nu} + \alpha(\operatorname{Tr}[\mathbf{C}_0])^{-1}\operatorname{Tr}[\mathbf{HC}_0] - 1 + 2\Big(1 - \frac{\alpha}{2}\operatorname{Tr}[\mathbf{H}]\Big)^{-1}\Big(\frac{\alpha}{2}\operatorname{Tr}[\mathbf{H}] - \frac{\alpha^2}{2}\operatorname{Tr}[\mathbf{H}^2]\Big).$$

At $\alpha = \alpha_{\text{opt}}$ we can simplify this expression by recalling that at this $\alpha$ Eq. (234) vanishes, and using Eq. (237):

$$\partial_\beta \ln L_{\text{approx}}(t)|_{\beta=0,\alpha=\alpha_{\text{opt}}} \tag{239}$$

$$= \alpha_{\text{opt}}(\operatorname{Tr}[\mathbf{C}_0])^{-1}\operatorname{Tr}[\mathbf{HC}_0] + 2\Big(1 - \frac{\alpha_{\text{opt}}}{2}\operatorname{Tr}[\mathbf{H}]\Big)^{-1}\Big(-\frac{\alpha_{\text{opt}}^2}{2}\operatorname{Tr}[\mathbf{H}^2]\Big)$$

$$= \Big[\nu\operatorname{Tr}[\mathbf{H}]\operatorname{Tr}[\mathbf{HC}_0] - (\nu-1)\operatorname{Tr}[\mathbf{C}_0]\operatorname{Tr}[\mathbf{H}^2]\Big]\frac{\alpha_{\text{opt}}}{\nu}(\operatorname{Tr}[\mathbf{C}_0])^{-1}(\operatorname{Tr}[\mathbf{H}])^{-1},$$

proving statement 2. $\qquad\square$

In Figure 6 we illustrate this result for a range of models with exact (not asymptotic) power laws (25) at different $\nu$ and $\varkappa$. In this case, in the noise-dominated regime, characteristic $\Xi$ is positive for sufficiently large $\varkappa$ but negative for smaller $\varkappa$. In agreement with Proposition 12, we observe the experimentally found optimal $\beta$ to be negative or positive, respectively.

## F EXPERIMENTS

### F.1 TRAINING REGIMES

We have considered four regimes of training with the increasing level of approximation for real network dynamics:

1. **Real neural network (*NN*).**
   We used fully-connected neural network with one hidden layer in the NTK parametrization:

$$f(\mathbf{x}, \mathbf{w}) = \frac{1}{\sqrt{N}}\sum_{l=1}^{d}\mathbf{w}_l^{(2)}\operatorname{ReLU}(\mathbf{w}_l^{(1)} + b_l)$$

   Here $\mathbf{w}_l^{(1,2)}$ are the weight matrices, $b_l$ are the biases for neuron $l$, and $d$ is the number of units in the hidden layer. We have used $D = 1000$ in our experiments. For the sake of simplicity, the network outputs a single value for each input, i.e the classification task is treated as regression to the label with the MSE loss. The model is trained via SGD in the standard way: at each step a subset of size $b$ is sampled without repetition from the training dataset and the weights are updated according to the equation (1).

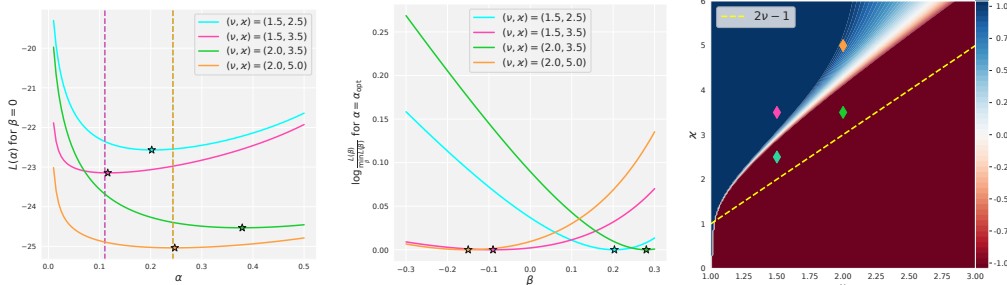

Figure 6: Theoretical and experimental study of optimal $\beta$. **Left**: Dependence of the loss on the learning rate $\alpha$ at $\beta = 0$, for different $\nu$ and $\varkappa$ (synthetic data) in the "noisy" phase. The stars show the experimentally determined optimal learning rates $\arg\min_\alpha L(\alpha)$, while the dashed vertical lines show the respective theoretical values $\alpha_{\text{opt}} = (\sum_k \lambda_k)^{-1} \frac{2(\nu-1)}{3\nu-1}$. The experimental and theoretical optima agree well for two pairs $(\nu, \varkappa)$ deep in the noisy phase, but not so much for $(\nu, \varkappa)$ near the phase boundary (cf. right figure). **Center**: Dependence of the loss log-ratio $\log \frac{L(\beta)}{\min_\beta L(\beta)}$ on $\beta$ at experimental $\alpha_{\text{opt}}$. The stars denote $\text{argmin}_\beta L(\beta)$. In two scenarios the minimum is attained at $\beta < 0$, and in the other two at $\beta > 0$. **Right**: The characteristic $\Xi$ given by (30) and characterizing improvement from $\beta < 0$ at large $t$ by Proposition 12, computed for the range $(\nu, \varkappa) \in [1,3] \times [0,6]$ of power-law models (25). Diamonds show the pairs $(\nu, \varkappa)$ in the experiments on the left and central figures. In agreement with our theoretical prediction, the characteristic $\Xi$ is positive in the scenarios where the optimal $\beta$ is negative, and vice versa. The dashed yellow line $\varkappa = 2\nu - 1$ separates the noise-dominated (top) and the signal-dominated (bottom) regimes.

2. **Linearized regime with sampled batches (*linearized* or *sampled*).**
   Model outputs $f(\mathbf{x})$ evolve on the dataset of interest according to the equation (47). At each step a batch of size $b$ is sampled without replacement from the whole dataset of size $N$ and SGD step is performed.

3. **All second matrix moments dynamics (*all sm*).**
   We simulate the dynamics of the whole second moment matrix that is obtained after taking expectation over all possible noise configurations, defined by (58) with the noise term given in the last line of (60).

4. **spectrally expressible regime (*SE*).**
   In the spectrally expressible regime, the diagonal of the second moment matrix is invariant under SGD dynamics, i.e diagonal elements evolve independently from off-diagonal terms. Therefore, one needs to keep track only of the diagonal terms. The dynamics in the output space follows Eq. (65).

## F.2 DATASETS

In the experiments we have used synthetic data with the NTK spectrum and second moment matrix obeying a power-law decay rule $\lambda_k = k^{-\nu}$ and $\lambda_k C_{kk,0} = k^{-\varkappa-1}$ with specific $\nu$ and $\varkappa$ and the subset of size $N$ of the MNIST dataset. The whole train dataset of size $60000$ requires much memory for the computation of NTK and is too costly from computational side for running large series of experiments, therefore we have used only part of the data of size $1000 - 10000$ depending on the experiment. The digits are distributed uniformly in the dataset, i.e there are $\frac{N}{10}$ samples of each class. We have observed that this subset of data is sufficient to approximate the power-law asymptotic for NTK and partial sums since the exponents do not change significantly with the increase of the amount of data.

## F.3 EXPERIMENTAL DETAILS

For Fig. 1 one we have taken a subset of $N = 3000$ digits from MNIST and ran $10^4$ SGD updates for the real neural network with one hidden layer, linearized dynamics, and in the spectrally expressible

appoximation with $\tau = \pm 1$. One can see that $\tau = 1$ provides more accurate approximation for smaller batch size $b = 10$, whereas for $b = 100$ both $\tau = \pm 1$ are rather accurate.

For Fig. 2 we ran 10000 gradient descent updates for every point on the grid of learning rates $\alpha$ and $\beta$, where we have taken 100 values of $\alpha$ uniformly selected on $[0, 4]$ and 50 values for $\beta$ uniformly on $[0, 1]$. The grey regions on the 2d plots correspond to the values of $\alpha$ and $\beta$ for which the optimization procedure has diverged. The size of the dataset is 1000 both in the synthetic and MNIST case.

For Figs 5, 6 (left, center and right) we have taken the dataset of size $N = 10000$ both for synthetic and MNIST case and performed 10000 steps of gradient descent. In Fig. 6 (center) we have taken 100 points uniformly on the interval $\alpha \in [0, 0.5]$, and 100 points uniformly on the interval $\beta \in [-0.3, 0.3]$ for Fig. 6 (right). Concerning the correspondence of optimal learning rates 6 with the analytical formula (30) we observe that is quite accurate for large ratio $\kappa / \nu$ since the ratio $\frac{C_{\text{noise}}}{C_{\text{signal}}} \simeq 100$ for $(\nu, \varkappa) = (1.5, 3.5), (2.0, 5.0)$, whereas for $\frac{C_{\text{noise}}}{C_{\text{signal}}} \simeq 1$ for $(\nu, \varkappa) = (1.5, 2.5), (2.0, 3.5)$, therefore both the signal and noise asymptotic are of same order for quite a long time, since the difference between the 'signal' exponent $\xi$ and 'noisy' $2 - \frac{1}{\nu}$ is relatively small.

### F.4 VALIDITY OF SPECTRALLY EXPRESSIBLE APPROXIMATION

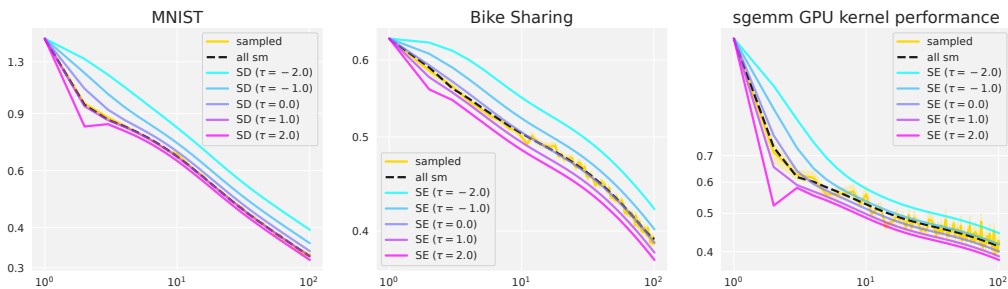

Figure 7: Loss trajectories for linearized stochastic loss dynamics (*solid yellow*), exact simulation of the loss dynamics with full second matrix (*dashed black*) (55), and the loss trajectories for several values of $\tau$ (shown with different colors). Linearized dynamics with stochastic sampling of batches is averaged over 1000 runs.

In section 3 we introduced several scenarios where SE approximation is exact. However, more interesting and at the same time much more difficult question is how well SE approximation well suited to describe the practical problems, where we are given a particular Jacobian $\mathbf{\Psi}$, initial state $\mathbf{C}_0$ and then generate the sequence of states $\mathbf{C}_t$ according to (4) with true noise term (5).

We can approach this question in a different ways. First approach is to consider individually the true dynamics with (5) and approximated dynamics with noise term (7). Then we can look at certain quantities of practical interest and check whether they are close to each other. In figure 2 we compared stability regions in $(\alpha, \beta)$ plane for MNIST and found a good agreement between actual dynamics and SE approximation with $\tau_1 = 1, \tau_2 = 1$. In figure 1 we did it for the loss values on MNIST, but instead of noise-averaged loss values we compared with loss on a single stochastic trajectory. To make the comparison for loss values more thorough, in figure F.4 we add average of several stochastic trajectories (*sampled*), true second moments dynamics (all sm), and several SE approximations with $\tau_1 = 1$ but different $\tau_2$. For MNIST we find excellent agreement with $\tau_2 = 1$, for Bike Sharing dataset agreement is good with $\tau_2 = 0$, and for GPU performance dataset the agreement is worse with optimal $\tau_2$ changing during optimization from 0 to $-1$.

The second approach is to run true dynamics of second moments, and on each step compare how true noise covariance $\mathbf{\Sigma}_t$ given by (5) with SE noise covariance $\mathbf{\Sigma}_t^{SE}(\tau_1, \tau_2)$, which we take to be given by

$$\mathbf{\Sigma}_t^{SE}(\tau_1, \tau_2) = \tau_1 \mathbf{H} \operatorname{Tr}[\mathbf{H}\mathbf{C}_t] - \tau_2 \mathbf{H}\mathbf{C}_t\mathbf{H} \tag{240}$$

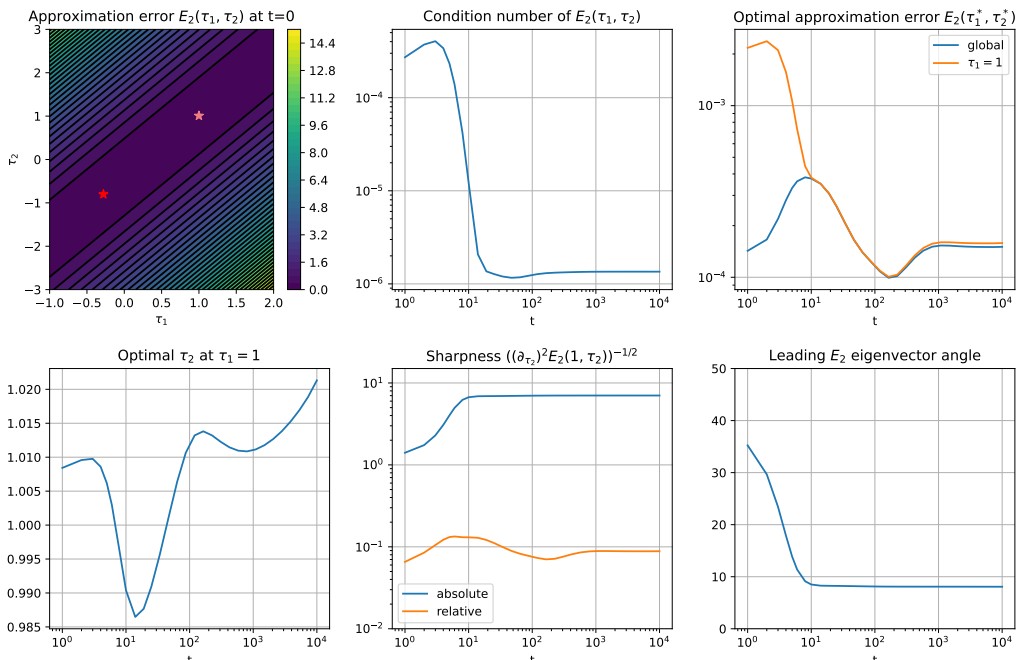

Figure 8: Various characteristics of relative approximation error $E_2(\tau_1, \tau_2)$ for SGD optimization on MNIST dataset. **Left top:** Dependence of $E_2(\tau_1, \tau_2)$ on $\tau_1, \tau_2$ at initialization. Red star denotes the global minimum, while light red star denotes the minimum on the line $\tau_1 = 1$. **Center top:** The ratio of smaller to bigger eigenvalue of the $2 \times 2$ matrix $\mathbf{E}$ which determines purely quadratic part of $E_2(\tau_1, \tau_2)$. **Right top:** Comparison of globally optimal approximation errors and approximation error optimal on the line $\tau_1 = 1$. **Left bottom:** Value $\tau_2$ minimizing $E_2(\tau_1, \tau_2)$ on the line $\tau_1 = 1$. **Center bottom:** length scale on line $\tau_1 = 1$ on which either approximation error increases by $1$ (*absolute*) when moving from optimum $\tau_2^*$, or increases by optimal approximation error (*relative*) when moving from optimum $\tau_2^*$. **Right bottom:** Angle between $\tau_2$ axis and leading egenvector of $\mathbf{E}$.

We measure closeness of true and approximated noise covariances with relative quadratic trace distance

$$E_2(\tau_1, \tau_2) = \frac{\text{Tr}\left[\left(\mathbf{\Sigma}_t - \mathbf{\Sigma}_t^{SE}(\tau_1, \tau_2)\right)^T \left(\mathbf{\Sigma}_t - \mathbf{\Sigma}_t^{SE}(\tau_1, \tau_2)\right)\right]}{\text{Tr}\left[\mathbf{\Sigma}_t^T \mathbf{\Sigma}_t\right]} \tag{241}$$

Note that linearity of $\mathbf{\Sigma}_t^{SE}(\tau_1, \tau_2)$ implies that $E_2(\tau_1, \tau_2)$ is quadratic in $\tau_1, \tau_2$. We analyze quadratic form $E_2(\tau_1, \tau_2)$ in Figure F.4. From the bottom left part we see that the ellipses corresponding to level sets of $E_2(\tau_1, \tau_2)$ are extremely stretched, meaning that there exist a direction along which approximation error changes very weekly. To check that conclusion for during the whole SGD optimization, we plot in center top part the ratio of smaller to bigger principal axes of level sets ellipses and see that during optimization the stretching only increases. One particular consequence is that there is a ambiguity in choosing the optimal $\tau_1, \tau_2$, which means that we can safely put $\tau_1 = 1$ and vary only $\tau_2$. This conclusion is illustrated on right top part where both global and on $\tau_1 = 1$ optimal aprroximation error are very small $\lesssim 10^{-3}$ during the whole optimization. Finally, we observe left bottom part of F.4 that optimal $\tau_2$ is extremely close to $1$, which well agrees with the conclusion of the first approach based on predicting the evolution of loss values and stability regions.

## F.5 ADDITIONAL EXPERIMENTS

To study the validity of the theoretical framework developed in this work in more practically interesting setting we computed empirical NTKs [3] of the pretrained models from torchvision and conducted experiments with model training in the linearized regime and spectrally expressible approximation with $\tau = \pm 1$ on the CIFAR10 dataset Krizhevsky et al. (2009). One can observe from Fig.9 that spectrally expressible approximation can accurately describe the averaged dynamic of the linearized model. The NTK eigendecomposition and target expansion in the NTK eigenvectors follow again power-law decay rule. In the experiments below we used popular ResNet-18 and MobileNet-V2 models pretrained on the ImageNet dataset Russakovsky et al. (2015).

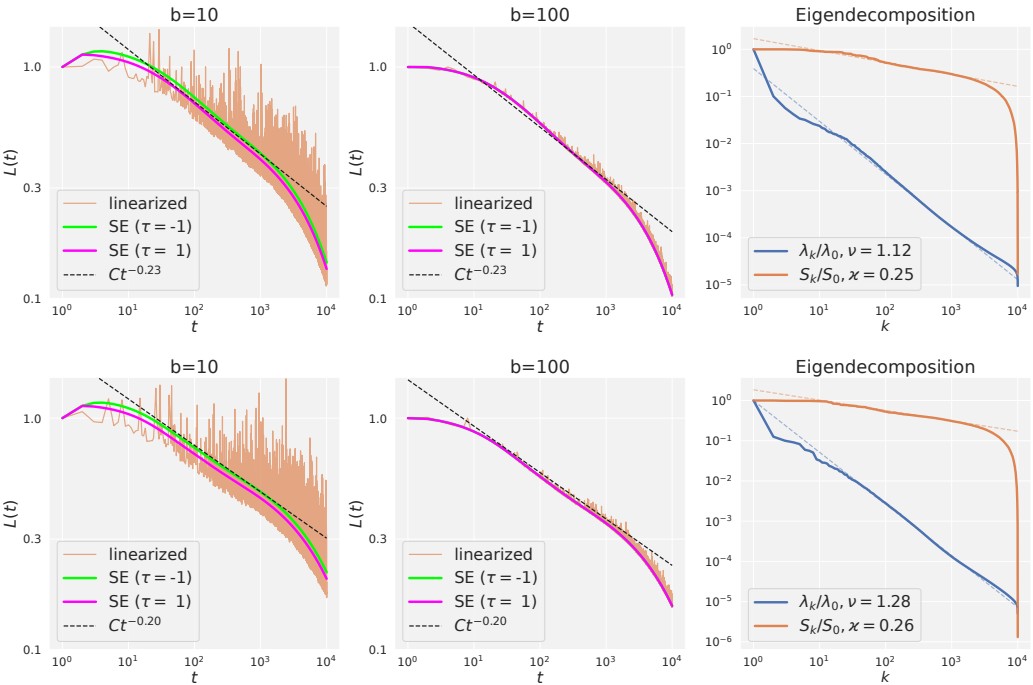

Figure 9: Loss trajectories and spectrum for linearized regression of prertrained ResNet-18 (**Top**) and MobileNet-V2 (**Bottom**). All notations follow Fig.1. The power-law exponents are $\nu \simeq 1.12, \varkappa \simeq 0.25, \zeta \simeq 0.23$ for ResNet-18 and $\nu \simeq 1.28, \varkappa \simeq 0.26, \zeta \simeq 0.20$ for MobileNet-V2.

In addition to the image and synthetic data we ran experiments with one hidden layer fully-connected neural network and with the approximate dynamics for two tabular datasets from UCI Machine Learning Repository: Bike Sharing dataset [4] and SGEMM GPU kernel performance dataset [5]. The former is a regression problem, namely, prediction of the count of rental bikes per hour or per day, and we've selected the per hour dataset. There are 17389 instances in the dataset and 16 features per input sample. The latter is a regression problem as well, there are 14 features, describing the different properties of the computation hardware and software and the outputs are the execution times of 4 runs of matrix-matrix product for two matrices of size $2048 \times 2048$. We predict the average of four runs. There are 241600 instances in the dataset. See results in Fig. 10.

In order to investigate the problem of the loss convergence and determination of the critical learning rate we performed experiments with different batch sizes $b = 1, 10, 1000, 1000$ on MNIST and synthetic data with $\nu = 1.5, \varkappa = 0.75$ (the 'signal' regime) and with h $\nu = 1.5, \varkappa = 3.0$ (the 'noisy' regime). For each value of momentum $\beta$ we plot the critical value of learning rate $\alpha_{\text{crit}}$ such that the dynamics starts to diverge. See results in Figures 12, 13 and 14. Moreover, we carried out

---

[3] We adopted the open source implementation `https://github.com/aw31/empirical-ntks`
[4] `https://archive.ics.uci.edu/ml/datasets/Bike+Sharing+Dataset`
[5] `https://archive.ics.uci.edu/ml/datasets/SGEMM+GPU+kernel+performance`

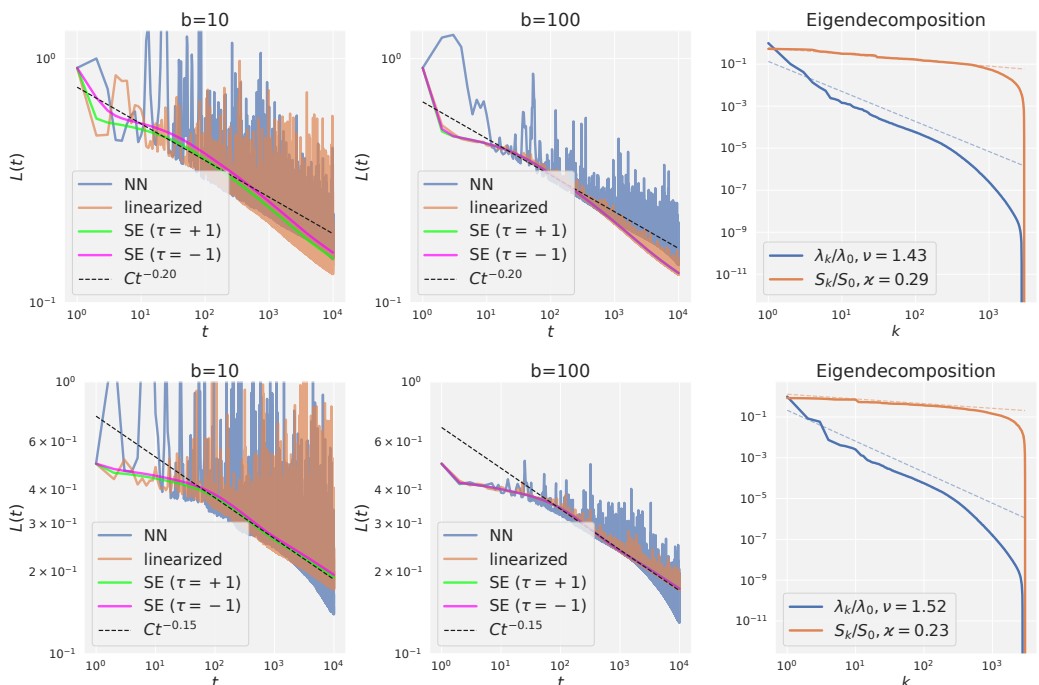

Figure 10: Loss trajectories and spectrum for Bike Sharing dataset (**Top**) and Sgemm GPU kernel performance dataset (**Bottom**) . All notations follow Fig.1. The power-law exponents are $\nu \simeq 1.43, \varkappa \simeq 0.29, \zeta \simeq 0.20$ for Bike Sharing dataset and $\nu \simeq 1.52, \varkappa \simeq 0.23, \zeta \simeq 0.15$ for Sgemm GPU kernel performance dataset.

the same experiment with empirical NTK computed on CIFAR10 for pretrained ResNet18 and batch sizes $b = 10, 100$ (see Figure 11).

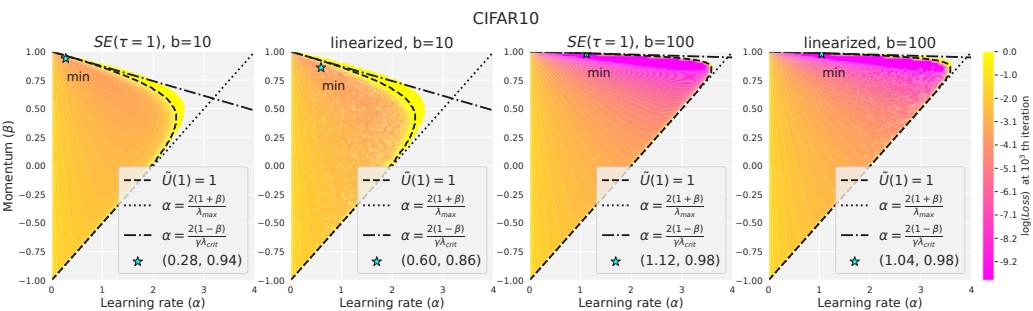

Figure 11: $L(t = 10^4)$ for different learning rates $\alpha$, momenta $\beta$ and batch sizes $b$ on CIFAR10. Legend and notations follow Fig. 2.

## G   MINI-BATCH SGD VS. SGD WITH ADDITIVE NOISE

In the paper we considered SGD with the natural noise structure induced by random sampling of batches. However, it is instructive to consider SGD with an additive noise model and observe how this surrogate noise model qualitatively changes optimization. Specifically, we will assume that on each iteration, a Gaussian vector with covariance $\mathbf{G}$ is added to the true loss gradient: $\nabla_{\mathbf{w}}^{(t)} L(\mathbf{w}) = \nabla_{\mathbf{w}} L(\mathbf{w}) + \mathbf{g}_t, \ \mathbf{g}_t \sim \mathcal{N}(0, \mathbf{G})$. For quadratic problems the SGD iteration becomes

$$\begin{pmatrix} \Delta \mathbf{w}_{t+1} \\ \mathbf{v}_{t+1} \end{pmatrix} = \begin{pmatrix} \mathbf{I} - \alpha_t \mathbf{H} & \beta_t \mathbf{I} \\ -\alpha_t \mathbf{H} & \beta_t \mathbf{I} \end{pmatrix} \begin{pmatrix} \Delta \mathbf{w}_t \\ \mathbf{v}_t \end{pmatrix} - \alpha_t \begin{pmatrix} \mathbf{g}_t \\ \mathbf{g}_t \end{pmatrix} \tag{242}$$

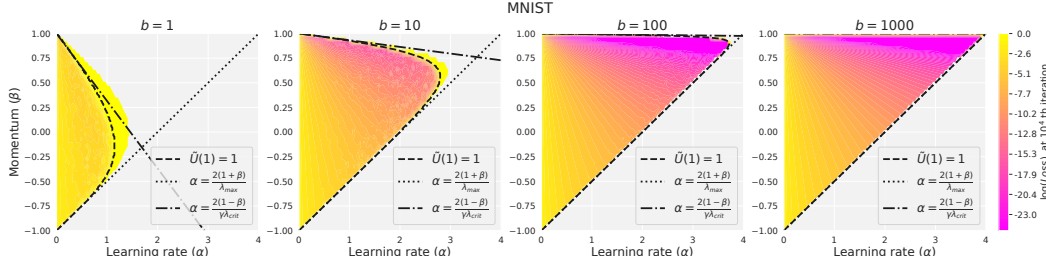

Figure 12: $L(t = 10^4)$ for different learning rates $\alpha$, momenta $\beta$ and batch sizes $b$ on MNIST. Legend and notations follow Fig. 2

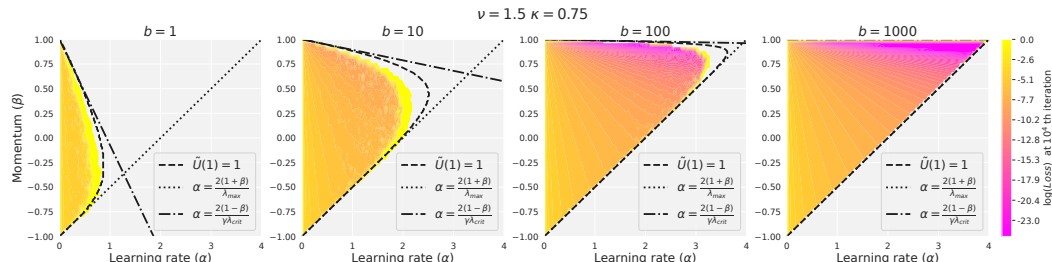

Figure 13: $L(t = 10^4)$ for different learning rates $\alpha$, momenta $\beta$ and batch sizes $b$ on synthetic dataset with $\nu = 1.5$ and $\varkappa = 0.75$. Legend and notations follow Fig. 2

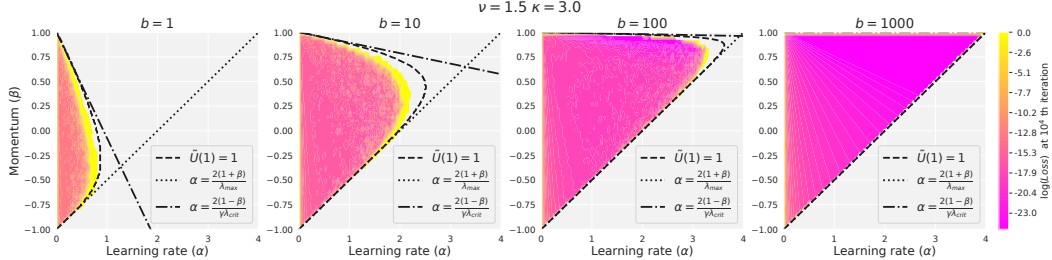

Figure 14: $L(t = 10^4)$ for different learning rates $\alpha$, momenta $\beta$ and batch sizes $b$ on synthetic dataset with $\nu = 1.5$ and $\varkappa = 3.0$. Legend and notations follow Fig. 2

Then, due to $\mathbb{E}[\mathbf{g}_t] = 0$, the dynamics of first moments is identical to that of noiseless GD, like in the case of sampling noise. However, dynamics of second moments is different:

$$\mathbf{M}_{t+1} = \begin{pmatrix} \mathbf{I} - \alpha_t \mathbf{H} & \beta_t \mathbf{I} \\ -\alpha_t \mathbf{H} & \beta_t \mathbf{I} \end{pmatrix} \mathbf{M}_t \begin{pmatrix} \mathbf{I} - \alpha_t \mathbf{H} & \beta_t \mathbf{I} \\ -\alpha_t \mathbf{H} & \beta_t \mathbf{I} \end{pmatrix}^T + \alpha_t^2 \mathbf{G} \begin{pmatrix} 1 & 1 \\ 1 & 1 \end{pmatrix} \tag{243}$$

Considering again constant learning rates $\alpha_t = \alpha, \beta_t = \beta$, we see that by making a linear shift of second moments we get the noiseless dynamics:

$$\mathbf{M}_{t+1} - \mathbf{M}^\infty = \begin{pmatrix} \mathbf{I} - \alpha_t \mathbf{H} & \beta_t \mathbf{I} \\ -\alpha_t \mathbf{H} & \beta_t \mathbf{I} \end{pmatrix} (\mathbf{M}_t - \mathbf{M}^\infty) \begin{pmatrix} \mathbf{I} - \alpha_t \mathbf{H} & \beta_t \mathbf{I} \\ -\alpha_t \mathbf{H} & \beta_t \mathbf{I} \end{pmatrix}^T, \tag{244}$$

where $\mathbf{M}^\infty$ is the "residual uncertainty" determined from the linear equation

$$\mathbf{M}^\infty - \begin{pmatrix} \mathbf{I} - \alpha_t \mathbf{H} & \beta_t \mathbf{I} \\ -\alpha_t \mathbf{H} & \beta_t \mathbf{I} \end{pmatrix} \mathbf{M}^\infty \begin{pmatrix} \mathbf{I} - \alpha_t \mathbf{H} & \beta_t \mathbf{I} \\ -\alpha_t \mathbf{H} & \beta_t \mathbf{I} \end{pmatrix}^T = \alpha^2 \mathbf{G} \begin{pmatrix} 1 & 1 \\ 1 & 1 \end{pmatrix} \tag{245}$$

For simplicity, let's restrict ourselves to SGD without momentum ($\beta = 0$). Then in the eigenbasis of $\mathbf{H}$ we have

$$C_{kk,t+1} - C_{kk}^\infty = (1 - \alpha \lambda_k)^2 (C_{kk,t} - C_{kk}^\infty), \quad C_{kk}^\infty = \frac{\alpha}{\lambda_k (2 - \alpha \lambda_k)} G_{kk} \tag{246}$$

Thus, optimization converges with the speed of noiseless GD, but the limiting loss is nonzero:

$$\lim_{t \to \infty} L(\mathbf{w}_t) = L^\infty = \frac{1}{2} \sum_k \lambda_k C_{kk}^\infty > 0. \tag{247}$$

The key difference of SGD with sampling noise (4), (5) is the multiplicative character of the noise, in the sense that it vanishes as the optimization converges to the true solution and $M_t$ becomes close to 0. To summarize, the two largest differences between SGD with additive and multiplicative noises are

1. Loss converges to a value $L^\infty > 0$ for additive noise and to 0 for multiplicative noise.

2. Convergence rate is independent of noise strength for additive noise, but significantly depends on it for multiplicative noise.

## H   NOISELESS CONVERGENCE CONDITION

The convergence condition $\alpha < \frac{2(1+\beta)}{\lambda_{\max}}$ for noiseless GD is known (see e.g. (Tugay & Tanik, 1989)). For completeness, below we present its derivation in our setting.

First observe that, in the noiseless GD, different eigenspaces of $\mathbf{H}$ evolve independently from each other. Thus we need to separately require convergence in each eigenspace of $\mathbf{H}$. Denoting the eigenvalue of the chosen subspace by $\lambda$, we write the GD iteration as

$$\begin{pmatrix} \Delta\mathbf{w}_{t+1} \\ \mathbf{v}_{t+1} \end{pmatrix} = \mathbf{S} \begin{pmatrix} \Delta\mathbf{w}_{t+1} \\ \mathbf{v}_{t+1} \end{pmatrix}, \quad \mathbf{S} = \begin{pmatrix} 1 - \alpha\lambda & \beta \\ -\alpha\lambda & \beta \end{pmatrix} \tag{248}$$

The characteristic equation for eigenvalues $s$ of matrix $\mathbf{S}$ is

$$s^2 - (1 - \alpha\lambda + \beta)s + \beta = 0, \tag{249}$$

yielding solutions

$$s_{1,2} = \frac{1 - \alpha\lambda + \beta \pm \sqrt{(1 - \alpha\lambda + \beta)^2 - 4\beta}}{2} \tag{250}$$

The GD dynamics converges iff $|s_1| < 1$ and $|s_2| < 1$. First note that $|s_1||s_2| = |\beta|$ and therefore for $|\beta| \geq 1$ at least one of $|s_{1,2}|$ will be $\geq 1$ and GD will not converge. Therefore we only need to consider $\beta \in (-1, 1)$.

When $\beta \in \left((1 - \sqrt{\alpha\lambda})^2, (1 + \sqrt{\alpha\lambda})^2\right)$, the eigenvalues $s_{1,2}$ are complex and $|s_1| = |s_2| = \sqrt{|\beta|}$. The convergence condition in this region is $|\beta| < 1$ which is automatically satisfied.

Now consider the region where $s_{1,2}$ are real. As $s_1 > s_2$, the convergence conditions in this region are $s_1 < 1$ and $s_2 > -1$. Note that for $\beta \leq 0$ the eigenvalues $s_{1,2}$ are always real, while for $\beta > 0$ they are real for $\lambda \in [0, \infty) \setminus (\lambda_1, \lambda_2)$ with $\lambda_1 = (1 - \sqrt{\beta})^2/\alpha$ and $\lambda_2 = (1 + \sqrt{\beta})^2/\alpha$.

Let us first check the condition $s_1 < 1$. The equation $s_1(\lambda) = 1$ gives $\lambda = 0$. For $\beta > 0$ we have $s_1(\lambda = 0) = 1$, $s_1(\lambda = \lambda_1) = \sqrt{\beta} < 1$ and $s_1(\lambda = \lambda_2) = -\sqrt{\beta} < 1$ and therefore $s_1(\lambda) < 1$ on $(0, \infty) \setminus (\lambda_1, \lambda_2)$ due to continuity of $s_1(\lambda)$. For $\beta < 0$ the condition $s_1(\lambda) < 1$ is always satisfied because $s_1(\lambda)$ is decreasing with $\lambda$.

Proceeding to condition $s_2 > -1$ we find that equation $s_2(\lambda^\star) = -1$ gives $\alpha\lambda^\star = 2(\beta + 1)$. Note that $\lambda^\star > \lambda_2$. For $\beta > 0$ we have $s_2(\lambda = 0) = \beta > -1$, $s_2(\lambda = \lambda_1) = \sqrt{\beta} > -1$, $s_2(\lambda = \lambda_2) = -\sqrt{\beta} > -1$ and $s_2(\lambda \to \infty) \to -\infty$. Therefore, due to continuity of $s_2(\lambda)$, we have $s_2(\lambda) > -1$ on $(0, \lambda^\star)/(\lambda_1, \lambda_2)$ and $s_2(\lambda) < -1$ on $(\lambda^\star, \infty)$. Similarly, for $\beta < 0$ we again have $s_2(\lambda) > -1$ on $(0, \lambda^\star)$ and $s_2(\lambda) < -1$ on $(\lambda^\star, \infty)$

Combining all the observations above, we get a single convergence condition for non-stochastic GD:

$$\alpha < \frac{2(1 + \beta)}{\lambda_{\max}}, \quad \beta \in (-1, 1) \tag{251}$$

where $\lambda_{\max}$ is the largest eigenvalues of $\mathbf{H}$.

