# OpenReview forum: "A view of mini-batch SGD via generating functions: conditions of convergence, phase transitions,  benefit from negative momenta."
_ICLR.cc/2023/Conference — ICLR 2023 poster_

### Official Review · Reviewer_NWpJ · 2022-10-23

**Confidence:** 2
**Correctness:** 4
**Technical Novelty And Significance:** 4
**Empirical Novelty And Significance:** 4
**Recommendation:** 8

**Clarity, Quality, Novelty And Reproducibility:**

This work is well-written and completely new in this field (including both its technique and results).

**Strength And Weaknesses:**

***Strenght***
This paper applies the generating function to analyze the higher moments of SGD noises. The new tool reveals the explicit stablility condition, which answer when SGD can converge and diverge. Most intersting part is that this paper introduces a situation where the negative momenta is better.  Also, the theoretical result perfectly explains the dynamic behavior of SGD over MNIST dataset.

***Weaknesses***
The loss function is relatively special, though I agree that this loss is widely used in modern machine learning tasks.


**Summary Of The Paper:**

This work builds a new convergence analysis framework for SGD algorithm (with momentum). It proposes a special family functions of ”Spectrally Expressible” approximations, which provides a new perspective to understand the behavior of classical SGD. A specific senario where a negative momenta can be the optimal choice is constructed.

**Summary Of The Review:**

I recommend to accept this paper as its results are impressive and really new to the community.

---

> ### Author Response · Authors · 2022-11-16
> **Response to the review**
>
> Thank you for the positive evaluation of our work! We are especially happy that you appreciated our explicit convergence condition.  This is indeed one of our main contributions, as it shows in a simple way the fundamental difference between mini-batch SGD (where effective learning rate is bounded) and full-batch GD (where it is unbounded).

---

### Official Review · Reviewer_LoFN · 2022-10-24

**Confidence:** 2
**Correctness:** 3
**Technical Novelty And Significance:** 3
**Empirical Novelty And Significance:** 2
**Recommendation:** 6

**Clarity, Quality, Novelty And Reproducibility:**

**Clarity and Quality**: As someone not familiar with this line of work, I personally found the manuscript dense and hard to parse. The focus of the manuscript is on the technical discussion, with the discussion reduced to enumerating the different regimes of the dynamics. The introduction largely overstates the importance of least squares.

**Novelty**: I am not familiar with the related literature, so I have limited myself to browsing some works on momentum for SGD on least squares.

**Reproducibility**: The appendix is well detailed, and code for reproducing the plots is provided by the authors.


**Strength And Weaknesses:**

**Comments and Questions**:

**[Q1]**: A series of recent works have shown that the behaviour of one-pass SGD on small neural networks [Goldt et al. '20] and random features [Goldt et al. '22] as well as full-batch learning [Loureiro et al. '21] and mini-batch SGD [Bordelon, Pehlevan '21] on linear models trained on fixed feature maps are well approximated by an equivalent model with Gaussian features in the high-dimensional limit. As mentioned in the manuscript, in this case the "Spectrally expressible" approximation is exact. By the way, all of the aforementioned works contain experiments of Gaussian equivalence on MNIST, which is consistent with the authors observation that their SE approximatiom works well in this case. I wonder whether Gaussian equivalence is only sufficient for SE to hold or also necessary. In other words: do the authors have an example where the SE approximation holds but the phenomenology is different from having Gaussian features?

**[Q2]**: The sentence in the introduction:
> The goal of the present work is to derive explicit and easy-to-use analytic evolution laws for the SGD (1) applicable to modern deep neural networks trained on real world datasets.

is a huge overstating, not only given the theoretical setting considered in this work (least-squares regression) but also the lack of empirical evidence for that (i.e. checking the theory on MNIST). Moreover, the following sentence:

**[Q3]**: Moreover, in the next sentence in the introduction:
>First, in many (though not all) cases the nonlinear network training is reasonably well approximated (Lee et al., 2020) by the linearized NTK regime (Jacot et al., 2018), in which the loss L is quadratic with respect to parameters w.

Where in (Jacot et al., 2018) it is shown that NTK = square loss?

**[Q4]**: I am not very familiar with the related literature. But browsing some papers for this review I stumbled upon [Paquette, Paquette '21], who studies SGD with momentum in least squares with Gaussian random design. Their conclusion is that in the strong convex case momentum does not improve the performance of SGD, while in the non-strongly convex case it is possible to get considerable improvement. Is this consistent with taking a Marchenko-Pastur distribution for the Hessian matrix here? How does this conclusion changes as a function of the choice of feature map?

**References**:

[[Bordelon, Pehlevan '21]](https://arxiv.org/abs/2106.02713) B Bordelon, C Pehlevan. *Learning Curves for SGD on Structured Features*. arXiv: 2106.02713 [cs.LG]

[[Loureiro et al. '21]](https://proceedings.neurips.cc/paper/2021/hash/9704a4fc48ae88598dcbdcdf57f3fdef-Abstract.html) B Loureiro, C Gerbelot, H Cui, S Goldt, F Krzakala, M Mezard, L Zdeborová. *Learning curves of generic features maps for realistic datasets with a teacher-student model*. Part of Advances in Neural Information Processing Systems 34 (NeurIPS 2021).

[[Goldt et al. '20]](https://journals.aps.org/prx/abstract/10.1103/PhysRevX.10.041044) S Goldt, M Mézard, F Krzakala, L Zdeborová. *Modelling the influence of data structure on learning in neural networks: the hidden manifold model*. Phys. Rev. X 10, 041044 – Published 3 December 2020.

[[Goldt et al. '22]](https://proceedings.mlr.press/v145/goldt22a.html) S Goldt, B Loureiro, G Reeves, F Krzakala, M Mezard, L Zdeborova. *The Gaussian equivalence of generative models for learning with shallow neural networks*. Proceedings of the 2nd Mathematical and Scientific Machine Learning Conference, PMLR 145:426-471, 2022.

[[Paquette, Paquette '21]](https://papers.nips.cc/paper/2021/hash/4cf0ed8641cfcbbf46784e620a0316fb-Abstract.html) C Paquette, E Paquette. *Dynamics of Stochastic Momentum Methods on Large-scale, Quadratic Models*.  Part of Advances in Neural Information Processing Systems 34 (NeurIPS 2021).

**Summary Of The Paper:**

This work investigates stochastic gradient descent with momentum for the least squares problem at fixed design. In particular, it derives closed-form equations for the evolution of second moments of the weights and momentum parameters under a "spectrally expressible" (SE) approximation, which allow one to compute the trajectory of observables throughout the dynamics from the initial weight covariance matrix and covariance of the features (i.e. the Hessian). In particular, this approximation holds for translational invariant kernels in a grid and for Gaussian features, and Theorem 1 provides different equivalent characterisations of the proposed SE approximation. By rewriting the evolution and identifying it with an interacting gas of rank-one operators, the authors separate the contributions of the signal and sampling noise.  The authors study this evolution equations under the SE approximation and derive different regimes for the dynamics as a function of the learning rate, the momentum strength and the spectrum of the features covariance. In particular, it is observed that an unusual choice of small negative momentum can be desirable in the noise dominated phase.


**Summary Of The Review:**

This work provides a characterisation of the SGD dynamics with momentum for the least squares problems with fixed design. The introduction overstates the importance of the studied setting. The manuscript is dense and hard to parse, and I feel that developing better the discussion and highlighting better the main contributions would improve the reading. For a non-expert, it is not very clear what are the challenges and the progress made the manuscript with regard to previous literature. Therefore, I believe this work could benefit from a revision.

---

> ### Author Response · Authors · 2022-11-16
> **Response to comments**
>
> Thank you for your feedback!
> We have revised the paper and believe to have improved the writing in various aspects, please refer to our general comment for more details. Some changes addressing your concerns are:
>
> - ``*The introduction largely overstates the importance of least squares.*''
>     We believe that the revised introduction does not overstate the importance of least square. We additionally motivate the use of least squares in the context of deep learning by the observation that networks can often be linearized in the late stages of training (Fort et al., 2020).
>
> - ``*...developing better the discussion and highlighting better the main contributions would improve the reading. For a non-expert, it is not very clear what are the challenges and the progress made the manuscript with regard to previous literature.*''
>       - We believe that the revised introduction and discussion clearly list multiple advances made by our work.
>       - In the Related Work section (A) we provide a detailed and comprehensive discussion of our work in comparison to previous research.
>
> Regarding your questions:
> 1. This is an interesting question.
>      - On the one hand, we definitely see cases when non-Gaussian ($\tau\ne -1$) SE approximation is more appropriate than the Gaussian one. This can be seen theoretically for translation-invariant models or models with pointwise losses independent of feature eigencomponents (the first two items in our current list of evidence for SE in section 3). We also observe empirically that non-Gaussian values $\tau\ge 0$ are more suitable to at least some tasks  (this is especially well seen in Figure 7 where we perform averaging to better estimate the mean losses).
>      - On the other hand, we don't see any fundamental difference between the Gaussian and non-Gaussian cases. The exponents in our power-law loss asymptotics do not depend on $\tau$; only the leading coefficients depend on it. The phase structure and other general effects that we discuss, such as the dynamic phase transitions and the effects of momenta, also only include $\tau$ through some coefficients but not in any fundamental way (at least in the case $\tau> 0$ that we limit ourselves to for technical reasons).
> 2.
>      - We have substantially revised the paper and removed all controversial claims.
>      - On the other hand, we have added new experiments with ResNet/MobileNet on CIFAR10 that agree well with our theory. We have also tried to better explain what exactly agrees with what (e.g. in Figure 1). We believe that our paper now has a sufficient amount of empirical evidence.
> 3. That was indeed a wrong statement, now removed -- thank you for pointing this out.
> 4.
>      - Thank you for bringing that paper to our attention. The setting in that paper is actually very different from ours. We work with a fixed Hessian that is a compact operator with a tail of discrete eigenvalues converging to 0, and a non-random initial condition for SGD. We don't perform any limits involving growing numbers of degrees of freedom and rescaling of the learning rate. It's not obvious how to relate the Marchenko-Pastur distribution to our particular spectral power laws on which our momentum statements depend very sensitively (presumably only the MP with the critical aspect ratio would be suitable for that). So the differences between our settings are so significant that we admit that we find it not easy to establish a correspondence between our results.
>      - In our setting, the benefit of momentum is determined not by the presence of spectral gap, but rather by how fast the target expansion coefficients converge to 0 (see e.g. our Figure 4, in which faster convergence corresponds to larger $\varkappa$ and lower optimal $\beta$). One can probably say that the strongly convex case corresponds to eigencoefficients vanishing for very small eigenvalues  -- in this respect our conclusions roughly agree.

---

> > ### Comment · Reviewer_LoFN · 2022-11-27
> > **Post-discussion update**
> >
> > Dear authors,
> >
> > Thank you for your rebuttal. After a careful reading of the other reviews and your replies, I am raising my score from 5 to 6. I am happy to see that the authors have welcomed most of the reviewers suggestions, and I believe the revised version has substantially improved over the original submission. In particular, the addition of new experiments strengths the relevance of the theoretical setting, and the new introduction is more to the point.

---

### Official Review · Reviewer_KUvw · 2022-11-02

**Confidence:** 3
**Correctness:** 4
**Technical Novelty And Significance:** 2
**Empirical Novelty And Significance:** Not applicable
**Recommendation:** 6

**Clarity, Quality, Novelty And Reproducibility:**

Clarity: the problem and setting are explained well. However, the intuitions and motivation of the approximation they use is not explained enough.

Quality: the mathematical equations are sound and look correct as much as I checked.

Novelty: the major novelty of the paper is using the generating function to analyze the loss trajectory.


**Strength And Weaknesses:**

The major strength of the paper is that its analytical framework matches well enough with the empirical observation.

The major weakness of the paper is that the provided framework is applicable for too limited cases due to their assumptions and approximations. Besides the paper is not well-written and easy to follow.


**Summary Of The Paper:**

This paper develops a theoretical framework to analysize the noise averged properties of  for mini-batch SGD with momentum (bSGDM) for the linear models when learning rate, mini-batch size and momentum parameter is fixed. To handle the noise terms appearing in the analysis, they approximate that variance term with specterally exprissible approximations. This approximation allows them to form a generative function for the loss function trajectory. Analysing this generating function 1- they specify the noise dominant and signal dominant phase in loss trajectory 2- when Hessian spectral distribution follows power low distribution, they analyze the stability of loss function and show when divergence and convergence happens 3- show that negative momentum sometimes can helps. They give some theoretical approximation for optimal parameter values and the phase transition times. Empirically they show that their analysis matches the practice for MNIST and some artificial data sets.


**Summary Of The Review:**

The major problem I have with the paper is its writing. I believe it is hard to understand and read the paper easily. Here are some examples:

1- there isn’t any part in the paper to explains the notation and operations used in the formula. For example there U \in O(N) where U is a matrix.

2- You claim that you drive exat loss asymptotics for bSGDM however you use a lot of approximations in your analysis. So making such a general claim is not accurate.

3- You have some terms which are not defined explicitly. For example, \lambda_crit or regularization parameter is not defined anywhere.

4-  The idea behind approximating eq 5 by eq 6 isn’t explained well. The paper says this is due to the existence of non-spectral details in eq 5 but this is not enough. It would be helpful to show these details and show how eq 7 would solve it.

5- In eq 8 which is an approximate for eq 5, there are cross terms between feature maps of two different samples that don’t exist in eq 5 and it can remove the stochasticity from the dynamics. So the question is how good is this approximation i.e. the upper bound of the difference of these two eqs.

6- For your approximation of L(t) in “Transitions between phases”, it adds eq 27a and eq 27b. However, depending on \zetta and \nu one of these equations can happen for L(t). So why the sum of them is also a good approx for L(t)?

7- One of the claimed contributions is that “the optimal convergence rate can be achieved at negative momenta”. However, the analysis in the “negative momenta” section is hand-wavy and for a very limiting case.

---

> ### Author Response · Authors · 2022-11-16
> **Response to the comments**
>
> Thank you for your feedback!
> We believe that we have improved writing in the revised version of the paper. Please also refer to our general comment. Regarding your specific points:
> 1. We have tried to remove everywhere ambiguous notation (such as $O(N)$) and generally ensure that the meaning of all formulas is clear. We decided not to include a separate section with notation because the paper has much different notation in different sections, and so we find it more efficient to introduce notation along the way.
> 2.
>     - We have completely rewritten the introduction and removed all controversial claims.
>     - We have tried to restructure the paper to clarify our hierarchy of assumptions. This hierarchy is, in fact, rather simple: 1) Initial linear model; 2) Our SE approximation; 3) Power-law spectral assumptions; 4) Minor auxiliary assumptions such as the one relating individual losses to their cumulative sums (Eq. (28)).
>     - Different results in our paper require different levels of assumptions. Our first results (Section 3) provide various justifications for the SE approximation from several fundamental perspectives. Next results (Sections 4 and 5) use only SE approximation to solve the SGD dynamics and analyze its stability. Later results (Section 6) provide detailed analysis of various subtle effects and naturally require more detailed (power-law) assumptions. At the same time, we show experimentally that all the assumptions are reasonable and the results agree well with experiment. In the revision we added experiments with CIFAR10 and ResNet/MobileNet, and also tried to better clarify this agreement.
> 3. We explicitly defined $\lambda_\mathrm{crit}$ in Proposition 4 in the new version. We also made definition of the other terms more explicit (e.g. $C_\mathrm{signal}$ and $C_\mathrm{noise}$).
> 4.
>      - We added a new Proposition 7 and a remark in the main text clarifying a potential effect of non-spectral details. Roughly speaking, we show that, in general, non-spectral details do not allow us to predict the mean loss under SGD.
>      - At the same time, our main point is that in many cases we can still neglect non-spectral details (thus obtaining the SE approximation (7)). In the revised version we give an extended list of 5 theoretical and empirical arguments supporting this claim.
> 5. This is an interesting question. Actually, the role of (8) in our paper was not to approximate (5), but rather to consider a related (more general) family of noise terms and use it to gain a better understanding of the SE approximation. We slightly changed the wording in the paper to make it more clear. We admit that (8) is more of a mathematical device; we don't claim it to describe any real learning dynamics and at present don't know of a natural interpretation of the cross-terms.
> 6. As we mention after theorem 2, the loss is actually described by the sum of two terms resulting from the two terms in the expansion of $\tfrac{d}{dz}\widetilde L(z)$ (Eq. (27) in the revised version). These two loss terms can be approximated as $C_{\mathrm{signal}}t^{-\zeta}$ and $C_{\mathrm{noise}}t^{1/\nu-2}$, so that the full loss $L_t\approx C_{\mathrm{signal}}t^{-\zeta}+C_{\mathrm{noise}}t^{1/\nu-2}$. However, at large $t$ one of the terms (depending on the sign of $\zeta-2+1/\nu$) dominates the other, so we can write $L_t\approx \max(C_{\mathrm{signal}}t^{-\zeta}, C_{\mathrm{noise}}t^{1/\nu-2})$. The transition between phases occurs when the dominant term changes.
> 7.
>      - We have completely rewritten the section on positive vs. negative momenta in the main text. We believe that our analysis was actually solid, but the exposition was somewhat convoluted. We have now stated  precise Proposition 5 describing the effect, and moved technical details to the appendix.
>      - Moreover, we have generalized the statement for the "signal-dominated" regime: we show now that $\beta>0$ are beneficial in this regime for all parameters $0<\tau\le 1, 0\le \gamma\le 1.$  We believe that this confirms the generality of the effect. As for the improvement from $\beta<0$ in the "noise-dominated" regime: our goal was rather to show that this can happen at least in some cases, and can be analytically predicted -- which we have achieved.

---

### Author Response · Authors · 2022-11-16
**Manuscript revision**

We thank the reviewers for useful comments. Following the feedback, we have substantially revised the paper. The main changes are:
- We have generally revised the text for better clarity and accuracy:
    - We completely reworked the introduction, excluding the controversial statements and also providing a better motivation for our approach and explaining its novelty.
    - We restructured the main section of the paper so that each section corresponds to a single stage in our framework. In particular, we put all the results which require the power-law assumption into a single section. (Our stability analysis does not require this assumption.)
    - We moved some of the technical details and derivations into the appendix, and added to the main text more theorems and discussions of the results  instead.
- We added new experiments with pretrained ResNet-18 and MobileNet-V2 applied to CIFAR10
    and showed that our SE approximation and power-law ansatz work well in these cases too (See section E.5, Figures 9 and 11).
- To make our theory more convincing, we have added several new theoretical results and strengthened or formalized previous results:
    - Added a new estimate of the impact of non-spectral details (Sections 3 and D.1);
    - Added an argument that SE approximation with $\tau_1=\tau_2=1$ results if pointwise losses are statistically independent of the eigencomponents (Sections 3 and D.5);
    - Added Proposition 4, rigorously establishing the necessity and tightness of the simplified convergence condition in terms of the effective learning rate;
    - Moved Proposition 5 on the effect of momenta into the main text and generalized its "signal-dominated" part to all $0\le \tau\le 1,0<\gamma\le 1$.

---

### Decision · Program_Chairs · 2023-01-20

**Decision:**

Accept: poster

**Justification For Why Not Higher Score:**

None of the reviewers confidently wanted to highlight this paper.

**Justification For Why Not Lower Score:**

All reviewers recommend acceptance.

**Metareview: Summary, Strengths And Weaknesses:**

Summary: This paper develops a theoretical framework to analysize the noise averged properties of for mini-batch SGD with momentum (bSGDM) for the linear models when learning rate, mini-batch size and momentum parameter is fixed. To handle the noise terms appearing in the analysis, they approximate that variance term with specterally exprissible approximations. This approximation allows them to form a generative function for the loss function trajectory. Analysing this generating function 1- they specify the noise dominant and signal dominant phase in loss trajectory 2- when Hessian spectral distribution follows power low distribution, they analyze the stability of loss function and show when divergence and convergence happens 3- show that negative momentum sometimes can helps. They give some theoretical approximation for optimal parameter values and the phase transition times. Empirically they show that their analysis matches the practice for MNIST and some artificial data sets.

All reviewers agreed that this paper makes an interesting contribution to ICLR. Some concerns were expressed about the clarity of the writing in the original reviews, but the authors have addressed them nicely in a revision for which both reviewers LoFN and KUvw thought improved significantly the paper and updated their recommendation (after discussion) to acceptance.


**Note From Pc:**

if the above contains the word "oral" or "spotlight" please see: "oral" presentation means -> notable-top-5% and "spotlight" means -> notable-top-25%. As stated in our emails, we are disassociating presentation type from AC recommendations